# Practical Mechanism via Simple Input Control for Fault-Tolerant Spiking Neural Networks

## Abstract

Spiking Neural Networks (SNNs) attract researchers due to their energy-efficient operations in neuromorphic devices. Despite their energy efficiency, hardware-implemented SNNs in neuromorphic devices are vulnerable to hardware faults, which impair the functionality of learnable parameters (e.g., Stuck-At-Faults (SAFs) in synaptic weights). This impairment reduces the capacity to absorb information. When input data contains information exceeding the capacity, SNNs may not absorb information correctly, referred to as **the bottleneck problem**. Existing approaches have relied on complex algorithms or direct modification to most synaptic weights in hardware-implemented SNNs, limiting their practicality in neuromorphic devices. This paper proposes a simple yet effective input control mechanism to address the problem, grounded in a thorough motivation study. Our mechanism divides the input samples into small fragments, following the best fragmentation strategy, derived by analyzing the characteristics of the input samples and diagnosing the current influence of faults. Experimental results demonstrate that our mechanism significantly enhances fault tolerance over existing methods, achieving these gains without complex algorithms or direct weight modification in various SNN models. Additionally, our mechanism improves the fault tolerance of SNN models implemented with actual hardware devices.

## 1 Introduction

Spiking Neural Networks (SNNs) attract researchers to develop neuromorphic devices that are necessary to implement Artificial Intelligence (AI) in low-end devices (Garaffa et al., 2021; Jeong et al., 2025). SNNs are third-generation neural networks that process data using spikes. They are well-suited for neuromorphic devices with limited power sources and biological operations, as SNNs consume less energy than other neural networks and exhibit high biological plausibility (Schuman et al., 2022; Pfeiffer & Pfeil, 2018). Although SNNs are essential for neuromorphic devices, hardware-implemented SNNs remain highly vulnerable to permanent hardware faults, which frequently occur in the electrical components of neuromorphic devices and significantly impair the SNNs' learning performance (Spyrou et al., 2021; Lee & Lim, 2023). Their fault vulnerability stems from the instability of hardware components and the fault sensitivity of SNNs' neuron models in neuromorphic devices (Garaffa et al., 2021).

Previous approaches to improve hardware-implemented SNNs' tolerance against hardware faults rely on complex algorithms to regulate abnormal neuronal activities or demand hardware reconfigurability to manage electric components directly in neuromorphic devices (Vu et al., 2019; Putra et al., 2022). Although these approaches have improved the fault tolerance of the SNNs, they exhibit the following problems that reduce their practicality in implementation.

***1. Algorithmic complexity in conventional mechanisms:*** The previous approaches require complex algorithms that are impractical for hardware implementation (Vu et al., 2019; Yang et al., 2022; Han et al., 2023). These approaches cannot work properly in neuromorphic devices that demand low-power operation and prevent stable operation (Basu et al., 2018). This is because the complex algorithms frequently malfunction due to unexpected events such as wrong input values and hardware faults (Liu et al., 2017).

***2. Difficulties for direct modifications to synapses:*** Modifying synaptic weights directly is essential for the previous approaches to improve hardware-implemented SNNs' fault tolerance, such as

synapse pruning and weight bounding. These approaches forcibly adjust whole synaptic weights and configurations in neuromorphic devices (Putra et al., 2022; Chen & Chakrabarty, 2021). However, external methods that enable direct modification require additional costs to design reconfigurable hardware (Garaffa et al., 2021; Takano & Amano, 2022; Putra et al., 2023).

Pragmatic approaches for meeting these two problems are necessary to improve SNN's fault tolerance in the real world. To address these limitations, we identify a critical issue that significantly degrades SNN performance in faulty neuromorphic devices through a detailed motivation study. We name this issue **the bottleneck problem**, severely damaging SNNs' usable learning capacity. Here, we explain how the bottleneck problem occurs. When faults appear in SNNs' synapses, the weights of the faulty synapses become fixed during training. This means that the capacity is reduced because the faulty synapses, which do not change their weight during training, cannot be used for memorizing data. Furthermore, the pre-activation value (linear combination) of spiking neurons lies at an abnormal point of the surrogate gradient function, causing a serious gradient vanishing problem, which results in the capacity degradation. With the low usable capacity of faulty SNNs, they cannot memorize input data properly when the data contains information exceeding the learning capacity. The bottleneck problem occurs in neuromorphic devices that use gradients for on-chip learning during training (Eslami et al., 2024; Martemucci et al., 2025). We aim to solve this problem as neuromorphic devices with gradient-based online learning frequently suffer from hardware faults despite the necessity of gradient-based online learning to enhance the data-processing ability of hardware-implemented SNNs (Spyrou et al., 2021; Lee & Lim, 2023; Rostami et al., 2022; Lagorce et al., 2015; Stewart et al., 2020; Cramer et al., 2022; Payvand et al., 2020; Renner et al., 2024; Yin et al., 2024).

Motivated by **flow control methods used in computer networks** (Kurose & Ross, 2012), we first propose a practical mechanism based on **simple input control** to solve the bottleneck problem based on **data fragmentation**. Our mechanism enhances the fault tolerance of SNNs by dividing input data samples into small fragments. Our control scheme addresses the constraint of SNNs' usable learning capacity in faulty neuromorphic devices by exploiting an effective fragmentation strategy for fault mitigation based on analysis of input images' characteristics and the influence of faults. The novelty of our mechanism is as follows. **Unlike the previous approaches, our mechanism does not require complex algorithms to control neuronal activities and hardware reconfigurability for modifying synapses directly.** This novelty is derived from the following features of our mechanism.

*1. Data sample division into small fragments:* Our algorithm fragments input data samples to decrease the input samples' size and shrink information in the input samples, leading the faulty SNN models to memorize the information despite their degraded usable learning capacity due to faults.

*2. Fragmentation method to minimize the adverse effect from faults:* To ensure that our mechanism provides the fragments that models can handle, we develop a fragmentation strategy that adapts fragment geometry (i.e., the cut angle) to mitigate fault-induced damage during the forward pass.

We develop our mechanism through a thorough motivation study with faulty SNN models. With our mechanism, various SNN models achieve significantly higher classification accuracy than models using previous approaches for fault mitigation under fault-injected conditions, while consuming less energy due to our simple approach. We additionally conduct experiments with real hardware SNNs built in a Field-Programmable Gate Array (FPGA) device. Our work has the following contributions.

- We develop a practical mechanism to enhance hardware-implemented SNNs' fault tolerance without complex algorithms and direct synapse modifications, which easily malfunction and demand high hardware reconfigurability.

- We present a concrete theoretical basis for our mechanism by mathematically and experimentally investigating how synaptic faults degrade the usable learning capacity of SNN models in our motivation study. Due to our detailed motivation study, we propose a theoretically sound fragmentation mechanism.

- We provide a rich set of evaluation results in various scenarios, including hardware environments. The evaluation results demonstrate that our mechanism, which is based on simple data fragmentation, improves the fault tolerance of SNNs more significantly than previous approaches.

## 2 BACKGROUNDS

### 2.1 SPIKING NEURAL NETWORKS

SNNs are third-generation neural networks motivated by the learning mechanisms of the human brain (Yao et al., 2023). In SNNs, spiking neurons fire and emit output spikes at every time interval, corresponding to each time step. Here, the time step is a unit of time for spike occurrence. The spiking neurons generate spikes only when the membrane potential of the neurons reaches a threshold.

Researchers use various neuron models to build SNNs. Among them, the Integrate-and-Fire (IF) model and Leaky Integrate-and-Fire (LIF) model, which have leakage in membrane potential unlike IF, are widely used (Moitra et al., 2023). Synaptic weights determine how input spikes from pre-synaptic neurons of the previous layer affect the post-synaptic neurons (Venkatesha et al., 2021). SNNs update the weights with two approaches: supervised and unsupervised learning rules. Supervised learning rules calculate gradients to update weights using surrogate gradient functions. Unsupervised learning rules utilize the time difference between pre-synaptic and post-synaptic activity to update weights.

SNNs are necessary to implement neuromorphic devices. They exhibit less energy consumption than conventional neural networks for the following reasons. First, spiking neurons fire and update synaptic weights only when a specific event occurs (Lee & Lim, 2024). The SNN's infrequent spike generation is associated with sporadic data processing, resulting in low power consumption. Second, SNNs replace complex Multiply-ACcumulate (MAC) operations with simple ACcumulate (AC) operations, eliminating weight multiplication to input data while accumulating input information.

### 2.2 SYNAPTIC FAULTS

Synaptic faults are persistent or transient defects in the synaptic weights of a connection. They distort pre-activations and inject biased or structured noise, which disrupts the learning process of SNN models. A representative case of them is the Stuck-At Faults (SAFs), where a weight is fixed to the highest (SA1) and lowest (SA0) synaptic weight, ignoring updates and introducing systematic bias (Vatajelu et al., 2019). In other words, SAFs make synapses permanently stuck at a max or min weight value, regardless of training or input. Note that SA0 and SA1 do not mean the weights are stuck at the exact values 0 and 1. Another common case is Random Weight Faults (RWFs). RWFs force synapses to randomly fluctuate around their original weight values due to thermal noise. They transiently remove intended synaptic connections or create unintended connections (Vatajelu et al., 2019). Additionally, Connectivity Error Faults (CEFs) permanently change the connections between synapses, ruining synaptic connections. These faults shift the pre-activation away from useful operating regions, shrink the effective gradient signal, and reduce usable learning capacity.

## 3 STATE OF THE ARTS

### 3.1 ANALYSIS ABOUT FAULTS IN NEUROMORPHIC DEVICES

Researchers have deeply investigated how faults affect neuromorphic devices. They inject faults into synapses and neurons of SNNs in neuromorphic devices, and how these faults ruin the classification performance of SNNs (Vatajelu et al., 2019). They build a memristive neuromorphic simulator and analyze how faults disturb data classification (Lee & Lim, 2023). In this study, researchers prove that the faults occurring in synapses correlated with important features of data samples influence the devices more severely. Researchers also apply various fault types in neuromorphic devices and study how spiking neurons act in detail (Ali El Sayed, 2021; Garaffa et al., 2021). However, these analysis studies overlook the excessive updates caused by hardware faults in neuromorphic devices.

### 3.2 MECHANISMS TO IMPROVE FAULT TOLERANCE OF HARDWARE-IMPLEMENTED SNNS

The conventional methods to improve the fault tolerance of hardware-implemented SNNs in neuromorphic devices rely on modifying faulty elements and implementing additional fault-mitigation architectures. Researchers utilize the error correction ability of binary codes in output decoding to enhance the fault tolerance of neural networks (Liu et al., 2019; Yu et al., 2023). They also induce spikes to avoid faults of hardware-implemented SNNs in neuromorphic devices to reduce

the bad effects caused by faults (Vu et al., 2019; Yang et al., 2022). They build a fault map of hardware-implemented SNNs to identify spiking neurons severely affected by faults and reduce these neurons' influence in neuromorphic devices (Putra et al., 2022; Wicaksana Putra et al., 2021; Yang et al., 2022). Additionally, they mask faulty elements by setting affected pre-trained weights to zero, then retrain with per-layer threshold (Siddique & Hoque, 2023). Researchers also employ self-recovering mechanisms from astrocytes in the human brain to neuromorphic devices. With their self-recovering ability, these approaches significantly strengthen neuromorphic devices' fault tolerance (Han et al., 2023; Varshika et al., 2023). Enhancing the astrocyte-based approaches, they augment SNNs with an astrocyte-inspired leaky integrator, stabilizing spiking dynamics and markedly improving fault tolerance (Yunusoglu et al., 2025). Lightweight approaches, such as suppressing abnormal pre-activation, removing fault-affected neurons, and tuning thresholds, have enhanced the fault tolerance either (Saha et al., 2023; Spyrou et al., 2021; Saha et al., 2024). Despite their enhancement of fault tolerance, these works require complex architectures based on complicated algorithms and neglect the reconfigurability of electric components in neuromorphic devices.

## 4 MOTIVATION STUDY

### 4.1 OVERVIEW

We have discovered that the faults cause the bottleneck problem with the following procedures.

1. Synaptic faults increase the magnitude of pre-activation $|z|$, which is a linear combination of inputs ($z = Wx + b$) (Berzal, 2025). This is because the pre-activation changes significantly when faults perturb the weights to $W + \Delta W$. Here, $W$ is the synaptic weights, $x$ is an input sample, and $b$ is a bias. $\Delta W$ indicates a weight change caused by faults.

2. As $|z|$ grows and moves away from the (spiking) threshold of spiking neurons, the surrogate gradient near the threshold collapses toward zero, so the learning signal cannot propagate backward.

3. The faulty weights are fixed to abnormal values, and the non-faulty weights barely change due to near-zero gradients. This problem significantly reduces SNNs' usable learning capacity, creating a bottleneck that prevents the model from fitting the data it should learn.

We provide thorough explanations, describing mathematically how faults fail the learning process of SNN models in Appendix C.

### 4.2 PRE-ACTIVATION MAGNITUDE INCREASE BY FAULTS

To demonstrate that the synaptic faults cause the pre-activation magnitude to increase, we inject the SAFs ($SA0 : SA1 = 1.75 : 9.04$) (Chen et al., 2017) and RWFs (representative permanent and transient faults) into 50% of synapses in a spiking Multi-Layered Perceptron (MLP) with 4 layers, VGG-7, and ResNet-18 with MNIST, CIFAR-10, and CIFAR-100 during training. We use the Poisson encoder to convert input into spikes. We observe the change in pre-activation magnitude of all neurons in the SNN models due to faults after training models with an Adam optimizer and Root-Mean-Square Error (RMSE) as a loss function. We obtain the experimental results by repeating experiments 10 times and present the pre-activation magnitude as a 95% confidence interval.

**Table 1:** Pre-activation magnitude of MLP, VGG-7, and ResNet-18 SNN models under fault injection.

|         | MLP(MNIST)        | VGG-7(CIFAR-10)  | RESNET-18(CIFAR-100) |
|---------|-------------------|------------------|----------------------|
| NOMINAL | $2.7 \pm 0.13$    | $8.75 \pm 2.08$  | $14.54 \pm 3.56$     |
| SAFs    | $393.42 \pm 10.89$| $273.66 \pm 13.69$| $140.03 \pm 15.77$  |
| RWFs    | $8.08 \pm 1.75$   | $205.73 \pm 10.27$| $103.41 \pm 11.8$   |

Table 1 compares the summation of the pre-activation magnitude of all spiking neurons in the MLP (LIF), VGG-7 (LIF), and ResNet-18 (IF) SNN models with and without SAF and RWF injection. Our experimental results show that SAFs and RWFs significantly increase the pre-activation magnitude around all neurons with fault-injected synapses.

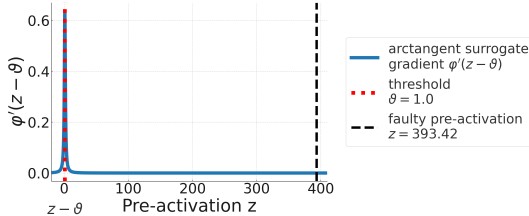

**Figure 1:** The value (average on the layers of the model) of surrogate gradient $\phi'(z - \vartheta)$ (arctangent) when we inject SAFs into the synapses of the MLP model.

## 4.3 GRADIENT COLLAPSE BY ABNORMAL PRE-ACTIVATION MAGNITUDE

We call the region where the surrogate derivative is non-negligible the *surrogate gradient corridor* and denote its half-width by $\delta$ (or threshold-aligned bound $z^\star$); outside this corridor, $\phi'(z - \vartheta) \approx 0$. As the pre-activation magnitude increases, the pre-activation value moves away from the corridor. This alignment error, due to an abnormal pre-activation magnitude, makes the surrogate gradient values nearly zero.

Figure 1 depicts the surrogate gradient function (arctangent) of an LIF neuron model and the position of pre-activation of the SAF-injected neurons in the MLP model. The surrogate gradient $\phi'(z - \vartheta)$ of the SAF-injected neurons is near zero, and the gradient of these neurons vanishes. This is because the following equation calculates the gradient: $\delta^{(l)} = \left(\nabla_{a^{(l)}} L\right) \odot \phi'\left(z^{(l)} - \vartheta\right)$ ($L$ is a loss function and $l$ is the index of a layer).

## 4.4 LEARNING ABILITY DEGRADATION BY GRADIENT COLLAPSE

We demonstrate that gradient vanishing due to faults causes degradation in the SNNs' learning ability.

**Table 2:** The gradients' L1 norm in 95% confidence interval upon all neurons and classification accuracy of MLP, VGG-7, and ResNet-18 SNN models under fault injection with 50% fault ratio.

|  | MLP(MNIST) | VGG-7(CIFAR-10) | RESNET-18(CIFAR-100) |
|---|---|---|---|
| NOMINAL (L1 NORM) | $11.67 \pm 1.71$ | $25.68 \pm 2.89$ | $7.76 \pm 1.92$ |
| NOMINAL (ACCURACY) | $97.57 \pm 0.38\%$ | $56.63 \pm 0.91\%$ | $25.59 \pm 1.24\%$ |
| SAFs (L1 NORM) | $0.01 \pm 0.00026$ | $1.83 \pm 0.16$ | $1.36 \pm 0.25$ |
| SAFs (ACCURACY) | $11.35 \pm 0.01\%$ | $9.99 \pm 0.08\%$ | $5.04 \pm 1.07\%$ |
| RWFs (L1 NORM) | $65.86 \pm 10.57$ | $2.18 \pm 0.45$ | $0.49 \pm 0.07$ |
| RWFs (ACCURACY) | $79.8 \pm 8.74\%$ | $22.28 \pm 5.34\%$ | $1.5 \pm 0.51\%$ |

Table 2 shows the gradients' L1 norm upon all neurons in the SNN models during training and the classification accuracy of the models with the various datasets after testing. The classification accuracy is proportional to the L1 norm of the gradient under fault injection with 50% fault ratio. This point demonstrates that vanishing surrogate gradients stall weight updates, preventing the network from making new decision boundaries and thus directly reducing its learning capacity (occurrence of the bottleneck problem). Interestingly, the MLP model does not accurately classify MNIST data samples, despite having a large gradient L1 norm, when the model is under RWF injection. This occurs because RWFs induce random changes in synaptic weights, which transiently perturb the loss and cause a temporary increase in the gradient (Foret et al., 2021). Since the MLP model has limited learning ability, it cannot effectively compensate for such perturbations.

## 4.5 PRE-ACTIVATION SENSITIVITY OF LAYERS IN MLP

Our motivation study outlines an interesting finding: in MLP, the gradients of neurons in the previous layers are more sensitive to abnormal changes in pre-activation by faults than the later layers.

Table 3 shows the gradients' L1 norm of all neurons in each layer of the MLP model. The error signal that reaches layer $\ell$ is obtained by repeatedly applying the Jacobians of all higher layers in MLP. Under faults, pre-activations drift away from the operating threshold, so the surrogate derivatives $\phi'(z - \vartheta)$ on affected layers become very small. During gradient calculation, gradients are multiplicatively contracted by a chain of small factors. Because earlier layers, such as the first

**Table 3:** The gradients' L1 norm in a 95% confidence interval upon all neurons in each layer of the MLP model under fault injection.

|  | Layer 1 | Layer 2 | Layer 3 | Layer 4 |
|---|---|---|---|---|
| MNIST (SAFs) | $(3.18 \pm 1.13) \times 10^{-6}$ | $(3.31 \pm 1.64) \times 10^{-3}$ | $(1.15 \pm 0.93) \times 10^{-3}$ | $(2.67 \pm 0.65) \times 10^{-3}$ |
| MNIST (RWFs) | $(5.61 \pm 1.46) \times 10^{-5}$ | $(5.19 \pm 1.52) \times 10^{-5}$ | $(1.37 \pm 0.58) \times 10^{-4}$ | $(1.39 \pm 0.29) \times 10^{-3}$ |
| FMNIST (SAFs) | $(3.45 \pm 0.91) \times 10^{-15}$ | $(1.01 \pm 0.37) \times 10^{-12}$ | $(1.6 \pm 0.69) \times 10^{-7}$ | $(1.02 \pm 0.36) \times 10^{-2}$ |
| FMNIST (RWFs) | $(4.91 \pm 1.55) \times 10^{-5}$ | $(5.11 \pm 1.71) \times 10^{-5}$ | $(1.46 \pm 0.62) \times 10^{-4}$ | $(1.76 \pm 0.82) \times 10^{-3}$ |

layer (smaller $\ell$), accumulate more of these factors, they suffer disproportionately severe gradient vanishing, explaining the front-loaded degradation we observe under faults.

### 4.6 SIMILARITY TO FLOW CONTROL IN COMPUTER NETWORKS

Data flow control mitigates the congestion problem in computer networks (Kurose & Ross, 2012; Wigren & Karaki, 2018). The information of input samples in SNNs is related to the data in the packets of computer networks, and the surrogate gradient corridor is related to the data capacity that a receiver can handle in computer networks. The bottleneck problem in neural networks is also similar to that of the receiver, which prevents the receiver from processing the large data packet simultaneously. The objective of flow control, which involves adjusting the size of data in a packet to satisfy the receiver's data capacity, is similar to that of enhancing fault tolerance: changing the size of information in the input samples to maintain the pre-activation within the surrogate gradient corridor.

## 5 PROPOSED MECHANISM

Our motivation study demonstrates that synaptic faults can inflate pre-activations beyond the surrogate gradient corridor, causing the gradient collapse. To prevent the gradients from vanishing in hardware-implemented SNNs of neuromorphic devices, we design an adaptive input fragmentation mechanism to avoid drift in pre-activation magnitude based on flow control in computer networks by shrinking the probability of cases where input data samples enter faulty synapses, which have abnormal weights that cause the pre-activation magnitude to increase significantly. The main idea of our paper is to enhance the fault tolerance of SNNs under synaptic faults by dividing inputs into fragments, selecting the division angle that optimally ensures the suppression of sudden pre-activation drift in SNNs. We mathematically prove why our mechanism is nearly-optimal in Appendix D.

### 5.1 OVERVIEW

The proposed mechanism consists of the sensitivity score, Gini coefficient, and fragment processing modules. Here, we briefly explain how the three modules cooperate relatively.

1. The sensitivity score module generates a sensitivity map that quantifies the extent to which each input pixel and its associated synapses influence pre-activation under fault conditions.

2. The Gini coefficient module searches over 1D projection angles on the sensitivity map, selecting the direction along which the accumulated sensitivity is most evenly distributed. This procedure defines a fair axis for fragmentation.

3. Our fragment processing module cuts the image along the fair axis into equal-sensitivity fragments, normalizes each fragment's energy via RMS normalization to keep pre-activations ($z$) inside the surrogate corridor (Zhang & Sennrich, 2019). Then, it accumulates time-step outputs with entropy-based weighting (Qiu et al., 2025).

Overall, these three modules cooperate by first identifying fault-sensitive pixels, then choosing the most balanced way to partition them, and finally enforcing the pre-activation in the surrogate corridor by RMS normalization (Zhang & Sennrich, 2019). Moreover, we ensure the accurate decoding of fragment-oriented outputs by the entropy-based approach (Qiu et al., 2025). We execute these three procedures per batch.

### 5.2 SENSITIVITY SCORE DEFINITION AND CALCULATION

**Key point 1: The sensitivity score represents which pixel changes the pre-activation the most significantly under faults.**

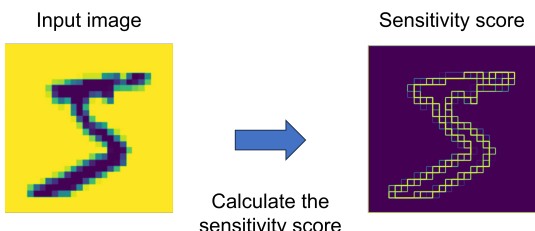

**Figure 2:** Sensitivity score calculation with an MNIST image.

Figure 2 depicts the sensitivity score of an MNIST image. To measure how input data samples are affected by synaptic faults and explore the best fragment shape that prevents a significant increase in pre-activation magnitude, we define the sensitivity score of input samples, consisting of an image sample's complexity and the influence measurement from faults.

$$I(x) = \text{PN}\Big(\big(\text{LoG}(x) + \text{Sobel}(x) + \text{Var}(x)\big) \odot \big(1 + \lambda_s\,\text{SP}(x) + \lambda_w\,\text{WP}(x)\big)\Big). \tag{1}$$

where $\text{LoG}(x)$, $\text{Sobel}(x)$, and $\text{Var}(x)$ are information about edges, blobs, and texture contrast of images. We adopt them to calculate the complexity of an image sample and design our algorithm to minimize each fragment's complexity (Lowe, 2004). This is because an image with high complexity significantly changes the pre-activation since its norm is large. $\text{SP}(x)$ is a saliency map (Petsiuk et al., 2018), and $\text{WP}(x)$ is the absolute value of the first layer's weight projection to input resolution. We decide to use the first layer's weight projection since the pre-activation drift in the first layer causes the most severe gradient vanishing. $\text{SP}(x)$ identifies the input pixels whose perturbations cause large shifts in pre-activations and, consequently, substantial changes in the final output. $\text{WP}(x)$ converts the pixel-derived map into a fault-influence field on the weights of the first layer. We apply it to our mechanism to minimize the increase in pre-activation by reducing the probability that pixel values enter many faulty synapses at once. $\odot$ is the Hadamard product. PN is a percentile normalization, which normalizes the values of the fault influence map in the range of 0 to 1. We adopt PN to prevent the sensitivity score from increasing excessively. For batch stability and fast operation, we measure the sensitivity score of the averaged image sample of a batch.

### 5.3 GINI COEFFICIENT CALCULATION WITH A 1D PROFILE

**Key point 2: The Gini coefficient indicates the equality of the sensitivity score (Farris, 2010). It should be minimized to increase the equality of the sensitivity score upon the fragments and prevent the pre-activation from falling outside of the surrogate gradient corridor.**

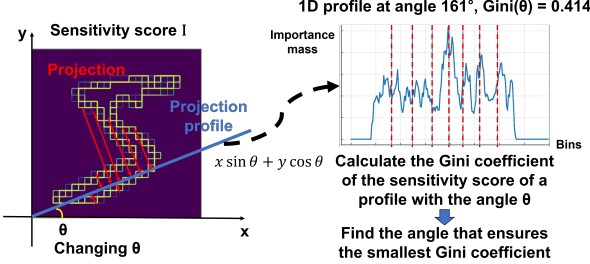

**Figure 3:** 1D projection and Gini coefficient calculation changing the projection angle.

In figure 3, we change the angle $\theta$ of the 1D sensitivity profile and project the pixels on the 1D profile. Specifically, we transform 2D coordinated pixel position $(x, y)$ into 1D bin index $s$ using the equation $s = x sin(\theta) + y cos(\theta)$. Then, we calculate the Gini coefficient with equation 2 (Farris, 2010)

$$\text{Gini}(p_\theta) \;=\; \frac{1}{2L\,\bar{p}_\theta} \sum_{i=0}^{L-1} \sum_{j=0}^{L-1} \big| p_\theta[i] - p_\theta[j] \big|. \tag{2}$$

where, $p_\theta$ is the 1D sensitivity profile bin obtained by projecting along an angle $\theta$ (range of $[0°, 360°]$). $p_\theta[i]$ is the value of bin $i$ ($i = 0, \ldots, L-1$). In other words, we obtain $p_\theta$ by projecting each pixel's sensitivity $I$ onto the index of the 1D profile $s$, and aggregating along that axis. $L$ is the number

of bins, which is related to the granularity of angle division. $\bar{p}_\theta$ is the mean value of $p_\theta$ over the number of bins ($L$). $|p_\theta[i] - p_\theta[j]|$ is absolute difference between $p_\theta[i]$ and $p_\theta[j]$. We divide the sum of pairwise absolute differences by $(2L\bar{p}_\theta)$ for normalization, so the coefficient is 0 for a uniform profile and grows with inequality.

We explore the angle $\theta$ which makes the 1D profile have the minimum Gini coefficient. This is because the Gini coefficient is a strictly Schur-convex function, ensuring the Gini coefficient strictly increases under majorization: if $x \succ y$ (i.e., $x$ is more unequal), then $\mathrm{Gini}(x) > \mathrm{Gini}(y)$, with equality only for permutations (Sandor, 2007). By minimizing the Gini coefficient, the fragment's equality of sensitivity score is maximized. We design our mechanism to maximize equality and prevent the pre-activation from leaving the corridor. This is because as a fragment's sensitivity score becomes more equal by minimizing Gini coefficient, the maximum of each fragment's energy $\|x_t\|_2$ decreases, so the upper bounds $|w^\top x_t| \le \|w\|_2\|x_t\|_2$ and $|u_t| = |w^\top x_t + m|$, $(m = b - \vartheta)$ are pushed below $\delta$, keeping the pre-activation fixed inside the corridor ($b$ is a bias in SNNs' layers.).

## 5.4 Fragment generation based on equal sensitivity score and in/out for SNNs

**Key point 3: The fragmentation line is set by the 1D profile cutting to make the 1D bins have an equal cumulative sum of the sensitivity score.**

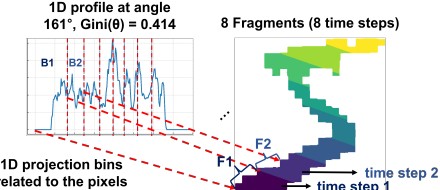

**Figure 4:** Dividing an image sample based on the cumulative sum of the sensitivity score.

After finding the angle that ensures the minimum Gini coefficient, we calculate the cumulative sum of the sensitivity score. As depicted in figure 4, the division line for fragmentation (dashed line in the figure) indicates the position to cut the 1D profile into bins $B_1, \ldots, B_T$, making the bins have equal sensitivity score. We generate fragments $F_1, \ldots, F_T \in \{0, 1\}^{H \times W}$ with the pixels correlated to the points in bins $B_1, \ldots, B_T$ and feed the SNN models the fragments $F_1, \ldots, F_T$ over time steps $t = 1..T$. We zero-pad the generated fragment to align the input dimension of fragments with the input dimension of the original samples because the input dimension of SNNs is not changeable during training and testing. We apply the same division angle to all data samples in the same batch.

To ensure that pre-activation $z_t$ of fragment $F_t$ positions in the corridor, keeping the scale of active pixels in each fragment to a target Root-Mean-Square (RMS) is also important. Therefore, we adopt RMS normalization to the input fragments with the equation 3 (Zhang & Sennrich, 2019)

$$\tilde{x}_t = g_t x_t, \quad \|\tilde{x}_t\|_2 = \alpha \;\Rightarrow\; |z_t| \le \|\hat{w}\|_2\, \alpha. \tag{3}$$

where $x_t$ is an input of time step $t$ and $g_t$ is the per-fragment normalization gain. $\alpha$ is the L2 norm of the $\tilde{x}_t$ (the multiplication of $x_t$ and $g_t$) and denotes the non-zero pixels in the fragment for the input of time step $t$. We set $g_t$ to ensure that $\|\hat{w}\|_2\alpha$ is always smaller than the bound of the surrogate corridor, placing $z_t$ inside the surrogate-derivative corridor.

We also adopt an entropy-based output decoding technique to aggregate the outputs (logits) from SNNs across all time steps accurately with the equation 4 (Qiu et al., 2025)

$$\bar{\ell} = \sum_{t=1}^{T} e_t\, \ell_t, \qquad e_t \;=\; \frac{\exp\big(-\tau H(\mathrm{softmax}(\ell_t))\big)}{\sum_{s=1}^{T}\exp\big(-\tau H(\mathrm{softmax}(\ell_s))\big)}. \tag{4}$$

where $\bar{\ell}$ is the final output vector from the entropy-weighted aggregation. $\ell_t$ is an output vector from SNNs at time $t$ and $T$ is the total number of steps. $H(\mathrm{softmax}(\ell_t))$ is the Shannon entropy of the output vector's softmax result, and $\tau$ controls how strongly low-entropy (confident) steps are emphasized. $e_t$ is a scaling factor to reflect the entropy of output vectors while decoding. $\exp$ is an exponential function, and $s$ is the start index of time steps.

# 6 EXPERIMENTS

## 6.1 EXPERIMENTAL SETTINGS

We conduct various experiments with MLP (LIF neurons), VGG-7/11/15 (LIF neurons), and ResNet-18/34 (IF neurons) SNN models based on SpikingJelly, widely used for SNN implementation, by classifying samples in MNIST/FMNIST/UCI-HAR/AudioMNIST (MLP), CIFAR-10/100 (VGG and ResNet), and Tiny-ImageNet (ResNet) (Fang et al., 2023; LeCun et al., 1998; Xiao et al., 2017; Krizhevsky, 2009; Reyes-Ortiz et al., 2013; Becker et al., 2024; Deng et al., 2015). We select these SNN models and datasets because current SNN technologies do not adequately train large and deep models on complex datasets (Fang et al., 2023; Schuman et al., 2022). We measure the classification accuracy of the SNN models using our mechanism and the benchmarks under SAFs (injected during training), setting the ratio of SA1 and SA0 to $SA0 : SA1 = 1.75 : 9.04$ (Chen et al., 2017) and the weight boundary to $[-1, 1]$ (Le Gallo et al., 2023; Lammie et al., 2022). Additionally, we inject RWFs and CEFs into the synapses of these models. We use ECOC (Liu et al., 2019), SoftSNN (Putra et al., 2022), Routing (Yang et al., 2022), Astrocyte (Han et al., 2023), FalVolt (Siddique & Hoque, 2023), and LIFA (Yunusoglu et al., 2025) for our benchmarks[1]. The SNN models without any fault-mitigation mechanism serve as the baseline. We use RMSE as a loss function and Adam as the optimizer for SNN models (Fang et al., 2023). The batch size is 100, and the learning rate is 0.001 for MLP/VGG and 0.01 for ResNet. We set the number of time steps (fragments) to 2, 4, and 8. We use 50 epochs for training. We set $\lambda_s$, $\lambda_w$ to 0.1, and $\tau$ to 2.0 by tuning these parameters through experimental repetition with the proposed mechanism. We repeat all experiments 10 times with different random seeds and present the experimental results in a 95% confidence interval. We inject faults into the synapses sporadically in a uniform distribution, resulting in the uniform position of synaptic faults. Note that the additional results from the additional datasets (UCI-HAR, AudioMNIST, and Tiny-ImageNet), different time steps (4 and 8 steps), other fault types (RWFs and CEFs), the ablation study with the combination of our mechanism, various hyperparameter ($\lambda_s$ and $\lambda_w$) settings, and the results with DNN models are presented in Appendix A. Additionally, we demonstrate that our mechanism successfully improves the fault tolerance of hardware-implemented SNNs through evaluations with an actual FPGA device.

## 6.2 CLASSIFICATION ACCURACY COMPARISON

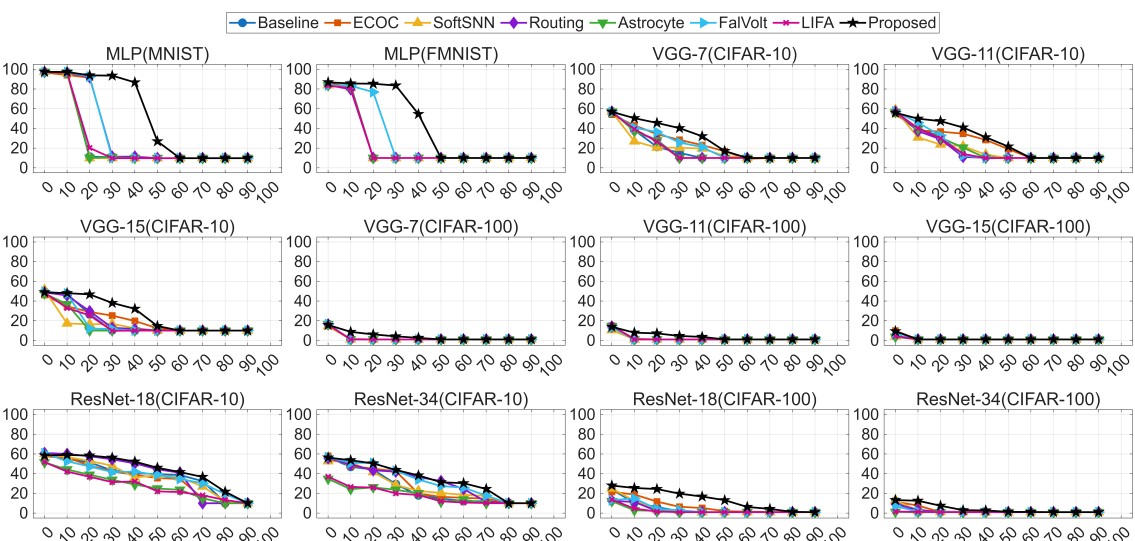

**Figure 5:** Average classification accuracy of various SNN models with the baseline, benchmarks, and proposed mechanism under SAFs using 2 time steps. The x-axis is the fault ratio (%) and the y-axis is the accuracy (%).

Figure 5 illustrates the classification accuracy of SNN models compared to the baseline, benchmarks, and proposed mechanism under SAFs with 2 time steps. Under SAFs, the SNN models with our mechanism exhibit the best classification accuracy across all datasets and models in most cases.

---

[1]We briefly explain how these benchmarks enhance the fault tolerance in Section 3.

### 6.2.1 MLP MODELS

In the MLP model, the classification accuracy drops dramatically as the fault ratio increases. This is because the faulty weights are directly multiplied by the input values, and the pre-activation magnitude increases significantly, allowing it to easily escape from the surrogate gradient corridor. Our mechanism definitely outperforms the baseline and benchmarks, since it utilizes the input saliency and weight projection map of the first layer in Gini-based equal fragmentation. The MLP model is vulnerable to faults in the first layer, as mentioned in Subsection 4.1. By adopting the saliency and weight projection map (fault influence map) of the first layer, we suppress its pre-activation, which decides the surrogate gradient, from increasing significantly and escaping from the corridor. Thus, the pre-activation does not lie far from the corridor, preserving the power of the first layer's gradient.

### 6.2.2 VGG MODELS

While VGG models with the benchmarks using CIFAR-10 maintain proper classification performance only up to a fault ratio of 30–40%, the models with the proposed mechanism sustain correct classification even at fault ratios as high as 50%. This is because our mechanism targets to make the pre-activation lie at the point in the surrogate gradient corridor with Gini-based equal mass fragmentation, despite the large amount of faults. Contrarily, the benchmarks do not consider the relations between pre-activation, corridor, and surrogate gradient, failing to bound the pre-activation in the corridor. Therefore, the classification accuracy of the model using the benchmarks degrades sharply with fewer faults. We also observe that the classification accuracy declines as the model gets deeper. This occurs because surrogate gradients in deep SNNs cannot reliably approximate the hypothetical gradients of LIF neurons (Guo et al., 2024). When we use CIFAR-100, the models do not classify the data samples accurately due to their low learning ability. However, the models with our mechanism exhibit the highest classification accuracy under SAFs in most cases.

### 6.2.3 RESNET MODELS

Different from VGG models, ResNet models integrated with the benchmarks and proposed mechanism maintain the classification accuracy up to a fault ratio of 80-90% when we use CIFAR-10. This is because ResNet models have internal mechanisms to compensate for errors in gradient calculations, such as residual blocks. They also classify CIFAR-100 samples more accurately than VGG models under faults, since they have a more powerful learning ability than VGG models. ResNet-18 using CIFAR-100 maintains its classification ability up to the ratio of 60-70% only with our mechanism, and the ResNet-34 with CIFAR-100 maintains the classification ability up to the ratio of 30-40% only with our mechanism. These results demonstrate that our mechanism successfully enhances the fault tolerance with complicated datasets and models. We observe that the astrocyte-based approaches (Astrocyte and LIFA) do not improve the deep ResNet models' fault tolerance at all. This problem derives from the fact that they only mimic biological mechanisms of neuronal activity in brains, which enhances the fault tolerance of shallow and highly bio-plausible models such as Diehl & Cook 2015, using a bio-plausible unsupervised learning rule (Han et al., 2023; Yunusoglu et al., 2025). However, our mechanism successfully strengthens the models' fault tolerance in most cases since we tackle a fundamental problem of faulty SNN models regardless of the types of SNN models, and develop a solution to mitigate the problem.

## 7 CONCLUSION

This paper introduces a simple yet effective fault mitigation mechanism for SNNs that does not require complicated architectures or direct weight modifications based on input data control. Our mechanism improves fault tolerance more effectively than conventional approaches in various SNN models and datasets. Experimental results exhibited improvement in the fault tolerance of our mechanism over benchmarks in various network models and datasets, including real hardware environments. We emphasize that this improvement is primarily achieved through an effective input data control mechanism based on detailed observation of how synaptic faults ruin the learning capability of SNNs. Our mechanism allows SNNs to maintain reliable operation and high fault tolerance in a practical and hardware-compatible manner, enabling more sustainable and reliable edge AI computing.

## ETHICS STATEMENT

This work does not involve human subjects, sensitive personal data, or potentially harmful applications. We trained and validated our models using publicly available datasets (e.g., MNIST, FMNIST, CIFAR-10, and CIFAR-100), without any private or identifiable information. We design our mechanism to enhance the robustness of neuromorphic systems against hardware faults. We declare no conflict of interest or external sponsorship that might have influenced the research outcomes.

## REPRODUCIBILITY STATEMENT

We conducted all experiments on publicly available datasets with standard train/validation/test splits. To facilitate replication, we provide our full implementation of the proposed mechanisms in the anonymous supplementary material. We also present the hyperparameter settings, model configurations, and hardware specifications to support reproducibility. We will publicly release the code and scripts on GitHub if our paper is accepted for the conference.

## LARGE LANGUAGE MODEL USAGE STATEMENT

We used Large Language Models (LLMs) as writing and experiment assistants to improve the clarity of writing/editing mathematical equations (fixing typos/suggesting algebraic simplifications), and assist us in conducting experiments with the benchmarks. We did not use LLMs for idea generation, methodological design, analysis, or to originate any mathematical arguments or claims. We derived, verified, and finalized all derivations, results, and claims.

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

# A DISCUSSION

In this section, we discuss the pros and cons of the proposed mechanism and the lightweight approaches mentioned in Subsection 3.2 (Saha et al., 2023; Spyrou et al., 2021; Saha et al., 2024). Furthermore, we demonstrate that our mechanism exhibits stronger fault mitigation ability than the lightweight approaches, consuming less overhead than the other approaches. Existing lightweight mechanisms for hardware SNN fault mitigation rely on continuously monitoring internal neuron currents or spike statistics and mitigating the adversarial effects of hardware defects (Spyrou et al., 2021; Saha et al., 2023). These designs are acceptable and lightweight for neuromorphic devices that frequently perform inferences due to their simple operations. However, their computational overhead significantly increases in large SNN models. This is because scanning all weights or neuron states and aggregating their statistics grows exponentially as the model's size increases. Furthermore, they inevitably discard fault-affected pre-activation values, which causes information loss. In contrast, our mechanism avoids direct inspection of entire elements in SNN models or modification of synaptic weights or circuits in hardware. Instead, it computes a sensitivity score from the input samples and a single reference layer for fragmentation. Therefore, the external controller only needs to exchange a small amount of metadata with the neuromorphic device instead of full weight or neuron maps (Khan et al., 2024). When it comes to test-based schemes, they amortize their cost by running only intermittently (Spyrou et al., 2021; Saha et al., 2024). However, they cannot react to permanent faults that arise during deployment. On the other hand, our mechanism always executes during training and inference, incurs modest per-execution overhead, and can immediately adapt the fragmentation strategy to the current fault state in agile environments. Overall, the proposed mechanism provides the following complementary points in the design space. First, it scales better than forward-pass approaches with scanning to large hardware-implemented SNNs, which are important for modern neuromorphic devices (Yin et al., 2024). Second, it trades higher continuous overhead than periodic self-test approaches, achieving substantially stronger mitigation of permanent faults. We compare the accuracy and inference time of the proposed mechanism to that of other representative lightweight mechanisms: input suppression, fault hopping, and threshold tuning (Saha et al., 2023; Spyrou et al., 2021; Saha et al., 2024).

**Table 4:** The SNN model's classification accuracy and summation of inference time over 100 repetitions in a 95% confidence interval with other lightweight approaches and our mechanism under the 30% fault ratio of SAFs.

| DATASETS (MODELS) | INPUT SUPPRESSION | FAULT HOPPING | THRESHOLD TUNING | PROPOSED |
|---|---|---|---|---|
| ACCURACY (%) IN SOFTWARE-BASED SNN MODELS | | | | |
| MNIST (MLP) | $88.38 \pm 1.15$ | $86.41 \pm 0.98$ | $87.23 \pm 1.52$ | $\mathbf{93.79 \pm 1.06}$ |
| FMNIST (MLP) | $80.45 \pm 1.22$ | $79.86 \pm 1.16$ | $78.98 \pm 1.39$ | $\mathbf{85.47 \pm 0.92}$ |
| CIFAR-10 (VGG-7) | $34.67 \pm 3.21$ | $30.28 \pm 3.34$ | $27.91 \pm 3.26$ | $\mathbf{45.94 \pm 3.17}$ |
| ACCURACY (%) IN FPGA-IMPLEMENTED SNN MODELS | | | | |
| MNIST (MLP) | $86.7 \pm 1.32$ | $84.28 \pm 1.28$ | $85.87 \pm 1.65$ | $\mathbf{91.43 \pm 1.17}$ |
| FMNIST (MLP) | $78.46 \pm 0.98$ | $75.39 \pm 1.14$ | $73.63 \pm 1.27$ | $\mathbf{84.05 \pm 1.09}$ |
| CIFAR-10 (VGG-7) | $31.94 \pm 3.8$ | $27.14 \pm 3.72$ | $24.75 \pm 2.87$ | $\mathbf{38.11 \pm 3.58}$ |
| INFERENCE TIME (SEC) IN FPGA-IMPLEMENTED SNN MODELS | | | | |
| MNIST (MLP) | $283.64 \pm 4.85$ | $254.43 \pm 2.1$ | $213.67 \pm 2.38$ | $\mathbf{191.28 \pm 1.25}$ |
| FMNIST (MLP) | $285.27 \pm 5.17$ | $255.79 \pm 1.98$ | $215.09 \pm 2.51$ | $\mathbf{193.54 \pm 1.62}$ |
| CIFAR-10 (VGG-7) | $500.59 \pm 7.34$ | $327.11 \pm 3.83$ | $292.41 \pm 4.86$ | $\mathbf{277.09 \pm 3.47}$ |
| INFERENCE TIME (SEC) IN FPGA-IMPLEMENTED SNN MODELS | | | | |
| MNIST (MLP) | $80.26 \pm 1.06$ | $62.35 \pm 0.93$ | $50.39 \pm 0.61$ | $\mathbf{45.26 \pm 0.73}$ |
| FMNIST (MLP) | $81.38 \pm 1.19$ | $64.84 \pm 1.25$ | $51.08 \pm 0.67$ | $\mathbf{46.38 \pm 0.76}$ |
| CIFAR-10 (VGG-7) | $124.22 \pm 2.42$ | $101.57 \pm 2.61$ | $78.53 \pm 1.3$ | $\mathbf{68.91 \pm 1.05}$ |

Table 4 presents the accuracy and summation of inference time over 100 iterations (assuming the actual scenarios) for the SNN models using the lightweight approaches and the proposed mechanism under 20% SAFs. Our mechanism exhibits more effective fault mitigation ability than other lightweight approaches, consuming less or comparable inference time to these approaches.

## B ADDITIONAL EXPERIMENTAL RESULTS ON CLASSIFICATION ACCURACY

We present additional experimental results that support the proposed mechanism in this section. The extra results demonstrate that our mechanism enhances the neural networks' fault tolerance more effectively than the previous methods[2] in various environments and scenarios, including real hardware. Furthermore, we discuss the changes in the proposed mechanism's fault mitigation ability by adopting various settings to our mechanism.

### B.1 ADDITIONAL DATASETS BEYOND MNIST, FMNIST, CIFAR-10, AND CIFAR-100

We use UCI-HAR, AudioMNIST, and Tiny-ImageNet to evaluate the fault mitigation ability of our mechanism on a sequential and large-scale dataset.

#### B.1.1 SEQUENTIAL DATASET

To demonstrate that our mechanism works well with the models using sequential datasets, we conduct experiments with UCI-HAR and AudioMNIST, which comprises six types of human activities collected by electric sensors and a dataset consisting of verbal sounds of digits (Reyes-Ortiz et al., 2013; Becker et al., 2024).

**Table 5:** The MLP model's classification accuracy in a 95% confidence interval using MNIST, FMNIST, UCI-HAR, and AudioMNIST with 2 time steps under SAFs.

| Fault ratio(%) | Baseline | ECOC | SoftSNN | Routing | Astrocyte | FalVolt | LIFA | Proposed |
|---|---|---|---|---|---|---|---|---|
| | | | | Accuracy (%) with MLP (UCI-HAR) | | | | |
| 0 | $64.14 \pm 4.91$ | $\mathbf{69.86 \pm 4.73}$ | $63.41 \pm 4.78$ | $65.17 \pm 4.65$ | $64.78 \pm 4.39$ | $65.76 \pm 4.55$ | $64.69 \pm 4.81$ | $63.14 \pm 4.8$ |
| 10 | $20.19 \pm 2.95$ | $23.01 \pm 4.28$ | $22.21 \pm 4.63$ | $23.1 \pm 4.58$ | $21.25 \pm 3.56$ | $23.24 \pm 4.01$ | $16.93 \pm 4.52$ | $\mathbf{50.31 \pm 4.68}$ |
| 20 | $19.66 \pm 2.54$ | $17.1 \pm 0$ | $20.52 \pm 3.85$ | $17.1 \pm 0$ | $20.59 \pm 2.81$ | $22.34 \pm 3.68$ | $17.1 \pm 0$ | $\mathbf{49.24 \pm 4.59}$ |
| 30 | $17.1 \pm 0$ | $17.1 \pm 0$ | $19.9 \pm 2.26$ | $17.1 \pm 0$ | $17.1 \pm 0$ | $19.85 \pm 2.92$ | $17.1 \pm 0$ | $\mathbf{47.31 \pm 4.36}$ |
| 40 | $17.1 \pm 0$ | $17.1 \pm 0$ | $17.1 \pm 0$ | $17.1 \pm 0$ | $17.1 \pm 0$ | $16.93 \pm 0$ | $17.1 \pm 0$ | $\mathbf{17.21 \pm 0.09}$ |
| 50 | $17.1 \pm 0$ | $17.1 \pm 0$ | $17.1 \pm 0$ | $17.1 \pm 0$ | $17.1 \pm 0$ | $16.93 \pm 0$ | $17.1 \pm 0$ | $17.1 \pm 0$ |
| 60 | $17.1 \pm 0$ | $17.1 \pm 0$ | $17.1 \pm 0$ | $17.1 \pm 0$ | $17.1 \pm 0$ | $16.93 \pm 0$ | $17.1 \pm 0$ | $17.1 \pm 0$ |
| 70 | $17.1 \pm 0$ | $17.1 \pm 0$ | $17.1 \pm 0$ | $17.1 \pm 0$ | $17.1 \pm 0$ | $16.93 \pm 0$ | $17.1 \pm 0$ | $17.1 \pm 0$ |
| 80 | $17.1 \pm 0$ | $17.1 \pm 0$ | $17.1 \pm 0$ | $17.1 \pm 0$ | $17.1 \pm 0$ | $16.93 \pm 0$ | $17.1 \pm 0$ | $17.1 \pm 0$ |
| 90 | $17.1 \pm 0$ | $17.1 \pm 0$ | $17.1 \pm 0$ | $17.1 \pm 0$ | $17.1 \pm 0$ | $16.93 \pm 0$ | $17.1 \pm 0$ | $17.1 \pm 0$ |
| | | | | Accuracy (%) with MLP (AudioMNIST) | | | | |
| 0 | $96.45 \pm 0.91$ | $\mathbf{96.88 \pm 0.97}$ | $95.89 \pm 0.92$ | $96.21 \pm 0.85$ | $96.29 \pm 0.93$ | $96.34 \pm 0.89$ | $96.09 \pm 0.93$ | $96.31 \pm 0.87$ |
| 10 | $94.47 \pm 1.07$ | $94.56 \pm 1.79$ | $92.56 \pm 1.61$ | $94.03 \pm 1.84$ | $93.08 \pm 1.92$ | $94.47 \pm 1.58$ | $93.28 \pm 2.18$ | $\mathbf{94.98 \pm 1.54}$ |
| 20 | $93.17 \pm 1.56$ | $93.29 \pm 1.98$ | $88.15 \pm 2.84$ | $92.59 \pm 2.02$ | $85.24 \pm 3.68$ | $93.29 \pm 2.13$ | $88.57 \pm 4.26$ | $\mathbf{93.78 \pm 1.98}$ |
| 30 | $92.89 \pm 2.05$ | $93.33 \pm 2.32$ | $75.83 \pm 3.51$ | $91.7 \pm 2.75$ | $51.47 \pm 5.56$ | $92.76 \pm 2.47$ | $55.81 \pm 6.01$ | $\mathbf{93.43 \pm 3.09}$ |
| 40 | $89.5 \pm 3.21$ | $92.51 \pm 2.15$ | $62.49 \pm 5.07$ | $91.24 \pm 3.18$ | $37.55 \pm 6.27$ | $90.63 \pm 2.99$ | $39.64 \pm 6.8$ | $\mathbf{92.65 \pm 3.17}$ |
| 50 | $86.15 \pm 4.18$ | $85.46 \pm 3.83$ | $48.61 \pm 6.92$ | $86.45 \pm 3.97$ | $22.13 \pm 4.49$ | $87.51 \pm 4.24$ | $24.48 \pm 5.07$ | $\mathbf{88.93 \pm 4.16}$ |
| 60 | $52.87 \pm 5.71$ | $66.42 \pm 5.2$ | $34.62 \pm 5.33$ | $64.92 \pm 5.34$ | $10.96 \pm 0.96$ | $65.97 \pm 5.93$ | $12.74 \pm 2.38$ | $\mathbf{67.83 \pm 5.77}$ |
| 70 | $14.6 \pm 1.23$ | $21.64 \pm 2.15$ | $14.58 \pm 1.4$ | $25.84 \pm 3.42$ | $10 \pm 0$ | $28.18 \pm 3.85$ | $10 \pm 0$ | $\mathbf{37.17 \pm 3.01}$ |
| 80 | $11.77 \pm 0.95$ | $13.38 \pm 0.94$ | $9.83 \pm 1.05$ | $14.19 \pm 1.36$ | $10 \pm 0$ | $15.53 \pm 2.36$ | $10 \pm 0$ | $\mathbf{19.75 \pm 2.89}$ |
| 90 | $11.21 \pm 1.01$ | $11.86 \pm 0.87$ | $9.46 \pm 1.22$ | $11.68 \pm 0.96$ | $10 \pm 0$ | $10 \pm 0$ | $10 \pm 0$ | $\mathbf{12.83 \pm 1.02}$ |

Table 5 shows the classification accuracy of the MLP model using UCI-HAR. Using the MLP model with a sequential dataset, the model with the proposed mechanism exhibits better fault tolerance than the baseline and benchmarks, classifying data samples more accurately than the models with the baseline and benchmarks. As results with UCI-HAR, the model with the proposed mechanism achieves the highest classification accuracy under SAFs. These experimental results with the two sequential datasets demonstrate that our mechanism successfully improves fault tolerance of SNNs on other domains, such as sensor-obtained and audio data samples.

---

[2]The benchmarks mentioned in Section 6.

### B.1.2 LARGE IMAGE DATASET

We use the ResNet-34 model to classify data samples in Tiny-ImageNet, which is a small version of the ImageNet dataset, consisting of $64 \times 64$ pixel images with 200 classes. We measure the classification accuracy of the model with Tiny-ImageNet under SAFs.

**Table 6:** The ResNet-34 model's classification accuracy in a 95% confidence interval using Tiny-ImageNet under SAFs with 2 time steps.

| FAULT RATIO(%) | BASELINE | ECOC | SOFTSNN | ROUTING | ASTROCYTE | FALVOLT | LIFA | PROPOSED |
|---|---|---|---|---|---|---|---|---|
| 0 | $3.38 \pm 2.11$ | $3.61 \pm 2.24$ | $3.27 \pm 2.08$ | $3.24 \pm 2.16$ | $0.65 \pm 0.15$ | $3.47 \pm 1.98$ | $0.59 \pm 0.09$ | $\mathbf{3.96 \pm 2.39}$ |
| 10 | $0.65 \pm 0.12$ | $0.69 \pm 0.15$ | $0.68 \pm 0.13$ | $0.71 \pm 0.19$ | $0.5 \pm 0$ | $0.72 \pm 0.16$ | $0.5 \pm 0$ | $\mathbf{1.17 \pm 0.28}$ |
| 20 | $0.5 \pm 0$ | $0.5 \pm 0$ | $0.5 \pm 0$ | $0.5 \pm 0$ | $0.5 \pm 0$ | $0.5 \pm 0$ | $0.5 \pm 0$ | $\mathbf{0.63 \pm 0.12}$ |
| 30 | $0.5 \pm 0$ | $0.5 \pm 0$ | $0.5 \pm 0$ | $0.5 \pm 0$ | $0.5 \pm 0$ | $0.5 \pm 0$ | $0.5 \pm 0$ | $\mathbf{0.56 \pm 0.06}$ |
| 40 | $0.5 \pm 0$ | $0.5 \pm 0$ | $0.5 \pm 0$ | $0.5 \pm 0$ | $0.5 \pm 0$ | $0.5 \pm 0$ | $0.5 \pm 0$ | $0.5 \pm 0$ |
| 50 | $0.5 \pm 0$ | $0.5 \pm 0$ | $0.5 \pm 0$ | $0.5 \pm 0$ | $0.5 \pm 0$ | $0.5 \pm 0$ | $0.5 \pm 0$ | $0.5 \pm 0$ |
| 60 | $0.5 \pm 0$ | $0.5 \pm 0$ | $0.5 \pm 0$ | $0.5 \pm 0$ | $0.5 \pm 0$ | $0.5 \pm 0$ | $0.5 \pm 0$ | $0.5 \pm 0$ |
| 70 | $0.5 \pm 0$ | $0.5 \pm 0$ | $0.5 \pm 0$ | $0.5 \pm 0$ | $0.5 \pm 0$ | $0.5 \pm 0$ | $0.5 \pm 0$ | $0.5 \pm 0$ |
| 80 | $0.5 \pm 0$ | $0.5 \pm 0$ | $0.5 \pm 0$ | $0.5 \pm 0$ | $0.5 \pm 0$ | $0.5 \pm 0$ | $0.5 \pm 0$ | $0.5 \pm 0$ |
| 90 | $0.5 \pm 0$ | $0.5 \pm 0$ | $0.5 \pm 0$ | $0.5 \pm 0$ | $0.5 \pm 0$ | $0.5 \pm 0$ | $0.5 \pm 0$ | $0.5 \pm 0$ |

Table 6 presents the average classification accuracy of the ResNet-34 model with the baseline, benchmarks, and proposed mechanism using Tiny-ImageNet. The classification accuracy of the model degrades because the dataset is complex, and SNN models have lower learning capabilities compared to DNN models. Despite the low classification accuracy, the model with our mechanism classifies data samples in Tiny-ImageNet with the highest accuracy. Moreover, the model with our mechanism exhibits higher accuracy than others in the clean scenario (without SAFs). This is because our mechanism leads the models to emit the output precisely through entropy-based decoding.

## B.2 Changing the number of time steps

We change the number of time steps to 4 and 8, observing the accuracy trend of all SNN models in the number of time steps. We obtain the experiment results with 4 and 8 time steps by repeating the experiments 10 times.

### B.2.1 4 time steps

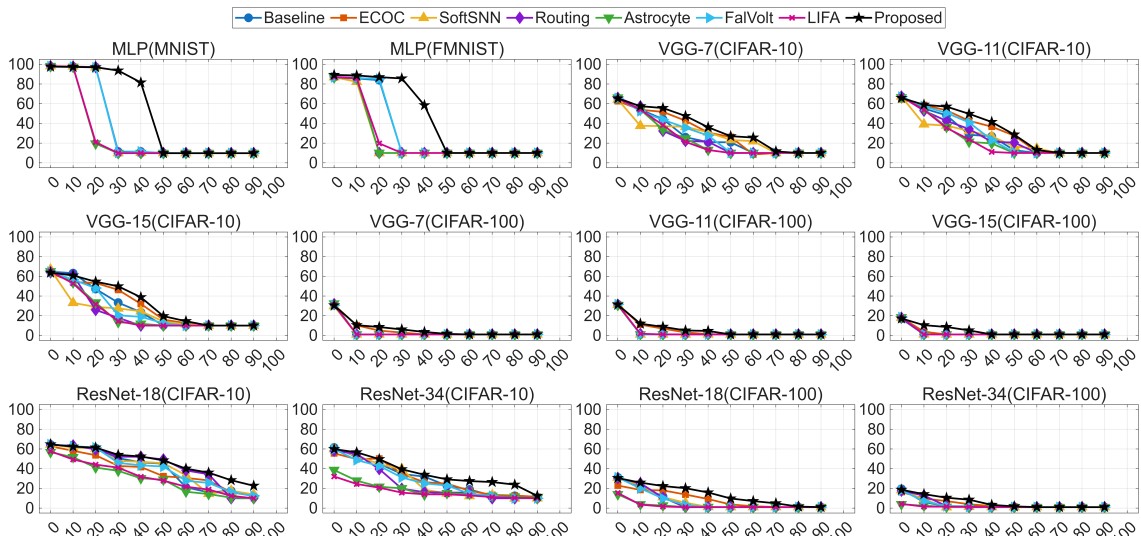

**Figure 6:** Average classification accuracy of various SNN models with the baseline, benchmarks, and proposed mechanism under SAFs using 4 time steps. The x-axis is the fault ratio (%) and the y-axis is the accuracy (%).

Figure 6 illustrates the classification accuracy of SNN models compared to the baseline, benchmarks, and proposed mechanism under SAFs with 4 time steps. The models with our mechanism exhibit the best fault tolerance to faults in most cases, like the experimental results with 2 time steps. We observe that the classification accuracy of all models overall improves as the number of time steps increases, because the large number of time steps improves the performance of SNNs (Li et al., 2024b).

## B.2.2 8 TIME STEPS

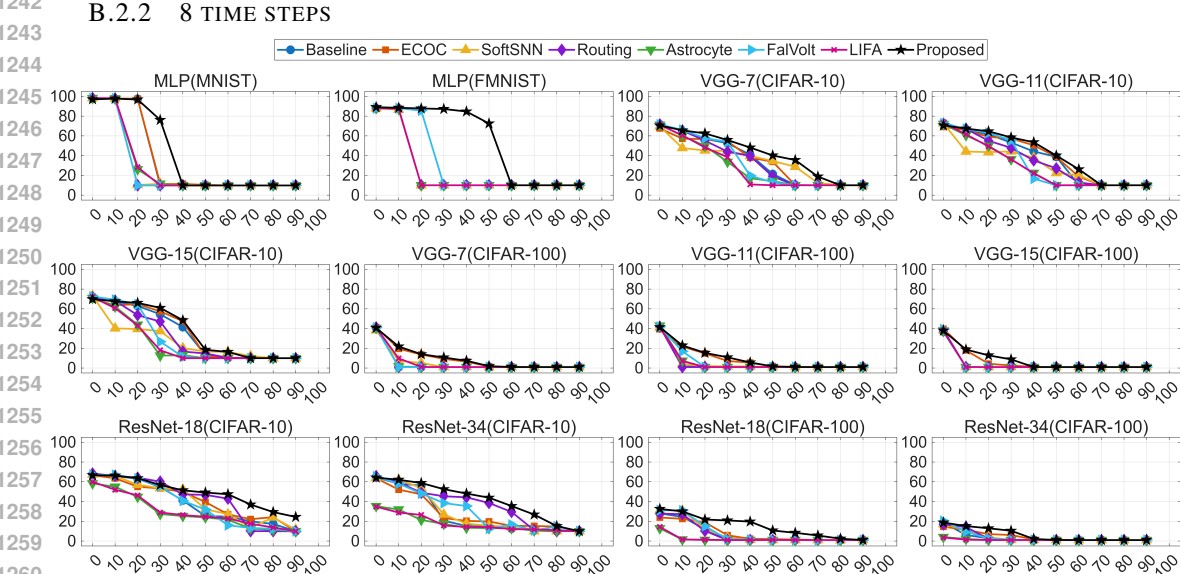

**Figure 7:** Average classification accuracy of various SNN models with the baseline, benchmarks, and proposed mechanism under SAFs using 8 time steps. The x-axis is the fault ratio (%) and the y-axis is the accuracy (%).

Figure 7 depicts the classification accuracy of SNN models compared to the baseline, benchmarks, and proposed mechanism under SAFs with 8 time steps. As demonstrated in the experimental results with 2 and 4 time steps, the models with our mechanism classify data samples most accurately. The models' accuracy is also higher than when using 2 and 4 time steps. Interestingly, the fault mitigation ability of our mechanism degrades only in the experiment with MNIST samples. This is because the MNIST samples contain fewer pixels than FMNIST, CIFAR-10, and CIFAR-100. However, they are divided into too small fragments, and these fragments do not have sufficient information for the MLP model to learn. Thus, the fault tolerance of the model with our mechanism weakens, although it is more fault-robust than models with the benchmarks.

## B.3 UNDER THE DIFFERENT TYPES OF SYNAPTIC FAULTS

We inject RWFs and CEFs into the synapses of the SNN models and measure the fault mitigation ability of the benchmarks and the proposed mechanism. The models with our mechanism classify data samples most accurately under RWFs and CEFs.

### B.3.1 RWFs

We use a Gaussian distribution to model RWFs, setting the standard deviation of the distribution to 0.5 (Garaffa et al., 2021; Spyrou et al., 2021; Vatajelu et al., 2019). We

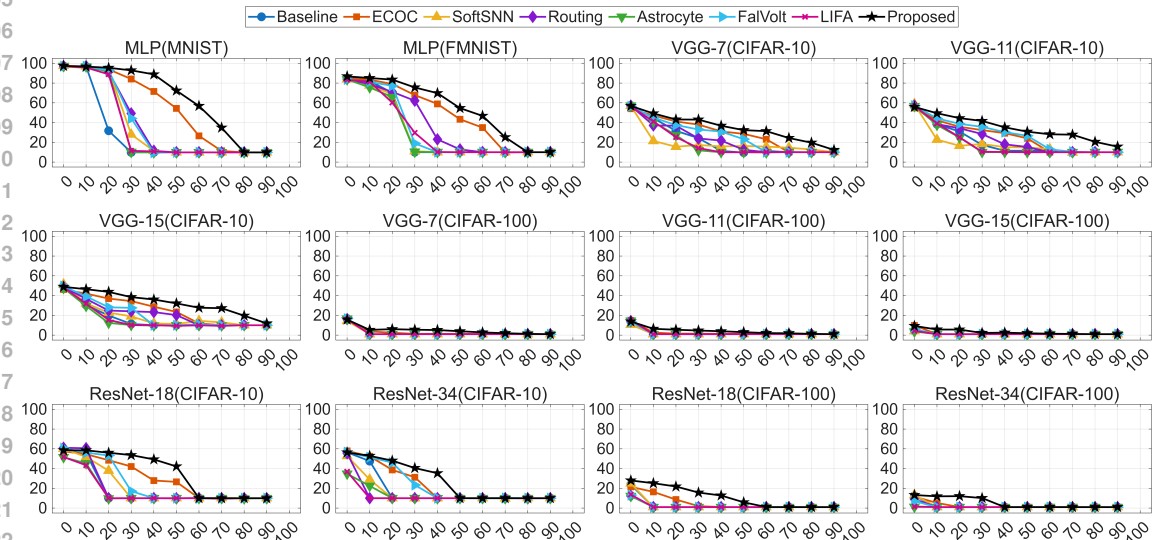

**Figure 8:** Average classification accuracy of various SNN models with the baseline, benchmarks, and proposed mechanism under RWFs using 2 time steps. The x-axis is the fault ratio (%) and the y-axis is the accuracy (%).

Figure 8 depicts the classification accuracy of SNN models with the baseline, benchmarks, and proposed mechanism under RWFs. The models with our mechanism exhibit the highest accuracy in classifying MNIST, FMNIST, CIFAR-10, and CIFAR-100. This is because our mechanism successfully prevents the pre-activation magnitude from increasing excessively by RWFs. Interestingly, ECOC presents high fault mitigation ability under RWFs. This is because ECOC uses error correcting codes, which are robust against Gaussian noise in channels to compensate for errors caused by faults in the last layer (Liu et al., 2019).

### B.3.2 CEFs

CEFs change the connections between spiking neurons randomly, ruining the learned information of SNN models (Vatajelu et al., 2019).

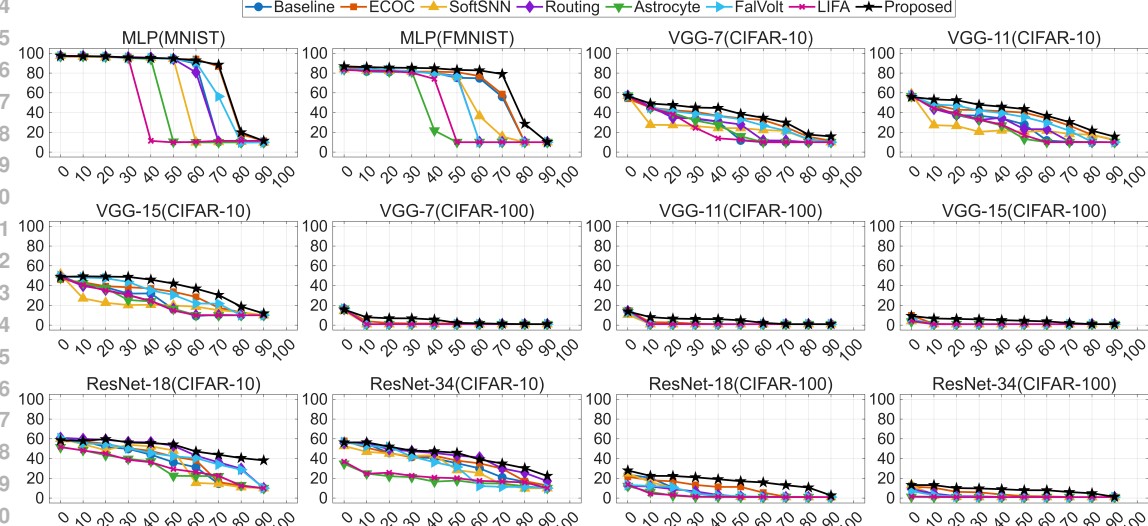

**Figure 9:** Average classification accuracy of various SNN models with the baseline, benchmarks, and proposed mechanism under CEFs using 2 time steps. The x-axis is the fault ratio (%) and the y-axis is the accuracy (%).

Figure 9 illustrates the classification accuracy of SNN models with the baseline, benchmarks, and proposed mechanism under CEFs. Our mechanism also presents the best fault mitigation ability. The classification accuracy of SNN models under CEFs is higher than that of the models under SAFs and RWFs. This is because the weights of faulty synapses are uniform under CEFs, and the pre-activation magnitude does not increase significantly. Thus, the pre-activation does not lie in a value that is far from the surrogate gradient corridor.

### B.4 ABLATION STUDY ON THE COMBINATION OF OUR MECHANISM

We conduct ablation studies by changing the settings of our mechanism (horizontally-fixed vs Gini-based and only complexity-based sensitivity score vs complexity and influence combined sensitivity score).

### B.4.1 MLP MODEL (MNIST, FMNIST)

**Table 7:** The MLP model's classification accuracy in a 95% confidence interval with different settings of our mechanism under SAFs.

| FAULT RATIO(%) | BASELINE | HORIZONTAL | GINI(COMPLEXITY ONLY) | GINI(PROPOSED) |
|---|---|---|---|---|
| ACCURACY (%) WITH MLP (MNIST) | | | | |
| 0 | $97.35 \pm 0.46$ | $86.68 \pm 3.51$ | $97.37 \pm 0.44$ | $\mathbf{97.44 \pm 0.39}$ |
| 10 | $96.94 \pm 1.01$ | $75.77 \pm 4.96$ | $96.76 \pm 0.95$ | $\mathbf{97.25 \pm 0.98}$ |
| 20 | $11.35 \pm 0$ | $71.13 \pm 4.81$ | $93.64 \pm 1.44$ | $\mathbf{93.84 \pm 1.37}$ |
| 30 | $10.82 \pm 0$ | $65.78 \pm 5.39$ | $91.56 \pm 2.61$ | $\mathbf{93.55 \pm 1.94}$ |
| 40 | $10.82 \pm 0$ | $11.35 \pm 0$ | $11.35 \pm 0$ | $\mathbf{86.71 \pm 4.68}$ |
| 50 | $9.8 \pm 0$ | $9.8 \pm 0$ | $9.8 \pm 0$ | $\mathbf{26.91 \pm 7.53}$ |
| 60 | $9.8 \pm 0$ | $9.8 \pm 0$ | $9.8 \pm 0$ | $9.8 \pm 0$ |
| 70 | $9.8 \pm 0$ | $9.8 \pm 0$ | $9.8 \pm 0$ | $9.8 \pm 0$ |
| 80 | $9.8 \pm 0$ | $9.8 \pm 0$ | $9.8 \pm 0$ | $9.8 \pm 0$ |
| 90 | $9.8 \pm 0$ | $9.8 \pm 0$ | $9.8 \pm 0$ | $9.8 \pm 0$ |
| ACCURACY (%) WITH MLP (FMNIST) | | | | |
| 0 | $83.99 \pm 0.86$ | $78.09 \pm 1.03$ | $86.29 \pm 0.94$ | $\mathbf{86.8 \pm 0.89}$ |
| 10 | $83.74 \pm 1.79$ | $76.78 \pm 1.9$ | $85.49 \pm 1.43$ | $\mathbf{85.6 \pm 1.23}$ |
| 20 | $10 \pm 0$ | $74.22 \pm 2.54$ | $80.41 \pm 2.19$ | $\mathbf{85.33 \pm 1.45}$ |
| 30 | $10 \pm 0$ | $70.65 \pm 6.17$ | $79.6 \pm 3.72$ | $\mathbf{83.53 \pm 2.69}$ |
| 40 | $10 \pm 0$ | $17.16 \pm 8.84$ | $21.93 \pm 7.63$ | $\mathbf{54.55 \pm 6.76}$ |
| 50 | $10 \pm 0$ | $10 \pm 0$ | $10 \pm 0$ | $10 \pm 0$ |
| 60 | $10 \pm 0$ | $10 \pm 0$ | $10 \pm 0$ | $10 \pm 0$ |
| 70 | $10 \pm 0$ | $10 \pm 0$ | $10 \pm 0$ | $10 \pm 0$ |
| 80 | $10 \pm 0$ | $10 \pm 0$ | $10 \pm 0$ | $10 \pm 0$ |
| 90 | $10 \pm 0$ | $10 \pm 0$ | $10 \pm 0$ | $10 \pm 0$ |

Table 7 presents the classification accuracy of the baseline, horizontally-fixed fragmentation, Gini-based fragmentation using image complexity for the sensitivity score, and Gini-based fragmentation using image complexity and fault influence of the first layer for the sensitivity score (proposed) when using MNIST models to classify MNIST and FMNIST data samples. The proposed version significantly enhances the fault tolerance of the MLP model, as demonstrated by its performance on MNIST and FMNIST, compared to other settings. This is because the MLP model is vulnerable to faults in the first layer, as mentioned in Section 4. Thus, using fault influence for the sensitivity score enhances our mechanism's fault mitigation ability since it induces the mechanism to minimize the pre-activation magnitude. We also observe that the wrong fragmentation strategy degrades classification performance because it damages the information of data samples.

### B.4.2 VGG-7 AND RESNET-18 MODELS (CIFAR-10 AND CIFAR-100)

**Table 8:** The VGG-7 and ResNet-18 models' classification accuracy in a 95% confidence interval with different settings of our mechanism under SAFs.

| FAULT RATIO(%) | BASELINE | HORIZONTAL | GINI(COMPLEXITY ONLY) | GINI(PROPOSED) |
|---|---|---|---|---|
| ACCURACY (%) WITH VGG-7 (CIFAR-10) | | | | |
| 0 | $56.26 \pm 1.28$ | $55.75 \pm 1.33$ | $56.68 \pm 1.28$ | $\mathbf{56.86 \pm 1.63}$ |
| 10 | $40.09 \pm 3.34$ | $48.8 \pm 2.67$ | $49.13 \pm 2.32$ | $\mathbf{50.38 \pm 2.07}$ |
| 20 | $31.78 \pm 4.97$ | $42.69 \pm 3.83$ | $45.39 \pm 3.44$ | $\mathbf{45.58 \pm 3.76}$ |
| 30 | $11.74 \pm 4.81$ | $36.55 \pm 4.64$ | $38.23 \pm 4.91$ | $\mathbf{40.26 \pm 5.04}$ |
| 40 | $10.79 \pm 3.56$ | $29.09 \pm 5.72$ | $31.18 \pm 5.52$ | $\mathbf{32.07 \pm 5.15}$ |
| 50 | $10 \pm 0$ | $10 \pm 0$ | $10 \pm 0$ | $\mathbf{17.08 \pm 5.74}$ |
| 60 | $10 \pm 0$ | $10 \pm 0$ | $10 \pm 0$ | $10 \pm 0$ |
| 70 | $10 \pm 0$ | $10 \pm 0$ | $10 \pm 0$ | $10 \pm 0$ |
| 80 | $10 \pm 0$ | $10 \pm 0$ | $10 \pm 0$ | $10 \pm 0$ |
| 90 | $10 \pm 0$ | $10 \pm 0$ | $10 \pm 0$ | $10 \pm 0$ |
| ACCURACY (%) WITH RESNET-18 (CIFAR-100) | | | | |
| 0 | $23.77 \pm 4.56$ | $24.12 \pm 4.69$ | $25.11 \pm 4.74$ | $\mathbf{27.96 \pm 4.83}$ |
| 10 | $10.64 \pm 3.82$ | $23.98 \pm 4.41$ | $24.15 \pm 4.32$ | $\mathbf{25.5 \pm 4.79}$ |
| 20 | $6.1 \pm 2.65$ | $19.86 \pm 5.52$ | $21.92 \pm 5.48$ | $\mathbf{24.41 \pm 5.38}$ |
| 30 | $1.98 \pm 0.93$ | $16.62 \pm 4.63$ | $18.43 \pm 5.01$ | $\mathbf{19.72 \pm 5.23}$ |
| 40 | $1.03 \pm 0.82$ | $13.53 \pm 4.29$ | $14.26 \pm 4.57$ | $\mathbf{16.8 \pm 4.72}$ |
| 50 | $0.97 \pm 0.79$ | $11.89 \pm 4.17$ | $12.34 \pm 4.48$ | $\mathbf{13.13 \pm 4.52}$ |
| 60 | $1.01 \pm 0.86$ | $4.68 \pm 2.1$ | $5.12 \pm 2.96$ | $\mathbf{6.25 \pm 3.26}$ |
| 70 | $1 \pm 0$ | $2.25 \pm 1.12$ | $3.9 \pm 1.44$ | $\mathbf{4.27 \pm 1.56}$ |
| 80 | $1 \pm 0$ | $1 \pm 0$ | $1 \pm 0$ | $\mathbf{1.24 \pm 0.81}$ |
| 90 | $1 \pm 0$ | $1 \pm 0$ | $1 \pm 0$ | $1 \pm 0$ |

Table 8 exhibits our ablation study with VGG-7 and ResNet-18 models using CIFAR-10 and CIFAR-100. Although the proposed mechanism outperforms other settings, the improvement in fault mitigation ability is not as large as in the cases with the MLP models. This is because VGG-7 and ResNet-18 models are not as vulnerable to faults in the first layer as the MLP model since they have additional features to compensate for errors during gradient calculations. Therefore, containing the fault influence in the sensitivity score does not significantly enhance the fault mitigation ability of our mechanism.

## B.5 VARIOUS HYPERPARAMETER SETTINGS WITH OUR MECHANISM

We change $(\lambda_s, \lambda_w)$ to (0.5, 0.5), (0.5, 0.1), and (0.1, 0.5) to show the influence of hyperparameters on our mechanism.

### B.5.1 MLP MODEL (MNIST AND FMNIST)

**Table 9:** The MLP model's classification accuracy in a 95% confidence interval with different settings of the hyperparameter $(\lambda_s, \lambda_w)$ under SAFs.

| FAULT RATIO(%) | (0.5, 0.5) | (0.5, 0.1) | (0.1, 0.5) | (0.1, 0.1) (DEFAULT) |
|---|---|---|---|---|
| | ACCURACY (%) WITH MLP (MNIST) | | | |
| 0 | $97.28 \pm 0.53$ | $97.35 \pm 0.54$ | $97.29 \pm 0.51$ | $\mathbf{97.44 \pm 0.39}$ |
| 10 | $96.34 \pm 1.25$ | $96.58 \pm 1.38$ | $96.05 \pm 1.19$ | $\mathbf{97.25 \pm 0.98}$ |
| 20 | $95.29 \pm 1.67$ | $93.3 \pm 1.63$ | $95.43 \pm 1.58$ | $\mathbf{93.84 \pm 1.37}$ |
| 30 | $91.59 \pm 2.82$ | $92.15 \pm 2.97$ | $86.93 \pm 2.86$ | $\mathbf{93.55 \pm 1.94}$ |
| 40 | $9.8 \pm 0$ | $14.28 \pm 5.47$ | $10.48 \pm 1.39$ | $\mathbf{86.71 \pm 4.68}$ |
| 50 | $9.8 \pm 0$ | $12.6 \pm 5.61$ | $9.8 \pm 0$ | $\mathbf{26.91 \pm 7.53}$ |
| 60 | $9.8 \pm 0$ | $9.8 \pm 0$ | $9.8 \pm 0$ | $9.8 \pm 0$ |
| 70 | $9.8 \pm 0$ | $9.8 \pm 0$ | $9.8 \pm 0$ | $9.8 \pm 0$ |
| 80 | $9.8 \pm 0$ | $9.8 \pm 0$ | $9.8 \pm 0$ | $9.8 \pm 0$ |
| 90 | $9.8 \pm 0$ | $9.8 \pm 0$ | $9.8 \pm 0$ | $9.8 \pm 0$ |
| | ACCURACY (%) WITH MLP (FMNIST) | | | |
| 0 | $85.53 \pm 1.05$ | $85.56 \pm 1.14$ | $85.29 \pm 0.99$ | $\mathbf{86.8 \pm 0.89}$ |
| 10 | $85.46 \pm 1.46$ | $85.52 \pm 1.31$ | $85.23 \pm 1.36$ | $\mathbf{85.6 \pm 1.23}$ |
| 20 | $80.89 \pm 3.48$ | $83.57 \pm 2.12$ | $84.41 \pm 2.29$ | $\mathbf{85.33 \pm 1.45}$ |
| 30 | $80.23 \pm 3.8$ | $31.58 \pm 7.03$ | $75.39 \pm 5.45$ | $\mathbf{83.53 \pm 2.69}$ |
| 40 | $33.54 \pm 6.15$ | $18.84 \pm 7.65$ | $10 \pm 0$ | $\mathbf{54.55 \pm 6.76}$ |
| 50 | $10 \pm 0$ | $10 \pm 0$ | $10 \pm 0$ | $10 \pm 0$ |
| 60 | $10 \pm 0$ | $10 \pm 0$ | $10 \pm 0$ | $10 \pm 0$ |
| 70 | $10 \pm 0$ | $10 \pm 0$ | $10 \pm 0$ | $10 \pm 0$ |
| 80 | $10 \pm 0$ | $10 \pm 0$ | $10 \pm 0$ | $10 \pm 0$ |
| 90 | $10 \pm 0$ | $10 \pm 0$ | $10 \pm 0$ | $10 \pm 0$ |

Table 9 exhibits the MLP model's classification accuracy by changing ($\lambda_s$ and $\lambda_w$) under SAFs. The hyperparameter settings predominantly affect the classification accuracy of the MLP model, as they adjust how the mechanism mitigates the adverse influence of synaptic faults in the first layer, which damages the model the most severely. We observe that increasing the hyperparameters and strengthening the effects of the fault influence map do not always leverage the MLP model's fault tolerance. This is because the excessive effects of the fault influence map prevent our mechanism from setting the angle accurately by reflecting the complexity of the input samples and the fault influence in a balanced way. In addition, we observe that the weight projection map affects our mechanism more predominantly than the saliency map, since the weight projection map is more sensitive to changes in weights due to synaptic faults.

### B.5.2 VGG-7 AND RESNET-18 MODELS (CIFAR-10 AND CIFAR-100)

**Table 10:** The VGG-7 and ResNet-18 models' classification accuracy in a 95% confidence interval with different settings of the hyperparameter $(\lambda_s, \lambda_w)$ under SAFs.

| FAULT RATIO(%) | (0.5, 0.5) | (0.5, 0.1) | (0.1, 0.5) | (0.1, 0.1) (DEFAULT) |
|---|---|---|---|---|
| ACCURACY (%) WITH VGG-7 (CIFAR-10) | | | | |
| 0 | $56.48 \pm 1.84$ | $56.06 \pm 1.73$ | $56.03 \pm 1.89$ | $\mathbf{56.86 \pm 1.63}$ |
| 10 | $48.12 \pm 3.04$ | $48.88 \pm 2.85$ | $49.34 \pm 2.16$ | $\mathbf{50.38 \pm 2.07}$ |
| 20 | $44.68 \pm 3.69$ | $44.56 \pm 3.66$ | $45.3 \pm 3.59$ | $\mathbf{45.58 \pm 3.76}$ |
| 30 | $39.42 \pm 5.25$ | $39.22 \pm 5.59$ | $39.92 \pm 5.23$ | $\mathbf{40.26 \pm 5.04}$ |
| 40 | $29.62 \pm 5.38$ | $29.52 \pm 5.04$ | $30.21 \pm 5.4$ | $\mathbf{32.07 \pm 5.15}$ |
| 50 | $15.09 \pm 4.76$ | $15.28 \pm 5.17$ | $15.31 \pm 4.98$ | $\mathbf{17.08 \pm 5.74}$ |
| 60 | $10 \pm 0$ | $10 \pm 0$ | $10 \pm 0$ | $10 \pm 0$ |
| 70 | $10 \pm 0$ | $10 \pm 0$ | $10 \pm 0$ | $10 \pm 0$ |
| 80 | $10 \pm 0$ | $10 \pm 0$ | $10 \pm 0$ | $10 \pm 0$ |
| 90 | $10 \pm 0$ | $10 \pm 0$ | $10 \pm 0$ | $10 \pm 0$ |
| ACCURACY (%) WITH RESNET-18 (CIFAR-100) | | | | |
| 0 | $26.96 \pm 5.08$ | $26.02 \pm 4.93$ | $25.71 \pm 5.86$ | $\mathbf{27.96 \pm 4.83}$ |
| 10 | $24.26 \pm 4.29$ | $24.76 \pm 4.38$ | $21.75 \pm 4.71$ | $\mathbf{25.5 \pm 4.79}$ |
| 20 | $20.09 \pm 5.66$ | $21.23 \pm 5.96$ | $19.03 \pm 5.09$ | $\mathbf{24.41 \pm 5.38}$ |
| 30 | $16.88 \pm 5.48$ | $17.7 \pm 5.48$ | $15.89 \pm 5.42$ | $\mathbf{19.72 \pm 5.23}$ |
| 40 | $13.25 \pm 5.24$ | $13.68 \pm 5.05$ | $13.25 \pm 4.93$ | $\mathbf{16.8 \pm 4.71}$ |
| 50 | $12.04 \pm 4.93$ | $12.55 \pm 4.34$ | $9.14 \pm 4.69$ | $\mathbf{13.13 \pm 4.5}$ |
| 60 | $4.46 \pm 3.52$ | $4.77 \pm 3.15$ | $4.86 \pm 2.96$ | $\mathbf{6.25 \pm 3.26}$ |
| 70 | $3.61 \pm 1.95$ | $2.89 \pm 1.7$ | $3.05 \pm 1.78$ | $\mathbf{4.27 \pm 1.56}$ |
| 80 | $1 \pm 0$ | $1 \pm 0$ | $1 \pm 0$ | $\mathbf{1.24 \pm 0.81}$ |
| 90 | $1 \pm 0$ | $1 \pm 0$ | $1 \pm 0$ | $1 \pm 0$ |

Table 10 presents the VGG and ResNet Models' classification accuracy by changing ($\lambda_s$ and $\lambda_w$) under SAFs. Since the significance of the faults in the first layer of the VGG and ResNet Models is weaker than that of the MLP model, the sensitivity to hyperparameter setting is smaller than that of the MLP model. Despite the low significance of tuning hyperparameters when using the VGG and ResNet Models, setting the hyperparameters to the proper value is still important to ensure the best fault mitigation ability of our mechanism.

## B.6 USING DEEP NEURAL NETWORKS (DNNS)

We inject SAFs into synapses of Deep Neural Networks (DNNs) version of the SNN models, measuring their classification accuracy with the baseline, benchmarks, and proposed mechanism. We use Cross-Entropy (CE) as a loss function and Rectified Linear Unit (ReLU) as an activation function. We set the range of weights to [-100, 100] for MLP and [-500, 500] for CNNs (VGG and ResNet) since current DNN accelerator devices have a large weight range (Liu et al., 2019; Chen et al., 2017). Other settings are the same as the SNN models. We exclude Astrocyte, FalVolt, and LIFA from the benchmarks since they necessarily require bio-plausible spiking neuron models for operation. We use 2 fragments for our mechanism.

**Table 11:** The DNN models' classification accuracy in a 95% confidence interval using MNIST, FMNIST, CIFAR-10, and CIFAR-100 under SAFs.

| FAULT RATIO(%) | BASELINE | ECOC | SOFTSNN (TUNED FOR DNN) | ROUTING | PROPOSED |
|---|---|---|---|---|---|
| | | | ACCURACY(%) WITH MLP (MNIST) | | |
| 0 | $98.48 \pm 0.19$ | $\mathbf{98.6 \pm 0.26}$ | $98.49 \pm 0.23$ | $98.45 \pm 0.31$ | $98.54 \pm 0.25$ |
| 10 | $97.32 \pm 0.86$ | $97.28 \pm 0.81$ | $97.48 \pm 0.96$ | $97.28 \pm 0.89$ | $\mathbf{97.56 \pm 0.74}$ |
| 20 | $96.62 \pm 1.55$ | $9.8 \pm 0$ | $97.02 \pm 1.32$ | $96.76 \pm 1.47$ | $\mathbf{96.85 \pm 1.38}$ |
| 30 | $96.58 \pm 1.93$ | $9.8 \pm 0$ | $96.3 \pm 1.89$ | $96.64 \pm 1.69$ | $\mathbf{96.72 \pm 1.71}$ |
| 40 | $96.35 \pm 1.81$ | $9.8 \pm 0$ | $96.02 \pm 1.76$ | $96.43 \pm 1.74$ | $\mathbf{96.48 \pm 1.85}$ |
| 50 | $93.3 \pm 3.52$ | $9.8 \pm 0$ | $93.47 \pm 3.62$ | $93.54 \pm 4.02$ | $\mathbf{93.58 \pm 3.91}$ |
| 60 | $77.36 \pm 9.15$ | $9.8 \pm 0$ | $78.03 \pm 8.99$ | $79.89 \pm 8.56$ | $\mathbf{82.62 \pm 8.8}$ |
| 70 | $65.9 \pm 10.49$ | $9.8 \pm 0$ | $68.39 \pm 10.05$ | $67.79 \pm 9.95$ | $\mathbf{74.8 \pm 9.72}$ |
| 80 | $39.64 \pm 8.77$ | $9.8 \pm 0$ | $41.31 \pm 9.12$ | $18.71 \pm 8.89$ | $\mathbf{55.43 \pm 8.73}$ |
| 90 | $8.92 \pm 0$ | $9.8 \pm 0$ | $9.8 \pm 0$ | $15.5 \pm 5.7$ | $\mathbf{19.07 \pm 6.21}$ |
| | | | ACCURACY(%) WITH MLP (FMNIST) | | |
| 0 | $90.05 \pm 1.22$ | $\mathbf{91.24 \pm 1.3}$ | $90.08 \pm 1.24$ | $90.12 \pm 1.13$ | $90.51 \pm 1.19$ |
| 10 | $84.38 \pm 2.73$ | $10 \pm 0$ | $84.72 \pm 2.86$ | $86.03 \pm 2.69$ | $\mathbf{86.83 \pm 2.71}$ |
| 20 | $71.67 \pm 3.49$ | $10 \pm 0$ | $72.07 \pm 3.35$ | $74.51 \pm 3.48$ | $\mathbf{79.66 \pm 3.48}$ |
| 30 | $68.01 \pm 5.05$ | $10 \pm 0$ | $70.75 \pm 4.94$ | $73.54 \pm 4.75$ | $\mathbf{74.63 \pm 4.79}$ |
| 40 | $64.93 \pm 5.91$ | $10 \pm 0$ | $63.5 \pm 6.04$ | $64.06 \pm 5.97$ | $\mathbf{65.23 \pm 6.25}$ |
| 50 | $58.63 \pm 6.27$ | $10 \pm 0$ | $62.78 \pm 6.38$ | $60.92 \pm 6.09$ | $\mathbf{64.8 \pm 6.31}$ |
| 60 | $53.57 \pm 7.86$ | $10 \pm 0$ | $57.01 \pm 8.11$ | $54.51 \pm 9.23$ | $\mathbf{58.68 \pm 8.77}$ |
| 70 | $38.01 \pm 9.91$ | $10 \pm 0$ | $39.48 \pm 10.26$ | $35.12 \pm 9.84$ | $\mathbf{41.62 \pm 10.34}$ |
| 80 | $23.5 \pm 8.44$ | $10 \pm 0$ | $26.45 \pm 8.79$ | $25.41 \pm 7.98$ | $\mathbf{33.21 \pm 8.59}$ |
| 90 | $10 \pm 0$ | $10 \pm 0$ | $10 \pm 0$ | $10 \pm 0$ | $\mathbf{11.02 \pm 1.02}$ |
| | | | ACCURACY(%) WITH VGG-7 (CIFAR-10) | | |
| 0 | $83.21 \pm 2.76$ | $83.36 \pm 2.58$ | $83.58 \pm 2.53$ | $83.73 \pm 2.47$ | $\mathbf{84.29 \pm 2.63}$ |
| 0.01 | $10 \pm 0$ | $10 \pm 0$ | $52.97 \pm 4.84$ | $67.78 \pm 5.04$ | $\mathbf{70.36 \pm 4.81}$ |
| 0.025 | $10 \pm 0$ | $10 \pm 0$ | $10 \pm 0$ | $10 \pm 0$ | $10 \pm 0$ |
| 0.05 | $10 \pm 0$ | $10 \pm 0$ | $10 \pm 0$ | $10 \pm 0$ | $10 \pm 0$ |
| 0.075 | $10 \pm 0$ | $10 \pm 0$ | $10 \pm 0$ | $10 \pm 0$ | $10 \pm 0$ |
| | | | ACCURACY(%) WITH RESNET-18 (CIFAR-100) | | |
| 0 | $53.09 \pm 1.02$ | $40.96 \pm 1.56$ | $53.51 \pm 1.16$ | $53.6 \pm 1.24$ | $\mathbf{53.82 \pm 1.19}$ |
| 0.01 | $46.17 \pm 1.35$ | $37.87 \pm 1.53$ | $47.26 \pm 1.58$ | $46.99 \pm 1.56$ | $\mathbf{48.49 \pm 1.47}$ |
| 0.025 | $43.32 \pm 1.68$ | $34.05 \pm 1.79$ | $43.76 \pm 1.73$ | $44.07 \pm 1.79$ | $\mathbf{45.09 \pm 1.66}$ |
| 0.05 | $41.14 \pm 1.54$ | $32.28 \pm 2.01$ | $42.23 \pm 1.69$ | $42.51 \pm 1.55$ | $\mathbf{42.78 \pm 1.48}$ |
| 0.075 | $39.05 \pm 1.82$ | $31.64 \pm 1.98$ | $40.64 \pm 1.91$ | $39.89 \pm 2.05$ | $\mathbf{41.32 \pm 1.85}$ |

Table 11 presents the classification accuracy of DNN models with the baseline, benchmarks, and proposed mechanism. The models with ours exhibit the highest fault robustness among the DNN models since our mechanism prevents the pre-activation from increasing excessively, and gradients do not explode severely during training. The MLP DNN model presents higher fault tolerance than the SNN model regardless of the datasets. This is because the gradient vanishing caused by pre-activation magnitude growth does not occur severely, since gradients are active when the pre-activation is larger than 0 in ReLU. Conversely, the fault tolerance of the complicated CNN models (VGG and ResNet) degrades seriously. This situation appears because these models use lots of batch normalization layers. The normalization layers normalize the whole channels in the same scale calculated with the batch samples, causing the amplification of inputs that enter faulty synapses. However, the SNN VGG and ResNet models are more tolerant of faults than the DNN versions. This is because the spiking neurons block the perturbation from faults through their internal activation mechanism: only firing and emitting spikes when their membrane potential reaches the threshold (Liang et al., 2023).

## B.7 USING SNNS WITH UNSUPERVISED LEARNING

The SNN models using unsupervised learning are also important models to implement on-chip learning of neuromorphic devices. Therefore, we adopt the benchmarks and proposed mechanism to the widely used SNN model: Diehl&Cook2015 using Spike-Timing-Dependent Plasticity (STDP), which is a representative unsupervised learning rule (Lee & Lim, 2024; Diehl & Cook, 2015). We use a reliable framework to implement STDP-based SNNs, named BindsNET, for our experiments (Hazan et al., 2018).

**Table 12:** The Diehl&Cook2015 model's classification accuracy in a 95% confidence interval using MNIST and FMNIST with 250 time steps under SAFs.

| FAULT RATIO(%) | BASELINE | SOFTSNN | ROUTING | ASTROCYTE | FALVOLT | LIFA | PROPOSED |
|---|---|---|---|---|---|---|---|
| | | | ACCURACY (%) WITH DIEHL&COOK2015 (MNIST) | | | | |
| 0 | $86.37 \pm 1.23$ | $85.96 \pm 1.45$ | $86.34 \pm 1.28$ | $86.49 \pm 1.65$ | $\mathbf{86.41 \pm 1.61}$ | $86.16 \pm 1.54$ | $85.69 \pm 1.72$ |
| 10 | $77.07 \pm 1.86$ | $79.34 \pm 2.51$ | $78.5 \pm 2.08$ | $79.16 \pm 1.93$ | $78.89 \pm 2.02$ | $79.58 \pm 2.37$ | $\mathbf{81.03 \pm 2.44}$ |
| 20 | $76.41 \pm 1.9$ | $78.48 \pm 2.67$ | $77.27 \pm 2.36$ | $78.91 \pm 1.88$ | $78.04 \pm 2.45$ | $78.61 \pm 2.51$ | $\mathbf{80.27 \pm 2.53}$ |
| 30 | $74.45 \pm 2.79$ | $76.59 \pm 2.81$ | $77.03 \pm 3.05$ | $77.62 \pm 2.56$ | $78.04 \pm 3.16$ | $78.72 \pm 2.98$ | $\mathbf{79.86 \pm 3.14}$ |
| 40 | $70.87 \pm 4.52$ | $71.95 \pm 4.93$ | $72.35 \pm 4.74$ | $72.48 \pm 4.59$ | $73.19 \pm 4.72$ | $73.43 \pm 4.6$ | $\mathbf{76.57 \pm 3.66}$ |
| 50 | $69.7 \pm 5.15$ | $70.83 \pm 5.26$ | $72.08 \pm 4.91$ | $71.27 \pm 5.38$ | $71.96 \pm 5.84$ | $71.78 \pm 5.64$ | $\mathbf{74.36 \pm 5.42}$ |
| 60 | $67.79 \pm 5.98$ | $69.25 \pm 6.2$ | $70.7 \pm 6.12$ | $69.49 \pm 5.84$ | $70.35 \pm 6.27$ | $70.96 \pm 6.32$ | $\mathbf{72.18 \pm 5.96}$ |
| 70 | $63.46 \pm 5.76$ | $65.51 \pm 5.69$ | $65.28 \pm 5.57$ | $66.93 \pm 5.65$ | $65.97 \pm 5.96$ | $68.08 \pm 5.84$ | $\mathbf{70.4 \pm 5.71}$ |
| 80 | $54.62 \pm 4.84$ | $56.26 \pm 4.98$ | $55.93 \pm 5.05$ | $57.15 \pm 4.72$ | $56.59 \pm 5.13$ | $59.27 \pm 4.99$ | $\mathbf{62.11 \pm 5.27}$ |
| 90 | $33.25 \pm 3.61$ | $38.59 \pm 4.06$ | $40.27 \pm 3.96$ | $39.45 \pm 4.11$ | $41.73 \pm 4.5$ | $43.98 \pm 4.76$ | $\mathbf{52.64 \pm 4.98}$ |
| | | | ACCURACY (%) WITH DIEHL&COOK2015 (FMNIST) | | | | |
| 0 | $27.4 \pm 2.68$ | $27.61 \pm 2.75$ | $27.57 \pm 2.46$ | $27.43 \pm 2.37$ | $\mathbf{27.77 \pm 2.84}$ | $27.59 \pm 2.7$ | $26.76 \pm 2.36$ |
| 10 | $22.66 \pm 3.83$ | $24.54 \pm 3.96$ | $24.8 \pm 3.69$ | $24.96 \pm 3.92$ | $25.11 \pm 3.63$ | $25.7 \pm 3.91$ | $\mathbf{26.17 \pm 3.94}$ |
| 20 | $20.84 \pm 3.95$ | $22.55 \pm 4.12$ | $22.89 \pm 4.17$ | $22.74 \pm 3.86$ | $22.68 \pm 4.09$ | $23.59 \pm 4.28$ | $\mathbf{23.85 \pm 4.47}$ |
| 30 | $18.81 \pm 4.06$ | $20.72 \pm 4.27$ | $21.63 \pm 4.15$ | $20.98 \pm 4.07$ | $21.16 \pm 3.96$ | $22.07 \pm 3.96$ | $\mathbf{22.83 \pm 4.25}$ |
| 40 | $15.47 \pm 2.97$ | $16.8 \pm 3.11$ | $17.25 \pm 3.08$ | $16.97 \pm 3.2$ | $17.15 \pm 3.19$ | $18.23 \pm 3.38$ | $\mathbf{20.65 \pm 3.8}$ |
| 50 | $12.35 \pm 1.76$ | $13.15 \pm 2.57$ | $12.98 \pm 2.61$ | $13.41 \pm 2.85$ | $13.61 \pm 2.79$ | $14.02 \pm 2.91$ | $\mathbf{14.58 \pm 3.13}$ |
| 60 | $10 \pm 0$ | $10 \pm 0$ | $10 \pm 0$ | $10 \pm 0$ | $10 \pm 0$ | $10 \pm 0$ | $10 \pm 0$ |
| 70 | $10 \pm 0$ | $10 \pm 0$ | $10 \pm 0$ | $10 \pm 0$ | $10 \pm 0$ | $10 \pm 0$ | $10 \pm 0$ |
| 80 | $10 \pm 0$ | $10 \pm 0$ | $10 \pm 0$ | $10 \pm 0$ | $10 \pm 0$ | $10 \pm 0$ | $10 \pm 0$ |
| 90 | $10 \pm 0$ | $10 \pm 0$ | $10 \pm 0$ | $10 \pm 0$ | $10 \pm 0$ | $10 \pm 0$ | $10 \pm 0$ |

Table 12 shows the classification accuracy of Deihl&Cook2015 with the benchmarks and proposed mechanism under SAFs. We set the number of time steps to 250 and the number of fragments to 2. The model obtains the first fragment repeatedly during the first 125 time steps and the second fragment during the second 125 time steps. We apply this setting to Deihl&Cook2015 since it cannot classify data samples as accurately as SNN models with supervised learning (Diehl & Cook, 2015). We exclude ECOC from the experiment due to its implementation difficulty in STDP-based SNN models. This is because it is not available in Diehl&Cook since it disturbs the Winner-Takes-All (WTA) mechanism in Diehl&Cook2015 (Lee & Lim, 2024). As presented in the table, the model with our mechanism has the highest accuracy under SAFs. This is because our mechanism can mitigate the adversarial effects of hardware faults for the following reasons. Hardware faults excessively increase the absolute value of membrane potential (pre-activation in SNN using gradient-based learning rules), and this unnatural increase causes the over-firing of spiking neurons of the Diehl&Cook2015 (Putra et al., 2022; Han et al., 2023; Rastogi et al., 2021). When faulty Deihl&Cook2015 obtains the whole data samples that are not fragmented, fault-injected neurons' membrane potential always increases or decreases significantly, preventing the neurons from spiking properly since the pixel values easily enter the fault-injected synapses. We divide the input samples into small pieces by considering the complexity of the input samples and the influence of faults to minimize the adversarial effects of faults and prevent neurons from over- or under-firing. Through fragmentation, the proposed mechanism reduces the probability that the input pixels enter faulty synapses, and the neurons' membrane potential avoids increasing or decreasing abnormally. Interestingly, the astrocyte-based approaches show high fault mitigation ability integrated with Diehl&Cook2015. This is because they aim to improve the fault tolerance of SNN models using STDP (Han et al., 2023; Yunusoglu et al., 2025).

## B.8 Evaluations with real FPGA hardware

We implement the MLP SNN model on a real FPGA device (AMD Virtex UltraScale+ HBM VU47P of Amazon F2 instance) with SpikerPlus, which is a powerful library to convert Python scripts for SNNs to VHSIC Hardware Description Language (VHDL) (Carpegna et al., 2024). We choose the FPGA device for SNN implementation because FPGAs are necessary devices to develop hardware-based SNN models (Karamimanesh et al., 2025). Due to circuit-level limitations, current FPGA-based SNN models do not support on-chip training (Carpegna et al., 2024; Tao et al., 2020). Thus, we train the fault-injected SNN model in a software environment with a Graphics Processing Unit (GPU), saving the trained weights, and convert the Python script of the software-based models to the VHDL script. Then, we synthesize the FPGA circuit with Xilinx Vivado and Amazon FPGA Image (AFI), which is widely used for handling FPGAs on Amazon F2 instances. We build the SNN models on the FPGA chip and the proposed mechanism on the additional control processor connected to the FPGA chip. We set the bit-width for the membrane potential and the synaptic weights of the FPGA device to 8 and 6, respectively. We set the floating-point precision of the input data samples to 32. We select this setting, referring to the setting of real hardware (BrainChipInc, 2025). Other settings are the same as the settings in Subsection 6.1.

**Table 13:** The FPGA-based MLP SNNs' classification accuracy in a 95% confidence interval using MNIST, FMNIST, UCI-HAR, and AudioMNIST with 2 time steps under SAFs. Note that we adopt 2 time steps for training. For inference, we use 100 cycles to process data in the FPGA device.

| Fault ratio(%) | Baseline | ECOC | SoftSNN | Routing | Astrocyte | FalVolt | LIFA | Proposed |
|---|---|---|---|---|---|---|---|---|
| | | | Accuracy (%) with Hardware-implemented MLP (MNIST) | | | | | |
| 0 | 94.36 ± 0.52 | 93.86 ± 0.66 | 94.56 ± 0.63 | 94.34 ± 0.71 | 94.06 ± 0.68 | 93.92 ± 0.79 | 93.98 ± 0.85 | **94.59 ± 0.65** |
| 10 | 93.18 ± 1.29 | 90.94 ± 2.35 | 93.29 ± 1.56 | 93.58 ± 1.9 | 93.26 ± 1.44 | 92.69 ± 1.62 | 92.59 ± 1.74 | **94.01 ± 1.41** |
| 20 | 11.35 ± 0 | 88.11 ± 2.74 | 9.8 ± 0 | 9.8 ± 0 | 10.1 ± 1.05 | 89.97 ± 2.69 | 15.92 ± 5.31 | **91.14 ± 2.03** |
| 30 | 10.82 ± 0 | 11.35 ± 0 | 9.8 ± 0 | 9.8 ± 0 | 10.82 ± 0 | 9.8 ± 0 | 9.8 ± 0 | **90.57 ± 3.16** |
| 40 | 9.8 ± 0 | 9.8 ± 0 | 9.8 ± 0 | 11.35 ± 0 | 9.8 ± 0 | 9.8 ± 0 | 9.8 ± 0 | **80.49 ± 6.65** |
| 50 | 9.8 ± 0 | 9.8 ± 0 | 9.8 ± 0 | 11.35 ± 0 | 9.8 ± 0 | 9.8 ± 0 | 9.8 ± 0 | **21.46 ± 4.72** |
| 60 | 9.8 ± 0 | 9.8 ± 0 | 9.8 ± 0 | 9.8 ± 0 | 9.8 ± 0 | 9.8 ± 0 | 9.8 ± 0 | 9.8 ± 0 |
| 70 | 9.8 ± 0 | 9.8 ± 0 | 9.8 ± 0 | 9.8 ± 0 | 9.8 ± 0 | 9.8 ± 0 | 9.8 ± 0 | 9.8 ± 0 |
| 80 | 9.8 ± 0 | 9.8 ± 0 | 9.8 ± 0 | 9.8 ± 0 | 9.8 ± 0 | 9.8 ± 0 | 9.8 ± 0 | 9.8 ± 0 |
| 90 | 9.8 ± 0 | 9.8 ± 0 | 9.8 ± 0 | 9.8 ± 0 | 9.8 ± 0 | 9.8 ± 0 | 9.8 ± 0 | 9.8 ± 0 |
| | | | Accuracy (%) with Hardware-implemented MLP (FMNIST) | | | | | |
| 0 | 84.21 ± 1.79 | 83.58 ± 1.91 | 83.65 ± 2.13 | 84.19 ± 1.82 | 83.9 ± 1.75 | 83.97 ± 1.68 | 84.1 ± 1.58 | **87.01 ± 1.45** |
| 10 | 82.16 ± 2.28 | 82.75 ± 2.46 | 80.74 ± 2.67 | 83.58 ± 2.33 | 82.86 ± 2.09 | 83.11 ± 2.53 | 82.46 ± 2.38 | **86.14 ± 2.84** |
| 20 | 79.6 ± 3.05 | 10 ± 0 | 10 ± 0 | 10 ± 0 | 10 ± 0 | 82.54 ± 3.18 | 17.26 ± 5.98 | **83.98 ± 3.27** |
| 30 | 10 ± 0 | 10 ± 0 | 10 ± 0 | 10 ± 0 | 10 ± 0 | 10 ± 0 | 10 ± 0 | **81.71 ± 3.7** |
| 40 | 10 ± 0 | 10 ± 0 | 10 ± 0 | 10 ± 0 | 10 ± 0 | 10 ± 0 | 10 ± 0 | **52.75 ± 4.89** |
| 50 | 10 ± 0 | 10 ± 0 | 10 ± 0 | 10 ± 0 | 10 ± 0 | 10 ± 0 | 10 ± 0 | 10 ± 0 |
| 60 | 10 ± 0 | 10 ± 0 | 10 ± 0 | 10 ± 0 | 10 ± 0 | 10 ± 0 | 10 ± 0 | 10 ± 0 |
| 70 | 10 ± 0 | 10 ± 0 | 10 ± 0 | 10 ± 0 | 10 ± 0 | 10 ± 0 | 10 ± 0 | 10 ± 0 |
| 80 | 10 ± 0 | 10 ± 0 | 10 ± 0 | 10 ± 0 | 10 ± 0 | 10 ± 0 | 10 ± 0 | 10 ± 0 |
| 90 | 10 ± 0 | 10 ± 0 | 10 ± 0 | 10 ± 0 | 10 ± 0 | 10 ± 0 | 10 ± 0 | 10 ± 0 |
| | | | Accuracy (%) with Hardware-implemented MLP (UCI-HAR) | | | | | |
| 0 | 60.14 ± 3.87 | **65.27 ± 3.56** | 60.48 ± 3.81 | 61.15 ± 3.69 | 61.78 ± 3.48 | 61.83 ± 3.45 | 61.52 ± 3.62 | 60.08 ± 3.24 |
| 10 | 17.1 ± 0 | 17.1 ± 0 | 15.21 ± 0 | 17.1 ± 0 | 15.21 ± 0 | 18.34 ± 0 | 16.93 ± 0 | **47.01±5.17** |
| 20 | 18.52 ± 0 | 17.1 ± 0 | 18.52 ± 0 | 17.1 ± 0 | 18.52 ± 0 | 18.34 ± 0 | 17.1 ± 0 | **45.85 ± 5.39** |
| 30 | 17.1 ± 0 | 17.1 ± 0 | 17.1 ± 0 | 17.1 ± 0 | 17.1 ± 0 | 18.34 ± 0 | 17.1 ± 0 | **44.86 ± 4.91** |
| 40 | 17.1 ± 0 | 17.1 ± 0 | 17.1 ± 0 | 17.1 ± 0 | 17.1 ± 0 | 16.93 ± 0 | 17.1 ± 0 | 17.1 ± 0 |
| 50 | 17.1 ± 0 | 17.1 ± 0 | 17.1 ± 0 | 17.1 ± 0 | 17.1 ± 0 | 16.93 ± 0 | 17.1 ± 0 | 17.1 ± 0 |
| 60 | 17.1 ± 0 | 17.1 ± 0 | 17.1 ± 0 | 17.1 ± 0 | 17.1 ± 0 | 16.93 ± 0 | 17.1 ± 0 | 17.1 ± 0 |
| 70 | 17.1 ± 0 | 17.1 ± 0 | 17.1 ± 0 | 17.1 ± 0 | 17.1 ± 0 | 16.93 ± 0 | 17.1 ± 0 | 17.1 ± 0 |
| 80 | 17.1 ± 0 | 17.1 ± 0 | 17.1 ± 0 | 17.1 ± 0 | 17.1 ± 0 | 16.93 ± 0 | 17.1 ± 0 | 17.1 ± 0 |
| 90 | 17.1 ± 0 | 17.1 ± 0 | 17.1 ± 0 | 17.1 ± 0 | 17.1 ± 0 | 16.93 ± 0 | 17.1 ± 0 | 17.1 ± 0 |
| | | | Accuracy (%) with Hardware-implemented MLP (AudioMNIST) | | | | | |
| 0 | 93.76 ± 1.34 | **93.82 ± 1.6** | 92.02 ± 1.66 | 92.63 ± 1.21 | 92.36 ± 1.53 | 91.38 ± 1.27 | 93.33 ± 1.29 | 92.34 ± 1.38 |
| 10 | 91.80 ± 1.22 | 90.89 ± 2.12 | 89.32 ± 2.01 | 89.87 ± 2.22 | 90.01 ± 2.45 | 90.71 ± 2.45 | 90.41 ± 2.74 | **91.56 ± 1.93** |
| 20 | 89.45 ± 2.1 | 88.92 ± 2.56 | 84.39 ± 3.41 | 89.59 ± 2.50 | 81.61 ± 3.81 | 89.42 ± 2.68 | 84.47 ± 4.79 | **90.86 ± 2.83** |
| 30 | 89.04 ± 2.44 | 89.65 ± 2.77 | 72.35 ± 3.93 | 88.19 ± 3.28 | 48.31 ± 5.73 | 89.57 ± 3.01 | 51.64 ± 6.53 | **89.8 ± 3.42** |
| 40 | 86.24 ± 3.59 | 90.26 ± 2.58 | 59.43 ± 5.48 | 87.99 ± 3.97 | 34.69 ± 6.61 | 87.8 ± 3.42 | 35.73 ± 7.49 | **89.27 ± 4.13** |
| 50 | 83.50 ± 4.63 | 82.63 ± 4.43 | 45.26 ± 7.31 | 82.39 ± 4.42 | 19.38 ± 5.2 | 83.21 ± 4.93 | 21.22 ± 5.56 | **86.29 ± 4.9** |
| 60 | 49.11 ± 6.32 | 62.20 ± 5.82 | 31.23 ± 5.78 | 61.16 ± 5.94 | 10 ± 0 | 62.48 ± 6.63 | 10 ± 0 | **63.88 ± 6.34** |
| 70 | 10 ± 0 | 16.74 ± 2.45 | 10 ± 0 | 21.92 ± 3.84 | 10 ± 0 | 24.72 ± 4.47 | 10 ± 0 | **34.75 ± 3.76** |
| 80 | 10 ± 0 | 10 ± 0 | 10 ± 0 | 10 ± 0 | 10 ± 0 | 13.15 ± 2.14 | 10 ± 0 | **17.29 ± 3.55** |
| 90 | 10 ± 0 | 10 ± 0 | 10 ± 0 | 10 ± 0 | 10 ± 0 | 10 ± 0 | 10 ± 0 | 10 ± 0 |

Table 13 presents the classification accuracy of the MLP model, along with the baseline, benchmarks, and our mechanism, under SAFs, using the MNIST, FMNIST, UCI-HAR, and AudioMNIST. The model incorporating our mechanism classifies the data samples more accurately than the baseline and benchmarks when implemented with hardware. We observe that the overall classification accuracy of the model decreases. This is because the precision for neurons' membrane potential and synaptic weights degrades due to low floating-point and bit width in the FPGA device, which damages the data stored in trained synaptic weights and neuronal activities.

We implement VGG-7/11/15 with the SyncNN framework and evaluate the fault tolerance of the benchmarks and proposed mechanism (Panchapakesan et al., 2021). We set the bit-width for synaptic weights to 8. Other settings for SyncNN are the same as the settings in the SyncNN paper (Panchapakesan et al., 2021). We adopt the same SNN model settings in Subsection 6.1 to the FPGA-based VGG models.

**Table 14:** The FPGA-based VGG-7/11/15 SNNs' classification accuracy in a 95% confidence interval using CIFAR-10 with 2 time steps under SAFs. Note that we adopt 2 time steps for training. For inference, we use 1800 cycles to process data in the FPGA device.

| Fault ratio(%) | Baseline | ECOC | SoftSNN | Routing | Astrocyte | FalVolt | LIFA | Proposed |
|---|---|---|---|---|---|---|---|---|
| | | | Accuracy (%) with Hardware-implemented VGG-7 | | | | | |
| 0 | 53.84 ± 4.94 | 51.09 ± 4.86 | 54.08 ± 4.67 | **54.15 ± 4.26** | 53.21 ± 4.6 | 53.91 ± 4.57 | 52.72 ± 4.65 | 53.89 ± 4.48 |
| 10 | 34.18 ± 4.15 | 39.26 ± 4.07 | 22.48 ± 4.46 | 33.37 ± 4.12 | 32.16 ± 4.54 | 36.92 ± 4.82 | 35.38 ± 4.73 | **45.93 ± 4.31** |
| 20 | 18.73 ± 3.07 | 31.97 ± 3.65 | 17.52 ± 5.11 | 24.06 ± 3.71 | 22.18 ± 3.95 | 31.91 ± 3.79 | 21.99 ± 4.02 | **41.23 ± 4.05** |
| 30 | 11.62 ± 1.62 | 25.78 ± 3.14 | 16.8 ± 3.92 | 10 ± 0 | 10 ± 0 | 20.7 ± 4.15 | 10 ± 0 | **35.97 ± 4.88** |
| 40 | 10 ± 0 | 19.29 ± 3.28 | 15.04 ± 3.17 | 10 ± 0 | 10 ± 0 | 17.28 ± 3.84 | 10 ± 0 | **26.29 ± 3.96** |
| 50 | 10 ± 0 | 12.81 ± 2.25 | 10 ± 0 | 10 ± 0 | 10 ± 0 | 10 ± 0 | 10 ± 0 | **13.39 ± 2.94** |
| 60 | 10 ± 0 | 10 ± 0 | 10 ± 0 | 10 ± 0 | 10 ± 0 | 10 ± 0 | 10 ± 0 | 10 ± 0 |
| 70 | 10 ± 0 | 10 ± 0 | 10 ± 0 | 10 ± 0 | 10 ± 0 | 10 ± 0 | 10 ± 0 | 10 ± 0 |
| 80 | 10 ± 0 | 10 ± 0 | 10 ± 0 | 10 ± 0 | 10 ± 0 | 10 ± 0 | 10 ± 0 | 10 ± 0 |
| 90 | 10 ± 0 | 10 ± 0 | 10 ± 0 | 10 ± 0 | 10 ± 0 | 10 ± 0 | 10 ± 0 | 10 ± 0 |
| | | | Accuracy (%) with Hardware-implemented VGG-11 | | | | | |
| 0 | 52.74 ± 4.63 | 50.34 ± 4.51 | **54.82 ± 4.32** | 54.29 ± 4.45 | 52.13 ± 4.63 | 53.89 ± 4.38 | 53.02 ± 4.47 | 52.87 ± 4.2 |
| 10 | 33.27 ± 3.95 | 33.02 ± 3.84 | 26.53 ± 3.72 | 32.98 ± 3.93 | 34.16 ± 4.05 | 41.81 ± 4.24 | 35.57 ± 4.13 | **45.02 ± 4.08** |
| 20 | 26.91 ± 3.28 | 30.83 ± 3.37 | 18.88 ± 3.6 | 24.17 ± 3.56 | 24.93 ± 3.75 | 27.3 ± 3.43 | 24.89 ± 3.94 | **42.58 ± 3.79** |
| 30 | 10 ± 0 | 25.65 ± 3.51 | 17.63 ± 3.92 | 10 ± 0 | 16.89 ± 3.81 | 10 ± 0 | 11.46 ± 1.46 | **36.62 ± 3.84** |
| 40 | 10 ± 0 | 20.27 ± 3.64 | 10 ± 0 | 10 ± 0 | 10 ± 0 | 10 ± 0 | 10 ± 0 | **26.84 ± 3.29** |
| 50 | 10 ± 0 | 14.44 ± 3.76 | 10 ± 0 | 10 ± 0 | 10 ± 0 | 10 ± 0 | 10 ± 0 | **17.92 ± 3.65** |
| 60 | 10 ± 0 | 10 ± 0 | 10 ± 0 | 10 ± 0 | 10 ± 0 | 10 ± 0 | 10 ± 0 | 10 ± 0 |
| 70 | 10 ± 0 | 10 ± 0 | 10 ± 0 | 10 ± 0 | 10 ± 0 | 10 ± 0 | 10 ± 0 | 10 ± 0 |
| 80 | 10 ± 0 | 10 ± 0 | 10 ± 0 | 10 ± 0 | 10 ± 0 | 10 ± 0 | 10 ± 0 | 10 ± 0 |
| 90 | 10 ± 0 | 10 ± 0 | 10 ± 0 | 10 ± 0 | 10 ± 0 | 10 ± 0 | 10 ± 0 | 10 ± 0 |
| | | | Accuracy (%) with Hardware-implemented VGG-15 | | | | | |
| 0 | 45.04 ± 4.07 | 41.92 ± 3.95 | **47.39 ± 3.81** | 43.58 ± 3.86 | 43.02 ± 3.77 | 45.58 ± 3.26 | 44.61 ± 3.74 | 44.35 ± 4.61 |
| 10 | 40.16 ± 5.24 | 31.19 ± 4.76 | 14.21 ± 1.91 | 41.5 ± 4.83 | 32.28 ± 5.13 | 41.87 ± 4.59 | 29.84 ± 5.33 | **42.94 ± 5.09** |
| 20 | 21.42 ± 4.64 | 23.7 ± 4.59 | 13.22 ± 2.08 | 25.28 ± 4.29 | 10 ± 0 | 10 ± 0 | 20.37 ± 3.91 | **41.47 ± 4.96** |
| 30 | 10 ± 0 | 21.26 ± 3.54 | 12.89 ± 1.75 | 10 ± 0 | 10 ± 0 | 10 ± 0 | 10 ± 0 | **33.62 ± 5.17** |
| 40 | 10 ± 0 | 15.28 ± 2.94 | 11.01 ± 1.01 | 10 ± 0 | 10 ± 0 | 10 ± 0 | 10 ± 0 | **27.59 ± 4.25** |
| 50 | 10 ± 0 | 10 ± 0 | 10 ± 0 | 10 ± 0 | 10 ± 0 | 10 ± 0 | 10 ± 0 | **11.74 ± 1.19** |
| 60 | 10 ± 0 | 10 ± 0 | 10 ± 0 | 10 ± 0 | 10 ± 0 | 10 ± 0 | 10 ± 0 | 10 ± 0 |
| 70 | 10 ± 0 | 10 ± 0 | 10 ± 0 | 10 ± 0 | 10 ± 0 | 10 ± 0 | 10 ± 0 | 10 ± 0 |
| 80 | 10 ± 0 | 10 ± 0 | 10 ± 0 | 10 ± 0 | 10 ± 0 | 10 ± 0 | 10 ± 0 | 10 ± 0 |
| 90 | 10 ± 0 | 10 ± 0 | 10 ± 0 | 10 ± 0 | 10 ± 0 | 10 ± 0 | 10 ± 0 | 10 ± 0 |

Table 14 exhibits the classification accuracy of the VGG-7/11/15 models, along with the baseline, benchmarks, and proposed mechanism under SAFs, using CIFAR-10. In the cases with deep convolution SNNs, our mechanism successfully enhances the fault tolerance of hardware-implemented SNNs based on the FPGA device.

## C EFFICIENCY ANALYSIS BASED ON TIME/SPATIAL COMPLEXITY AND TIME/ENERGY CONSUMPTION

To demonstrate that our mechanism enhances the fault tolerance of SNN models without requiring complex algorithms, we measure the computational and spatial complexities of our mechanism and compare them to those of the benchmarks. Additionally, we measure the energy consumption of our mechanisms on the FPGA device since energy consumption is a significant advantage of SNNs that makes them suitable for neuromorphic device implementation.

### C.1 COMPLEXITY ANALYSIS

We calculate the time and spatial complexities of the benchmarks and the proposed mechanism through experimental evidence that demonstrates the time and energy consumption of them in real devices.

#### C.1.1 TIME COMPLEXITY

We thoroughly analyze the time complexity of the benchmarks and the proposed mechanism. The following items present the time complexity of the benchmarks and the proposed mechanism.

1. **ECOC**: $T_{\mathrm{ECOC}} = \Theta\big(EP^2 n \ + \ NBCL\big)$.

2. **SoftSNN**: $T_{\mathrm{SoftSNN}} = \Theta\big(NW\big)$.

3. **Routing**: $T_{\mathrm{route}} = \Theta\bigg( \sum_{\ell=1}^{L} C_{\mathrm{out}}^{(\ell)} C_{\mathrm{in}}^{(\ell)} k_\ell^2 \ + \ \delta_{\mathrm{swap}} \sum_{\ell=1}^{L} K_\ell \log K_\ell \bigg)$.

4. **Astrocyte**: $\mathrm{T}_{\mathrm{Astro}} = \Theta\big(P + N P\big)$.

5. **FalVolt**: $\mathrm{T}_{\mathrm{FalVolt}} = \Theta\big(M \ + \ N\,(\,M \ + \ fW\,)\big)$.

6. **LIFA**: $\mathrm{T}_{\mathrm{LIFA}} = \Theta\big(P + NP\big)$.

7. **Proposed**: $T_{\mathrm{proposed,\,mlp}} = \Theta\big(N\,S\,[\,B+A+T+BTD\,]\big) \ + \ O\big(N_{\mathrm{sal}}\,F_{\mathrm{model}}^{\mathrm{MLP}}\big)$,
   $T_{\mathrm{proposed,\,conv}} = \Theta\big(N\,S\,[\,B+A+T+BTD\,]\big)$.

**Notations of time complexity equations**

1. **ECOC.** $B$: batch size;  $N$: number of training/inference steps (batches processed);  $C$: number of classes;  $E$: number of extension code blocks;  $m$: Hamming-code parameter; $n = 2^m - 1$: per code block length;  $L = E\,n$: code length;  $P$: candidate-pool size used in code book construction.

2. **Soft SNN.** $W = \sum_{i=1}^{M} P_i$: total number of trainable weights over the $M$ layers; $P_i$: number of weights in layer $i$; $P_{\max} = \max_i P_i$: size of the largest layer.

3. **Routing.** $L_{\mathrm{layers}}$: number of routed layers;  $C_{\mathrm{in}}^{(\ell)}, C_{\mathrm{out}}^{(\ell)}$: input/output channels of layer $\ell$;  $k_\ell$: kernel size of layer $\ell$ (so $k_\ell^2 = 1$ for MLP/Linear);  $K_\ell = \min\{C_{\mathrm{in}}^{(\ell)}, C_{\mathrm{out}}^{(\ell)}\}$: effective channel count for top-$K$ matching;  $\delta_{\mathrm{swap}} \in \{0, 1\}$: flag indicating whether sorting + channel-swap is enabled;  $W$: total number of trainable weights across routed layers.

4. **Astrocyte.** $N$: number of batch iterations in an epoch;  $P$: total number of trainable parameters over hooked layers;  $\chi$: output-channel chunk size used in the backward pass; $p_{\mathrm{out}}^{\max}$: maximum number of parameters associated with a single output channel (e.g., $9\,C_{\mathrm{in}}$ for a $3{\times}3$ conv).

5. **FalVolt.** $N$: number of batch iterations in an epoch;  $W$: total number of weights subject to potential fault mapping;  $M$: number of spiking/protected modules whose thresholds or states are managed;  $f \in [0, 1]$: fraction of weights affected by faults (worst case $f = 1$); $W_{\mathrm{mask}}$: number of stored fault-mask entries.

6. **LIFA.** $N$: number of batch iterations in an epoch; $P$: total number of trainable parameters across protected layers (for conv: $P = \sum_\ell C_{\text{out}}^{(\ell)} C_{\text{in}}^{(\ell)} k_\ell^2$; for linear: $k_\ell^2 = 1$); $C = \sum_\ell C_{\text{out}}^{(\ell)}$: total output-channel count across protected layers.

7. **Proposed.** $N$: number of batch iterations in an epoch; $S = H \times W$: spatial size (pixels) per sample; $B$: batch size; $D$: input channels; $T$: number of time steps (fragments) per sample; $A$: number of orientation candidates; *(optional only if the fault influence map (saliency and weight projection) is used)* $N_{\text{sal}}$: number of steps that compute saliency/backprop; $F_{\text{model}}^{\text{MLP}}$: per-step FLOPs of the MLP backbone under saliency.

With MNIST and FMNIST ($S = 28 \times 28 = 784$, $D = 1$), our fragmentation step scales as

$$T_{\text{proposed}} = \Theta\big(S\,B\,T\big) = \Theta\big(784\,B\,T\big), \tag{5}$$

while all weight-scanning benchmarks (Astrocyte and LIFA) based on astrocytes scale with the number of parameters:

$$T_{\text{scan}} = \Theta(W), \qquad T_{\text{FalVolt}} = \Theta(M + fW) \asymp \Theta(fW) \ \text{(for non-vanishing } f\text{)}, \tag{6}$$

where $W$ is the total trainable weights, $M$ the number of spiking/protected modules, and $f \in [0, 1]$ the fraction of weights affected by faults. Hence, the decisive ratios are

$$\frac{T_{\text{proposed}}}{T_{\text{scan}}} \ \asymp \ \frac{784\,B\,T}{W}, \qquad \frac{T_{\text{proposed}}}{T_{\text{FalVolt}}} \ \asymp \ \frac{784\,B\,T}{f\,W}. \tag{7}$$

For the CNNs in our setting (VGG-7/11/15, ResNet-18), $W$ is in the multi-million range even on MNIST/FMNIST; with common batches/fragments ($B \in [64, 128]$, $T \in [2, 4]$) one has $W \gg 784\,B\,T$, so strictly $T_{\text{proposed}} < T_{\text{scanning}}$. The same conclusion holds against FalVolt for any fixed, non-negligible $f$ (e.g., $f \geq 0.05$), since then $fW \gg 784\,B\,T$ in these networks, yielding $T_{\text{ours}} < T_{\text{FalVolt}}$ as well. ECOC differs in that its per-step cost is $T_{\text{ECOC}} = \Theta(B\,C\,L)$, giving

$$\frac{T_{\text{proposed}}}{T_{\text{ECOC}}} \ \asymp \ \frac{S\,T\,D}{C\,L} = \frac{784\,T}{C\,L}, \tag{8}$$

which is typically of the same order for $C{=}10$ and $L \in [64, 256]$, while ECOC still incurs a one-off build of $\Theta(E\,P^2 n)$. In summary, without the fault influence, our mechanism has strictly smaller per-step time complexity than all weight-scanning benchmarks under the MLP/VGG/ResNet models. It is competitive with (or smaller than) ECOC while avoiding the heavy one-time construction of ECOC. As shown in Appendix A.1 and A.2, our mechanism is not significantly dependent on the fault influence with CIFAR-10 and CIFAR-100 under VGG and ResNet models, indicating that our mechanism saves time by using only the complexity to make fragments under VGG and ResNet models with CIFAR-10 and CIFAR-100.

### C.1.2 SPATIAL COMPLEXITY

We evaluate the spatial complexity of the benchmarks and the proposed mechanism in detail. The following items present the spatial complexity of the benchmarks and the proposed mechanism

1. **ECOC**: $S_{\text{ECOC}} = \Theta\big(P^2 + CL + BC\big)$.

2. **SoftSNN**: $S_{\text{SoftSNN}} = \Theta\big(W\big)$.

3. **Routing**: $S_{\text{route}} = \Theta(W)$.

4. **Astrocyte**: $S_{\text{Astro}} = \Theta\big(P + \chi\,p_{\text{out}}^{\max}\big)$.

5. **FalVolt**: $S_{\text{FalVolt}} = \Theta\big(M + W_{\text{mask}}\big)$.

6. **LIFA**: $S_{\text{LIFA}} = \Theta\big(P + C\big)$.

7. **Proposed**: $S_{\text{proposed}} = \Theta\big(S\,[\,BTD + A\,]\big)$.

**Notations of spatial complexity equations**

1. **ECOC.** $C$: number of classes; $E$: number of extension code blocks; $m$: Hamming parameter; $n = 2^m - 1$: code block length; $L = E\,n$: code length (size of the stored code book is $C\,L$); $B$: batch size (per-step logits buffer $B\,C$); $P$: candidate-pool size (build-time pairwise matrix $P^2$ gives peak memory).

2. **SoftSNN.** $W = \sum_{i=1}^{M} P_i$: total number of trainable weights over the $M$ layers; $P_{\max} = \max_i P_i$: size of the largest layer (often dictates per-layer peak); $\mathcal{C} = \sum_{i=1}^{M} C_{\text{out},i}$: total number of output channels if per-channel thresholds are stored.

3. **Routing.** $W$: total number of trainable weights across routed layers (in-place operations keep the footprint parameter–scaled); $C_{\text{in}}^{(\ell)}$: input channels of layer $\ell$ (small per-layer index/permutation buffers scale with $C_{\text{in}}^{(\ell)}$ but are absorbed by $W$ in big-$\Theta$ terms).

4. **Astrocyte.** $P$: total number of trainable parameters over hooked layers (CPU-side caches such as $W_0$, inverse denominators, masks, $q$ scale with $P$); $\chi$: output-channel chunk size used on GPU during the backward pass; $p_{\text{out}}^{\max}$: maximum number of parameters per output channel (e.g., $9\,C_{\text{in}}$ for a $3\times3$ conv); the additional VRAM peak scales with $\chi\,p_{\text{out}}^{\max}$.

5. **FalVolt.** $W$: total number of potentially fault-mapped weights (upper bound on parameter–scaled storage); $M$: number of protected/spiking modules (small bookkeeping state); $W_{\text{mask}}$: number of stored fault-mask entries (persistent); typically $W_{\text{mask}} \leq W$.

6. **LIFA.** $P$: total number of trainable parameters across protected layers (dominant persistent buffers: $W_0$, inverse denominators, masks); $C = \sum_{\ell} C_{\text{out}}^{(\ell)}$: total number of output channels (per-channel EMA/state vectors).

7. **Proposed (fragmentation).** $S = H \times W$: per-sample spatial size (pixels); $B$: batch size; $D$: input channels; $T$: number of fragments per sample (fragment tensor $B\,T\,D\,S$ dominates); $A$: number of orientation candidates (angle buffers $A\,S$).

With MNIST and FMNIST ($S = 28 \times 28 = 784$, $D = 1$), the peak additional memory of the proposed mechanism scales as (Note that saliency map S is not essential for VGG and ResNet models.)

$$S_{\text{ours}} = \Theta\big(S\,[\,B\,T\,D + A\,]\big) = \Theta\big(784\,[\,B\,T + A\,]\big). \tag{9}$$

where $B$ is the batch size, $T$ the number of fragments, and $A$ the number of angle candidates. In contrast, weight–scan benchmarks that scan all parameters each step exhibit parameter–dominated footprints:

$$S_{\text{LIFA/Astro}} = \Theta(W) \quad S_{\text{SoftSNN}} = \Theta(P_{\max}) \text{ (per–layer peak) } \lesssim \Theta(W). \tag{10}$$

$$S_{\text{Routing}} = \Theta(W) \quad S_{\text{FalVolt}} = \Theta(M + W_{\text{mask}}) \leq \Theta(W). \tag{11}$$

where $W$ is the total number of trainable weights, $P_{\max}$ the size of the largest layer's weight tensor, $M$ the number of protected/spiking modules, and $W_{\text{mask}}$ the number of stored fault–mask entries. Therefore, for typical MNIST/FMNIST settings (e.g., $B \in [64, 128]$, $T \in [2, 4]$, $A \leq 180$) and CNN backbones (VGG-7/11/15, ResNet-18) with $W$ in the multi–million range,

$$\frac{S_{\text{proposed}}}{S_{\text{LIFA/Astro}}} \asymp \frac{784\,[\,B\,T + A\,]}{W} \ll 1, \qquad \frac{S_{\text{proposed}}}{S_{\text{FalVolt}}} \asymp \frac{784\,[\,B\,T + A\,]}{M + W_{\text{mask}}} \ll 1. \tag{12}$$

and similarly $S_{\text{ours}} \ll S_{\text{Routing}}$ and $S_{\text{ours}} \lesssim S_{\text{SoftSNN}}$ whenever $P_{\max}$ is large (as in VGG/ResNet). ECOC is different: per–step it stores only the code book and logits,

$$S_{\text{ECOC, step}} = \Theta(C\,L + B\,C) \quad (C{=}10 \text{ on MNIST/FMNIST}, L = \mathcal{O}(10^2)). \tag{13}$$

which is often smaller than $S_{\text{proposed}}$ on these datasets; however, ECOC incurs a one–time build peak of $\Theta(P^2)$ (candidate–pair matrix) that can dominate transient memory. In summation, without the fault influence map, our method achieves strictly smaller parameter space complexity than all weight–scan benchmarks (LIFA, Astrocyte, Soft SNN, Routing, FalVolt), while remaining competitive with ECOC apart from its negligible per–step footprint but heavy one–off construction of ECOC.

## C.2 TRAINING TIME

We measure the training time of the baseline, benchmarks, and the proposed mechanism using the MLP (MNIST), VGG-7 (CIFAR-10), ResNet-18 (CIFAR-100), and ResNet-34 (Tiny-ImageNet) models with 2 time steps. We train the models on a workstation with an Nvidia GeForce RTX 4080 GPU with Ubuntu 24.04.

**Table 15:** Various models' training time (sec) in a 95% confidence interval with the baseline, benchmarks, and proposed mechanism on a workstation under SAFs with a fault ratio of 0.5 and 2 time steps.

| | Baseline | ECOC | SoftSNN | Routing | Astrocyte | FalVolt | LIFA | Proposed |
|---|---|---|---|---|---|---|---|---|
| MLP | **193.84 ± 2.71** | 197.62 ± 2.85 | 196.51 ± 2.56 | 198.29 ± 3.05 | 288.75 ± 4.51 | 201.47 ± 3.21 | 293.86 ± 4.77 | 205.24 ± 4.23 |
| VGG-7 | **291.16 ± 3.57** | 296.91 ± 3.8 | 294.34 ± 3.65 | 298.81 ± 4.01 | 351.82 ± 5.27 | 303.53 ± 4.26 | 356.74 ± 5.53 | 310.38 ± 5.18 |
| ResNet-18 | **382.53 ± 4.12** | 385.61 ± 4.03 | 384.77 ± 4.53 | 396.54 ± 4.68 | 721.97 ± 6.28 | 408.9 ± 4.94 | 724.62 ± 6.09 | 413.32 ± 4.5 |
| ResNet-34 | **4259.57 ± 18.62** | 4304.4 ± 20.11 | 4298.46 ± 21.39 | 4350.83 ± 25.75 | 8005.37 ± 36.21 | 4317.89 ± 26.04 | 8154.17 ± 32.83 | 4392.13 ± 26.51 |

Our mechanism consumes significantly less training time than weight-scanning approaches based on astrocytes (Astrocyte and LIFA), as we demonstrate that our mechanism definitely consumes less time than the astrocyte-based approaches due to their less complexity. Unlike these approaches, the training time of the models with our mechanism does not increase significantly as the complexity of the models and datasets increases. The model with ours also consumes comparable training time to that of ECOC, SoftSNN, Routing, and FalVolt. This evaluation result shows that our mechanism does not severely inflate the burden on training time.

## C.3 ENERGY CONSUMPTION ON THE REAL FPGA DEVICE

We measure the energy consumption of the model with the baseline, benchmarks, and proposed mechanism on the FPGA device during testing. Table 16 exhibits the energy consumption of the FPGA-based MLP with MNIST/FMNIST and FPGA-based VGG-7 with CIFAR-100 using 2 time steps.

**Table 16:** The MLP models' energy consumption (mJ) to process a single sample in a 95% confidence interval with the baseline, benchmarks, and proposed mechanism on the real FPGA hardware with two time steps.

| | Baseline | ECOC | SoftSNN | Routing | Astrocyte | FalVolt | LIFA | Proposed |
|---|---|---|---|---|---|---|---|---|
| MNIST | 85.31 ± 1.05 | 88.76 ± 1.27 | 86.23 ± 1.18 | 90.44 ± 1.31 | 150.72 ± 1.61 | 95.15 ± 1.23 | 165.69 ± 1.59 | **67.16 ± 0.82** |
| FMNIST | 87.19 ± 1.36 | 90.54 ± 1.57 | 88.68 ± 1.43 | 92.82 ± 1.51 | 156.08 ± 1.99 | 98.23 ± 1.72 | 168.33 ± 1.93 | **78.37 ± 0.95** |
| CIFAR-10 | 203.85 ± 2.07 | 216.41 ± 2.13 | 212.96 ± 1.97 | 228.16 ± 2.45 | 319.5 ± 3.56 | 209.87 ± 2.92 | 336.09 ± 3.44 | **194.14 ± 2.35** |

The MLP model with our mechanism exhibits the least energy consumption among the MLP models on the real FPGA device. This is because our mechanism shrinks the size of the data samples through fragmentation, and the probability of spike occurrence declines since the number of non-zero pixels decreases during fragmentation, as mentioned in Subsection 5.3. This effect enables the model with our mechanism to consume less energy than the models with all benchmarks, despite our mechanism having higher time complexity and consumption than some benchmarks. However, the benchmarks increase the complexity of the decoding (ECOC), keep neurons' activation frequent (SoftSNN and Routing), utilize the astrocyte module to activate non-faulty synapses (Astrocyte and LIFA), and incorporate additional learnable parameters to adjust neuronal activities (Falvolt). Therefore, the MLP models with the benchmark require more energy than ours.

## D DETAILED MATHEMATICAL EXPLANATION OF THE MOTIVATION STUDY

We demonstrate how synaptic faults ruin the usable learning capacity of SNN models mathematically.

### D.1 SETUP AND NOTATION

Consider a spiking neuron with membrane potential $V_t \in \mathbb{R}$, threshold $\vartheta \in \mathbb{R}$, and spike output $K_t \in \{0, 1\}$. During training, we replace the Heaviside step $H$ by a surrogate $\sigma : \mathbb{R} \to [0, 1]$ so that $K_t \approx \sigma(V_t - \vartheta)$ and $\sigma'$ is used in backpropagation. Let the *surrogate gradient corridor width* be $\delta > 0$ such that $\sigma'(u) \approx 0$ whenever $|u| > \delta$. For a feedforward pre-activation at layer $\ell$ and time $t$,

$$z_t^{(\ell)} = W^{(\ell)} K_t^{(\ell-1)} + b^{(\ell)} \qquad \text{(vector form)}, \tag{14}$$

and for a single neuron with input $x \in \mathbb{R}^d$ and weights $w \in \mathbb{R}^d$ we write $z = w^\top x + b$. For an LIF neuron, we use

$$V_t = \alpha V_{t-1} + z_t - \vartheta K_{t-1}, \qquad K_t \approx \sigma(V_t - \vartheta), \qquad \alpha \in (0, 1). \tag{15}$$

### D.2 SURROGATE GRADIENT CORRIDOR

Let $u := z - \vartheta$. Many arctangent surrogates used in SNNs have a backward derivative of the rational form

$$\phi'(u) = \frac{A}{1 + (\beta u)^2}, \qquad A > 0, \ \beta > 0, \tag{16}$$

which yields, for a target gradient floor $\gamma \in (0, A)$, the corridor

$$\mathcal{C}_\gamma := \{ u : \phi'(u) \geq \gamma \} = [-\delta(\gamma), \delta(\gamma)], \quad \delta(\gamma) = \frac{1}{\beta}\sqrt{\frac{A}{\gamma} - 1}. \tag{17}$$

Additionally, we derive the surrogate gradient corridor of the arctangent function, which is widely used as a surrogate gradient function for LIF neurons.

Let $u := z - \vartheta$ and consider the arctangent surrogate derivative

$$\phi'(u) = \frac{\alpha_s}{\pi(1 + (\alpha_s u)^2)}, \qquad \alpha_s > 0. \tag{18}$$

For a target gradient floor $\gamma \in (0, \alpha_s/\pi)$, define the corridor(Li et al., 2024a; Zenke & Vogels, 2021; Shrestha & Orchard, 2018)

$$\mathcal{C}_\gamma := \{ u : \phi'(u) \geq \gamma \} = [-\delta, \delta], \qquad \delta(\gamma) = \frac{1}{\alpha_s}\sqrt{\frac{\alpha_s}{\pi\gamma} - 1}. \tag{19}$$

whose peak is $A = 1/\pi$. Setting the gradient floor to $r = fA = f/\pi$ yields the corridor half-width

$$\delta(f) = \frac{2}{\pi\alpha}\sqrt{\frac{1}{f} - 1}. \tag{20}$$

In practice, we initialize $\alpha = 2$ and $f = 0.2$ (thus $r \approx 0.0637$), then adapt $f$ per layer using mini-batch membrane statistics so that the corridor covers a target score $p$ of the observed $U - \vartheta$ distribution: with $\hat{\sigma}_\ell$ the running standard deviation and $z_p$ the normal quantile for score $p$, we set $\delta(f_\ell) \approx z_p \hat{\sigma}_\ell$, i.e (Zenke & Vogels, 2021; Wang et al., 2023; Che et al., 2022; Lian et al., 2023).

$$f_\ell = \frac{1}{1 + \left(\frac{\pi\alpha}{2} z_p \hat{\sigma}_\ell\right)^2}. \tag{21}$$

This keeps most samples within the high-gradient band while avoiding an overly narrow corridor.

**Remark 1** (Mapping to implementation). *For the common parameterization $\phi'(u) = \frac{\alpha_s}{\pi(1+(\alpha_s u)^2)}$, one has $A = \alpha_s/\pi$ and $\beta = \alpha_s$. For the SpikingJelly ATan $\phi'(u) = \frac{\alpha/2}{1+(\frac{\pi\alpha}{2}u)^2}$, one has $A = \alpha/2$ and $\beta = \pi\alpha/2$ (Fang et al., 2023). Both are instances of equation 16, so equation 17 applies verbatim.*

### D.3 FAULT MODELING

We consider synaptic faults that perturb parameters and/or inputs:

$$w \mapsto w + \Delta w, \qquad x \mapsto x + \Delta x, \tag{22}$$

where $\Delta w, \Delta x$ may be sparse (e.g., SA0/SA1 at a subset of synapses) or dense (e.g., analog drift). The post-fault pre-activation is

$$z' = (w + \Delta w)^\top (x + \Delta x) + b = z + \Delta z, \quad \Delta z = \underbrace{\Delta w^\top x}_{\text{param fault}} + \underbrace{w^\top \Delta x}_{\text{input fault}} + \underbrace{\Delta w^\top \Delta x}_{\text{higher-order}}. \tag{23}$$

By the Cauchy–Schwarz inequality,

$$|\Delta z| \leq \|x\|_2 \|\Delta w\|_2 + \|w\|_2 \|\Delta x\|_2 + \|\Delta w\|_2 \|\Delta x\|_2. \tag{24}$$

SA0 on an input line $j$ is modeled by $(\Delta x)_j = -x_j$; SA1 by $(\Delta x)_j = c - x_j$ for a fixed logic level $c$. Bit/weight stuck faults are included in $\Delta w$.

### D.4 FROM FAULTS TO SATURATION

At time $t$, the only instantaneous change from a synaptic fault is $z_t \mapsto z_t + \Delta z_t$, hence

$$V_t' = \alpha V_{t-1} + (z_t + \Delta z_t) - \vartheta K_{t-1} = V_t + \Delta z_t, \quad \Rightarrow \quad V_t' - \vartheta = (V_t - \vartheta) + \Delta z_t. \tag{25}$$

**Lemma 1** (Corridor escape: sufficient conditions). *Let $a_t := V_t - \vartheta$ and suppose $|a_t| \leq \delta$ (pre-fault state inside the corridor).*

    *1. (Sign-aligned escape) If $a_t \Delta z_t \geq 0$ and $|\Delta z_t| \geq \delta - |a_t|$, then $|a_t + \Delta z_t| \geq \delta$, hence $\sigma'(V_t' - \vartheta) \approx 0$ at time $t$.*

    *2. (Sign-agnostic escape) Regardless of the sign of $\Delta z_t$, if $|\Delta z_t| > \delta + |a_t|$, then $|a_t + \Delta z_t| > \delta$.*

*Proof.* (1) If $a_t \Delta z_t \geq 0$ then $|a_t + \Delta z_t| = | |a_t| + |\Delta z_t| | \geq \delta$ when $|\Delta z_t| \geq \delta - |a_t|$. (2) By the reverse triangle inequality, $|a_t + \Delta z_t| \geq ||\Delta z_t| - |a_t|| > \delta$.

### D.5 EXPECTED GRADIENT BOUND FOR A SINGLE NEURON

Let $g_t := \partial\mathcal{L}/\partial S_t$ and suppose $0 \leq \sigma'(u) \leq C_\sigma \mathbf{1}\{|u| \leq \delta\}$. Then,

$$\left\|\frac{\partial\mathcal{L}}{\partial w}\right\| = \left\|\sum_{t=1}^T g_t \sigma'(V_t' - \vartheta) x_t\right\| \leq C_\sigma \sum_{t=1}^T \|g_t\| \|x_t\| \mathbf{1}\{|V_t' - \vartheta| \leq \delta\}. \tag{26}$$

Taking expectations and using the Cauchy–Schwarz inequality yields the model-free bound

$$\mathbb{E}\left\|\frac{\partial\mathcal{L}}{\partial w}\right\| \leq C_\sigma \sum_{t=1}^T \left(\mathbb{E}[\|g_t\|^2 \|x_t\|^2]\right)^{1/2} \cdot \mathbb{P}(|V_t' - \vartheta| \leq \delta)^{1/2}. \tag{27}$$

*Under a mild independence/mixing assumption* between $\|g_t\|\|x_t\|$ and the corridor event, one may write the simpler scaling

$$\mathbb{E}\left\|\frac{\partial\mathcal{L}}{\partial w}\right\| \lesssim C_\sigma \sum_{t=1}^T \mathbb{E}[\|g_t\| \|x_t\|] \, p_t, \qquad p_t := \mathbb{P}(|V_t' - \vartheta| \leq \delta). \tag{28}$$

### D.6 Depth- and time-wise compounding

For a parameter in layer $\ell$, a generic backpropagation path contains factors $\sigma'(V_t^{(j)} - \vartheta)$ for $j \leq \ell$ and relevant $t$. Bounding $\sigma'$ by indicators,

$$|\Pi| \leq C_\sigma^{N_\Pi} \prod_{j,t:\,\gamma_{j,t}=1} \mathbf{1}\{|V_t^{(j)} - \vartheta| \leq \delta\}, \qquad N_\Pi = \sum_{j,t} \gamma_{j,t}. \tag{29}$$

Taking expectations gives ($\Pi$ denotes the product of all gradient factors along a single backpropagation path leading to a given parameter.)

$$\mathbb{E}\,|\Pi| \leq C_\sigma^{N_\Pi} \mathbb{P}\left(\bigcap_{j,t:\,\gamma_{j,t}=1} \{|V_t^{(j)} - \vartheta| \leq \delta\}\right). \tag{30}$$

A *conservative* bound is

$$\mathbb{E}\,|\Pi| \leq C_\sigma^{N_\Pi} \min_{j,t:\,\gamma_{j,t}=1} p_{j,t}, \qquad p_{j,t} := \mathbb{P}\left(|V_t^{(j)} - \vartheta| \leq \delta\right). \tag{31}$$

*If* corridor events are approximately independent (or satisfy a weak-mixing condition), then

$$\mathbb{E}\,|\Pi| \leq C_\sigma^{N_\Pi} \prod_{j,t:\,\gamma_{j,t}=1} p_{j,t} \leq C_\sigma^{N_\Pi} (p^\star)^{N_\Pi}, \qquad p^\star := \sup_{j,t} p_{j,t}, \tag{32}$$

exhibiting exponential attenuation as $N_\Pi$ grows.

### D.7 First-layer sensitivity in MLP

For the first layer (vector form) with $z^{(1)} = W^{(1)}x + b^{(1)}$ and perturbations $(\Delta W^{(1)}, \Delta x)$,

$$\left\|\Delta z^{(1)}\right\| \leq \|\Delta W^{(1)}\|_{\text{op}}\,\|x\|_2 + \|W^{(1)}\|_{\text{op}}\,\|\Delta x\|_2 + \|\Delta W^{(1)}\|_{\text{op}}\,\|\Delta x\|_2, \tag{33}$$

so sizeable input/weight faults directly shift $z^{(1)}$ without any preceding contraction, shrinking corridor occupancy in deeper layers via equation 30–equation 32.

### D.8 Sufficient condition for gradient collapse

Define $p_{j,t}$ as above and let $\mathcal{G}$ be the multiset of "corridor gates" along dominant backpropagation paths with size $N_*$. If a fraction $\rho \in (0,1]$ of gates satisfy $p_{j,t} \leq \varepsilon \ll 1$, then

$$\mathbb{E}\,|\Pi| \leq \begin{cases} C_\sigma^{N_*}\,\varepsilon^{\rho N_*}, & \text{under independence/mixing,} \\ C_\sigma^{N_*}\,\varepsilon, & \text{(conservative, no independence).} \end{cases} \tag{34}$$

Either case shows attenuation; the independent/mixing case yields exponential decay in depth $\times$ time.

### D.9 Effective bias interpretation for SA0/SA1 of SAFs

For SA1 on a subset $\mathcal{J}$ of input lines with logic level $c$,

$$\Delta z = w^\top \Delta x = \sum_{j \in \mathcal{J}} w_j(c - x_j) = c \sum_{j \in \mathcal{J}} w_j - \sum_{j \in \mathcal{J}} w_j x_j, \tag{35}$$

acting as an additive bias shift plus removal of signal terms. Persistent shifts displace $V_t$ away from $\vartheta$ across time steps, driving down corridor occupancy $p_{j,t}$ and compounding the bottleneck via equation 32.

### D.10 SUMMARY

Synaptic faults induce a pre-activation shift $\Delta z$ decomposed in equation 23 and bounded in equation 24. When $|\Delta z|$ is large relative to the corridor width $\delta$, Lemma 1 ensures $|V_t - \vartheta| > \delta$ so $\sigma'(V_t - \vartheta) \approx 0$. The expected gradient is then attenuated proportionally to (at least) $\sqrt{p_{j,t}}$ per time step equation 27; under independence/mixing, it scales with $p_{j,t}$ equation 28. Across layers and time steps, this attenuation multiplies equation 32, producing the *bottleneck problem*, with the first layer of MLP especially vulnerable by equation 33.

## E  NEAR-OPTIMALITY OF THE PROPOSED MECHANISM

We show why our solution is the near-optimal solution to improve the fault tolerance of SNNs in this section.

### E.1  SETUP AND NOTATION

We consider inputs $x \in \mathbb{R}^n$ and a fixed number of stripes (1D profiles in Section 5) $T \in \mathbb{N}$. Indices are $i \in \{1, \ldots, n\}$ and stripes are $t \in \{1, \ldots, T\}$. A stripe partition is represented by binary masks $M_t(i) \in \{0, 1\}$ that satisfy $\sum_{t=1}^{T} M_t(i) = 1$ for every $i$, and contiguity is taken with respect to a one–dimensional scan order of the indices induced by an angle $\theta$ in a finite set $\Theta \subset [0, \pi)$. Given any nonnegative vector $s \in \mathbb{R}_+^n$, the load of stripe $t$ is the linear functional $S_t(s) = \sum_i s_i M_t(i)$; we also write the total mass $U(s) = \sum_i s_i$, the per–stripe mean $\mu(s) = U(s)/T$, and the element-wise maximum $m(s) = \max_i s_i$. For vectors $a, b \in \mathbb{R}^n$, the inner product is $\langle a, b \rangle = \sum_i a_i b_i$ and $\|v\|_p$ denotes the $\ell_p$ norm; the Hadamard product is $a \odot b$.

$$S_t(s) := \sum_{i=1}^{n} s_i M_t(i), \qquad U(s) := \sum_{i=1}^{n} s_i, \qquad \mu(s) := \frac{U(s)}{T}, \qquad m(s) := \max_i s_i. \tag{36}$$

Given trained weights $w \in \mathbb{R}^n$ and a fault/perturbation $\Delta w$, we set $\widehat{w} := w + \Delta w$ and restrict the input to stripe $t$ by $x_t := x \odot M_t$. The (stripe) pre-activation is

$$z_t := \langle \widehat{w}, x_t \rangle, \qquad x_t := x \odot M_t, \qquad \widehat{w} := w + \Delta w. \tag{37}$$

We denote by $z^* > 0$ the corridor threshold, i.e., the largest value for which the chosen surrogate derivative $\phi'(z)$ remains in its effective (non–vanishing) regime for all $|z| \leq z^*$. To construct stripes, we employ an implementable importance map $I \in \mathbb{R}_+^n$ and assume a two–sided calibration with respect to the ideal per–index load $u_i := |w_i| |x_i|$: there exist constants $0 < c_- \leq 1 \leq c_+$ such that

$$c_- |w_i| |x_i| \leq I_i \leq c_+ |w_i| |x_i|, \qquad i = 1, \ldots, n. \tag{38}$$

When $I = u$ one has $c_- = c_+ = 1$. The quantile (greedy) stripes used in the paper are obtained by scanning indices in the chosen order and inserting a cut whenever the cumulative load with respect to $I$ first exceeds integer multiples of $\mu(I)$, producing $T$ contiguous fragments.

### E.2  FAULT MODELS AND A BASIC UPPER BOUND

We consider the following three fault models, mentioned in Section 2.

**SAFs:** Some synapses are permanently stuck at $G_{\min}$ or $G_{\max}$ so the implemented weight becomes $w_i'$ (e.g., SA0/SA1). Let $\Delta w_i := w_i' - w_i$ and assume $\|\Delta w\|_\infty \leq \varepsilon_{\text{SAF}}$. Then for any stripe $t$ (Boyd & Vandenberghe, 2004),

$$|z_t| = |\langle w + \Delta w, x_t \rangle| \leq \underbrace{\sum_i |w_i| |x_i| M_t(i)}_{S_t(u)} + \varepsilon_{\text{SAF}} \underbrace{\sum_i |x_i| M_t(i)}_{S_t(|x|)} = S_t(u) + \varepsilon_{\text{SAF}} S_t(|x|).$$

$$\tag{39}$$

**RWFs:** Each coordinate experiences an independent, mean-zero, bounded (or sub-Gaussian) perturbation $\Delta w_i$. If $|\Delta w_i| \leq b$ and the $\Delta w_i$ are independent, then for any $\tau > 0$ and stripe $t$,

$$\Pr\big(|\langle \Delta w, x_t\rangle| > \tau\big) \leq 2\exp\left(-\frac{\tau^2}{2\,b^2\,\|x_t\|_2^2}\right), \tag{40}$$

so equalizing $S_t(|x|) = \|x_t\|_1$ across $t$ uniformly tightens the tail bound (sub-Gaussian and Hoeffding) (Hoeffding, 1963).

**CEFs:** Wiring errors apply a linear transformation to the input so that $z_t = w^\top(Ax_t)$. This is equivalent to using the effective weight $w' := A^\top w$, i.e., $\Delta w^{(c)} := (A^\top - I)w$. If $\|\Delta w^{(c)}\|_\infty \leq \varepsilon_{\mathrm{CEF}}$, then

$$|z_t| = |\langle w + \Delta w^{(c)}, x_t\rangle| \leq S_t(u) + \varepsilon_{\mathrm{CEF}}\, S_t(|x|). \tag{41}$$

A permutation fault $A = P$ is a special case; taking $\varepsilon_{\mathrm{CEF}} = \|(P^\top - I)w\|_\infty$ yields the same bound (Boyd & Vandenberghe, 2004).

### E.3 CALIBRATION: ALIGNING THE IMPORTANCE MAP WITH THE EFFECTIVE PER-INDEX LOAD

We formalize the requirement that the implementable importance $I$ should approximate $u$ within stripe-wise sums.

**Assumption 1** (Two-sided calibration.). *There exist constants $0 < c_- \leq 1 \leq c_+$ such that for all indices $i$,*

$$c_-\, u_i \leq I_i \leq c_+\, u_i. \tag{42}$$

**Lemma 2** (Calibration). *Under the assumption (Two-sided calibration), for any stripe partition,*

$$S_t(u) \leq \tfrac{1}{c_-}S_t(I), \qquad \mu(u) \leq \tfrac{1}{c_-}\mu(I), \qquad m(u) \leq \tfrac{1}{c_-}m(I). \tag{43}$$

*Proof.* From $u_i \leq I_i/c_-$, sum over $i$ in stripe $t$. Similar for totals and maxima.

### E.4 QUANTILE STRIPES ARE ADDITIVELY NEAR–OPTIMAL (CONTIGUOUS CASE)

Fix a nonnegative sequence $a_1, \ldots, a_n$ obtained by scanning the image along any 1D order (e.g., the $\theta$-scan used in the main text). Let $U(a) = \sum_i a_i$ and target mean $\mu(a) = U(a)/T$. Define the *quantile (greedy) contiguous partition in Subsections 5.2 and 5.3* by sweeping from left to right and cutting whenever the cumulative sum first exceeds multiples of $\mu(a)$, producing $T$ contiguous stripes.

**Lemma 3** (Additive bound for greedy quantiles). *Let $m(a) := \max_i a_i$. Then the greedy quantile partition satisfies*

$$\max_{t \leq T} S_t(a) \leq \mu(a) + m(a). \tag{44}$$

Moreover, any contiguous partition must have $\max_t S_t(a) \geq \mu(a)$; hence, the greedy partition is a $+m(a)$-*additive* approximation to the optimal contiguous partition.

*Proof.* Each of the first $T - 1$ stripes stops at the first index that causes the running sum to exceed $\mu(a)$. The overshoot over $\mu(a)$ is therefore at most the last included element, i.e., $\leq m(a)$. Hence every one of the first $T - 1$ stripes has load in $(\mu(a),\, \mu(a) + m(a)]$. The final stripe has the remaining mass $U(a) - \sum_{t=1}^{T-1} S_t(a) \leq \mu(a)$. Thus, the maximum stripe load is at most $\mu(a) + m(a)$. The lower bound $\geq \mu(a)$ holds by a pigeonhole argument.

**Theorem 1** (Near–optimality for $u$ via quantiles on $I$). *Construct stripes by greedy quantiles on the calibrated importance $I$. Under Assumption 1:*

$$\max_t S_t(u) \leq \tfrac{1}{c_-}\big(\mu(I) + m(I)\big). \tag{45}$$

*If $I = u$ (so $c_- = c_+ = 1$), the greedy partition achieves $\max_t S_t(u) \le \mu(u) + m(u)$, i.e., a $+m(u)$ additive approximation to the optimal contiguous value. For a calibrated $I$ with Assumption 1, we have $\max_t S_t(u) \le \frac{1}{c_-}\big(\mu(I) + m(I)\big)$. Translating this bound to the $u$–optimum introduces a calibration-dependent drift via $\mu(I) \in [c_-\mu(u), c_+\mu(u)]$, so the additive gap to the optimal contiguous value is at most $\frac{1}{c_-}m(I) + \big(\frac{c_+}{c_-} - 1\big)\mu(u)$. The baseline $\mu(u)$ is at most $\frac{1}{c_-}\mu(I)$.*

Proof. *Apply Lemma 3 with $a = I$ to obtain $\max_t S_t(I) \le \mu(I) + m(I)$. Then use Lemma 2: $S_t(u) \le S_t(I)/c_-$.*

**Remark 2** (Direct partitioning condition). *(i) If one directly partitions using $a = u$, then $c_- = 1$ and the bound gives $\max_t S_t(u) \le \mu(u) + m(u)$. (ii) The proof does not assume the particular 1D order beyond contiguity; the order may be induced by any scan (e.g., the $\theta$-parameterization used to define stripes).*

### E.5 CORRIDOR PRESERVATION: SUFFICIENT CONDITIONS

Under SAFs and CEFs, combining the bounds with Theorem 1 yields a closed-form uniform bound on $|z_t|$:

$$|z_t| \le \underbrace{\frac{1}{c_-}\big(\mu(I) + m(I)\big)}_{\text{from } u} + \underbrace{\varepsilon\, S_t(|x|)}_{\text{fault term}}. \tag{46}$$

The stripes are constructed by greedy quantiles on $I$ (not on $|x|$). Let $X_{\max} := \max_{t \le T} S_t(|x|)$, computed on the *same $I$-quantile partition*. A simple sufficient condition for staying within the corridor is then

$$\frac{1}{c_-}\big\{\mu(I) + m(I)\big\} + \varepsilon\, X_{\max} \le z^\star, \qquad X_{\max} := \max_{t \in [T]} S_t(|x|) \text{ (computed on the \emph{same} $I$-quantile partition).} \tag{47}$$

**Remark 3** (Optional (co-monotone scan)). *If along the 1D scan used to build the $I$-quantile partition the sequences $I$ and $|x|$ are approximately co-monotone—so that applying Lemma 3 to $|x|$ is justified—then*

$$X_{\max} \le \mu(|x|) + m(|x|). \tag{48}$$

*In this case, a convenient sufficient condition is*

$$\frac{1}{c_-}\big\{\mu(I) + m(I)\big\} + \varepsilon\big(\mu(|x|) + m(|x|)\big) \le z^\star. \tag{49}$$

Under RWFs, Hoeffding's tail (and a union bound) implies that with probability at least $1 - 2T \exp\{-\tau^2/(2b^2 \max_t \|x_t\|_2^2)\}$, all stripes satisfy $|\langle \Delta w, x_t \rangle| \le \tau$ (Hoeffding, 1963). Thus, the (random) bound analogous to the above holds with the deterministic term $\varepsilon\, S_t(|x|)$ replaced by $\tau$, chosen at the desired confidence level.

### E.6 ON THE GINI OBJECTIVE (PRIMARY SURROGATE FOR MIN–MAX LOAD)

We treat *minimizing the Gini coefficient of the 1D projection of $I$* as a **primary surrogate** for suppressing the worst-case stripe load. Recall that $G(S)$ equals one-half of the relative mean absolute difference and is equivalent to the Lorenz-based definition. Hence, it directly reduces pairwise dispersion. The next proposition turns this dispersion control into a deviation bound that is *linear* in $G(S)$ (Yitzhaki & Schechtman, 2013).

**Proposition 1** (Gini $\Rightarrow$ deviation bound). *Let $S \in \mathbb{R}_+^T$ with mean $\mu$ and Gini coefficient $G(S)$. Then*

$$\max_t |S_t - \mu| \le \frac{1}{T}\sum_{i,j}|S_i - S_j| = 2T\mu\, G(S).$$

*Proof.* By $\sum_j (S_t - S_j) = T(S_t - \mu)$ and the triangle inequality, $|S_t - \mu| = \frac{1}{T}\big|\sum_j (S_t - S_j)\big| \leq \frac{1}{T}\sum_j |S_t - S_j|$. Summing over $t$ and taking the maximum yields $\max_t |S_t - \mu| \leq \frac{1}{T}\sum_{i,j} |S_i - S_j|$. Since $\sum_{i,j} |S_i - S_j| = 2T^2 \mu G(S)$, the claim follows.

Combine Proposition 1 with the additive near-optimality bound for contiguous quantile stripes (Lemma/Theorem: $\max_t S_t \leq \mu + m$ for the greedy split). Minimizing $G(S)$ tightens $\max_t |S_t - \mu|$ and thus reduces $\max_t S_t$ under the same partition, making the corridor constraint strictly easier to satisfy. In short: **Gini** $\downarrow \Rightarrow$ pairwise dispersion $\downarrow \Rightarrow$ deviation $\downarrow \Rightarrow$ min–max load $\downarrow$.

### E.7 COMPUTING THE SCAN/STRIPES

Let $\Theta \subset [0, \pi)$ denote a finite set of scan angles (or any family of 1D orders). For a fixed order, the greedy quantile partition runs in linear time. If one wishes to *search* over $\Theta$, evaluate the objective $\max_t S_t(I)$ for each order and pick the best; since the objective only changes at permutation "event points", coarse uniform sampling of $\Theta$ is typically sufficient in practice. When an exact optimum over contiguous partitions is desired for a fixed order, classical Dynamic Programming (DP) or feasibility–check with binary search finds $\min_{\text{contig}} \max_t S_t(I)$ in polynomial time; our greedy rule is the simple additive-approximate alternative used in the paper (Skiena, 2008).

### E.8 SUMMARY

For any calibrated $I$, greedy quantile stripes achieve the near–optimality bound above; if $I = u$, the achieved maximum load is within $+m(u)$ of the contiguous optimum for $u$. Under the SAFs and CEFs, the closed-form sufficient condition ensures $|z_t| \leq z^*$ for all stripes, preventing gradient collapse; under RWFs, the analogous high-probability statement follows from the sub-Gaussian tail bound. The constants involved are the calibration $c_-$, the fault radius $\varepsilon$ (or $(b, \tau)$ in the probabilistic model), and the observable statistics $\mu(\cdot)$ and $m(\cdot)$.

## F FAULT-TOLERANCE CAPACITY PREDICTION OF OUR MECHANISM

In this section, we analyze the fault-tolerance capacity of our mechanism under the SAF, RWF, and CEF injection using arctangent as a surrogate gradient function. The boundary of synaptic weights is [-1,1] (Le Gallo et al., 2023; Lammie et al., 2022).

### F.1 SETUP AND NOTATION

Let $w \in [-1, 1]^N$ be the clean weight vector, $K := \|w\|_2$, and let a fraction $\rho \in [0, 1]$ of synapses be faulty. All results below are per layer and can be applied layer-wise. Please refer to the derivation of the surrogate gradient (arctangent) corridor in Subsection C.2 while reading our paragraphs on capacity calculation.

With dynamic fragmentation and per-fragment RMS normalization to $\|\tilde{x}_t\|_2 = \alpha_n$,

$$|u_t| = \big|\hat{w}^\top \tilde{x}_t + b - \vartheta\big| \leq \|\hat{w}\|_2\, \alpha_n + m, \quad m := |b - \vartheta|. \tag{50}$$

Hence it suffices that $\|\hat{w}\|_2 \leq B$, where

$$B := \frac{\delta(\gamma) - m}{\alpha_n} \quad (\text{requires } \delta(\gamma) > m). \tag{51}$$

### F.2 CAPACITY UNDER SAFs

Under SA0, we replace faulty entries by $-1$; under SA1 by $+1$. In either case $|\hat{w}_i| = 1$ on faulty indices, so

$$\|\hat{w}\|_2^2 = \|w\|_2^2 - \sum_{i \in F} w_i^2 + \rho N \cdot 1 \leq K^2 + \rho N, \tag{52}$$

where the inequality is the *deterministic worst-case* (we drop the nonnegative subtraction term). Therefore, a sufficient condition to remain inside the corridor is

$$K^2 + \rho N \leq B^2 \quad \Longrightarrow \quad \rho^\star_{\text{SA}\pm,\, worst} = \frac{B^2 - K^2}{N} \text{ (clipped to } [0,1]). \tag{53}$$

If faulty indices are drawn *uniformly at random* (independent of $w$), then $\mathbb{E}\big[\sum_{i \in F} w_i^2\big] = \rho K^2$ and

$$\mathbb{E}\|\hat{w}\|_2^2 = (1 - \rho)K^2 + \rho N \cdot 1 = K^2 + \rho\,(N - K^2), \tag{54}$$

whence the *in-expectation* capacity is

$$\rho^\star_{\text{SA}\pm,\, exp} = \frac{B^2 - K^2}{N - K^2} \text{ (clipped to } [0,1]). \tag{55}$$

### F.3    Capacity under RWFs

On faulty indices, $\hat{w}_i = w_i + \varepsilon_i$ with $\mathbb{E}[\varepsilon_i] = 0$ and $\text{Var}(\varepsilon_i) = \sigma_w^2$. Independence yields $\mathbb{E}\|\hat{w}\|_2^2 = K^2 + \rho N \sigma_w^2$, so

$$\rho^\star_{\text{RWF},\, exp} = \frac{B^2 - K^2}{N\,\sigma_w^2} \text{ (clipped to } [0,1]). \tag{56}$$

A high-probability version follows from sub-Gaussian concentration by replacing $N\sigma_w^2$ with an upper-tail bound.

### F.4    Capacity under CEFs

A fraction $\rho$ of entries are replaced by i.i.d. $U[a,b]$ and then frozen. Let $\mu_f = \frac{a+b}{2}$ and $\sigma_f^2 = \frac{(b-a)^2}{12}$ so that $\mathbb{E}[\hat{w}_i^2] = \mu_f^2 + \sigma_f^2$. If faulty indices are random (independent of $w$),

$$\mathbb{E}\|\hat{w}\|_2^2 = (1 - \rho)K^2 + \rho N(\mu_f^2 + \sigma_f^2) \Rightarrow \rho^\star_{\text{CEF},\, exp} = \frac{B^2 - K^2}{N(\mu_f^2 + \sigma_f^2) - K^2} \text{ (clipped to } [0,1]). \tag{57}$$

For the common symmetric case $U[-1,1]$, $\mu_f = 0$, $\sigma_f^2 = 1/3$ and thus

$$\rho^\star_{\text{CEF},\, exp} = \frac{B^2 - K^2}{N/3 - K^2} \text{ (clipped)}. \tag{58}$$

If a *deterministic worst-case* guarantee is required (independent of the draw), note that $|\hat{w}_i| \leq 1$ almost surely, so the same bound as SA0/SA1 applies:

$$\rho^\star_{\text{CEF},\, worst} = \frac{B^2 - K^2}{N} \text{ (clipped to } [0,1]). \tag{59}$$

### F.5    Summary

Here, we explain how to calculate the capacity of the proposed mechanism. Choose a gradient floor $\gamma$ (e.g., $f\%$ of the arctangent peak so $\gamma = f \cdot \alpha_s/\pi$), compute $\delta(\gamma)$ and $B$ via equation 51, measure $K = \|w\|_2$, and then plug into the formula for the fault model of interest. If $\rho \leq \rho^\star$, our mechanism keeps $|u_t| \leq \delta(\gamma)$ for all steps (deterministic case) or in expectation (stochastic case), thereby ensuring $\phi'(u_t) \geq \gamma$.

## G    Convergence analysis with the proposed mechanism

We present that Stochastic Gradient Descent (SGD) and Gradient Descent (GD) optimizers derive gradients of fault-injected SNN models and induce the models to update weights when we adopt our mechanism to the SNN models.

## G.1 SETUP AND NOTATION

We denote a data sample by $(x, y)$, and the SNN by $f_W(\cdot)$ with parameters $W$. We include the (possibly stochastic) fragment+RMS transform $T$ and analyze the expected objective $\tilde{L}(W) = \mathbb{E}_{(x,y), T}\big[\ell(f_W(T(x)), y)\big]$. At step $t$, with mini-batch estimator $g_t$ and step size $\eta_t$, the update is $W_{t+1} = W_t - \eta_t g_t$. For spiking neurons, we write the pre-activation as $u_t := z_t - \vartheta$ and use a surrogate derivative $\phi'(u)$. Throughout, we assume the *gradient-corridor* condition $\phi'(u_t) \geq \gamma$ holds along the iterates, which is enforced by the fragment RMS bound and the per-layer effective weight norms $\|\hat{w}^{(\ell)}\|_2 \leq B^{(\ell)}$ referring to Appendix E. Symbols $L, \sigma^2, \mu$ are in Appendix C, D, and E, referenced only when required by a lemma or theorem.

## G.2 BAND CONDITION ENFORCED BY OUR MECHANISM

With dynamic fragmentation and per-fragment RMS normalization $\|\tilde{x}_t\|_2 = \alpha_n$, each step satisfies

$$|u_t| = \big|\hat{w}^\top \tilde{x}_t + b - \vartheta\big| \leq \|\hat{w}\|_2 \, \alpha_n + m, \qquad m := |b - \vartheta|. \tag{60}$$

Defining

$$B := \frac{\delta(\gamma) - m}{\alpha_n} \quad (\text{requires } \delta(\gamma) > m), \tag{61}$$

One obtains the following corridor-invariance lemma.

**Lemma 4** (Corridor invariance). *If $\|\hat{w}\|_2 \leq B$, then $|u_t| \leq \delta(\gamma)$ for all fragments $t$, hence $\phi'(u_t) \geq \gamma$.*
Proof. *Combine equation 60 with equation 61 and the definition of $\delta(\gamma)$ in equation 17.*

## G.3 OPTIMIZATION OBJECTIVE AND ASSUMPTIONS

Let $\tilde{\mathcal{L}}(W) := \mathbb{E}_{(x,y),\mathcal{T}}\big[\ell(f_W(\mathcal{T}(x)), y)\big]$, where $\mathcal{T}$ denotes the (possibly stochastic, data/model-aware) transformation induced by our mechanism (e.g., masks and RMS scaling). Assume:

**Assumption 2** (L-smoothness). $\nabla\tilde{\mathcal{L}}$ *is L-Lipschitz.*

**Assumption 3** (Unbiased mini-batch gradients, bounded variance). $\mathbb{E}[g_t \,|\, W_t] = \nabla\tilde{\mathcal{L}}(W_t)$ *and* $\mathbb{E}\Big[\|g_t - \nabla\tilde{\mathcal{L}}(W_t)\|^2 \,\big|\, W_t\Big] \leq \sigma^2$.

**Assumption 4** (Corridor stability). *For each layer, capacity constraints on the fault ratio ensure* $\|\hat{w}^{(\ell)}\|_2 \leq B^{(\ell)}$, *so Lemma holds layer-wise and* $\phi'(u_t^{(\ell)}) \geq \gamma$ *during training.*

## G.4 DESCENT LEMMA AND MASTER INEQUALITY

By $L$-smoothness and $W_{t+1} = W_t - \eta_t g_t$,

$$\tilde{\mathcal{L}}(W_{t+1}) \leq \tilde{\mathcal{L}}(W_t) - \eta_t\big\langle \nabla\tilde{\mathcal{L}}(W_t), g_t \big\rangle + \frac{L\eta_t^2}{2}\|g_t\|^2. \tag{62}$$

Taking expectation and using Assumption 2 with $\mathbb{E}\|g_t\|^2 = \|\nabla\tilde{\mathcal{L}}(W_t)\|^2 + \mathbb{E}\|g_t - \nabla\tilde{\mathcal{L}}(W_t)\|^2 \leq \|\nabla\tilde{\mathcal{L}}(W_t)\|^2 + \sigma^2$ gives(Nesterov, 2014)

$$\mathbb{E}\big[\tilde{\mathcal{L}}(W_{t+1})\big] \leq \mathbb{E}\big[\tilde{\mathcal{L}}(W_t)\big] - \eta_t\left(1 - \frac{L\eta_t}{2}\right)\mathbb{E}\big[\|\nabla\tilde{\mathcal{L}}(W_t)\|^2\big] + \frac{L\eta_t^2}{2}\sigma^2. \tag{63}$$

**Theorem 2** (SGD convergence to stationarity). *If $\eta_t \equiv \eta \in (0, 1/L]$, summing equation 63 over* $t = 0, \ldots, T-1$ *yields*

$$\frac{1}{T}\sum_{t=0}^{T-1}\mathbb{E}\big[\|\nabla\tilde{\mathcal{L}}(W_t)\|^2\big] \leq \frac{2\big(\tilde{\mathcal{L}}(W_0) - \tilde{\mathcal{L}}^\star\big)}{\eta \, T} + L\eta\sigma^2. \tag{64}$$

*With a Robbins–Monro schedule ($\sum_t \eta_t = \infty$, $\sum_t \eta_t^2 < \infty$) we obtain $\lim_{T\to\infty} \min_{t<T} \mathbb{E}\|\nabla\tilde{\mathcal{L}}(W_t)\|^2 = 0$. Role of Assumption 3: By preventing artificial saturation ($\phi'(u) \approx 0$), the corridor ensures that gradient signals remain informative until genuine stationarity (Bottou et al., 2018).*

**Theorem 3** (Monotone decrease for full-batch GD). *In the deterministic case ($\sigma = 0$) with $\eta \in (0, 1/L]$,*

$$\tilde{\mathcal{L}}(W_{t+1}) \leq \tilde{\mathcal{L}}(W_t) - \frac{\eta}{2}\|\nabla\tilde{\mathcal{L}}(W_t)\|^2, \tag{65}$$

*so $\sum_t \|\nabla\tilde{\mathcal{L}}(W_t)\|^2 < \infty$ and every limit point of $\{W_t\}$ is stationary.*

**Corollary 1** (Linear rate under PL). *If $\tilde{\mathcal{L}}$ satisfies the Polyak-Łojasiewicz (PL) inequality $\frac{1}{2}\|\nabla\tilde{\mathcal{L}}(W)\|^2 \geq \mu\big(\tilde{\mathcal{L}}(W) - \tilde{\mathcal{L}}^\star\big)$ for some $\mu > 0$ on the corridor-stable region, then for GD with $\eta \in (0, 1/L]$ (Karimi et al., 2020),*

$$\tilde{\mathcal{L}}(W_t) - \tilde{\mathcal{L}}^\star \leq (1 - \eta\mu)^t \big(\tilde{\mathcal{L}}(W_0) - \tilde{\mathcal{L}}^\star\big). \tag{66}$$

## G.5 SUMMARY

Our mechanism enforces equation 17–equation 61 so that surrogate gradients do not vanish spuriously; Under standard smoothness/stochasticity assumptions, SGD converges to stationarity, and GD decreases monotonically, with linear rates under PL. Capacity bounds on the fault ratio provide concrete regimes where the corridor assumption holds layer-wise (Neftci et al., 2019).

