# OpenReview forum: "Practical Mechanism via Simple Input Control for Fault-Tolerant Spiking Neural Networks"
_ICLR.cc/2026/Conference — Submitted to ICLR 2026_

### Official Review · Reviewer_EvpD · 2025-10-30

**Soundness:** 2
**Presentation:** 1
**Contribution:** 2
**Rating:** 2
**Confidence:** 3

**Summary:**

This paper proposed an input control mechanism to improve the fault tolerance of SNNs. It is claimed that the proposed method is also beneficial to SNNs implemented on FPGA devices.

**Strengths:**

1. The motivation study section (section 4) contains an analysis of different aspects.

**Weaknesses:**

1. The paper should clearly establish at the outset that the research focuses on hardware-implemented SNNs and their fault tolerance.

2. The section title “3.2 Mechanisms to Improve the Fault Tolerance of SNNs” is misleading, as SNNs themselves do not suffer the described faults and therefore do not require such fault tolerance. Moreover, the section discusses neuromorphic hardware fault-tolerance research, which does not fit well under this section title.

3. Section 5: The research appears to focus on on-chip SNN learning. This should be stated explicitly earlier in the paper, before detailing the methods.

4. Section 5: It is unclear how the three subsections work together to form the proposed mechanism. A brief summary at the beginning of this section would be helpful.

5. Abstract and Introduction: FPGA is mentioned only four times in the main text, without explaining how the method works on FPGAs or what benefits the proposed method provides to FPGA. Since the main paper does not include any FPGA-relevant analysis, it should not be presented as a primary contribution in the Abstract as well as the Introduction.

**Questions:**

See weakness for questions.

---

> ### Author Response · Authors · 2025-11-13
> **Rebuttal for Reviewer EvpD**
>
> # Thanks to Reviewer EvpD
>
> We appreciate your comments on the presentation quality of our paper. We have significantly improved our paper's clarity of content by addressing your comments.
>
> ---
>
> # Responses to the weaknesses
>
> **W1**: We really appreciate your valuable comment on the clarity of our paper. As you mentioned, our paper focuses on hardware faults of neuromorphic devices based on SNNs. To make this point clear, we have emphasized that our paper focuses on the fault tolerance of hardware-implemented SNNs in Abstract and Section 1 (Introduction). Please refer to Abstract and Section 1 of the revised manuscript.
>
> **W2**: We really appreciate your valuable comment on the name of Subsection 3.2. We agree with your opinion that neuromorphic devices suffer from hardware faults, not SNNs. To address your comment, we have changed the name of Subsection 3.2 to “Mechanisms to improve fault tolerance of SNN-based neuromorphic devices”. Furthermore, we have changed the contents in Subsection 3.2, which explains that faults occur in SNNs, to the contents explaining that faults occur in neuromorphic devices. Please refer to the name and contents of Subsection 3.2 in the revised manuscript.
>
> **W3**: We really appreciate your valuable comment on the order of contents in our paper. To address your concern, we have mentioned that our mechanism targets the hardware-implemented SNNs in Abstract and Section 1 of the revised paper. Furthermore, we have also written that the proposed mechanism concentrates on hardware-based SNNs in Section 5 of the revised paper.
>
> **W4**: We really appreciate your valuable comment on the relationship between the three parts of the proposed mechanism. The three parts in our mechanism cooperate with the following steps. First, the sensitivity-score module computes a sensitivity map that quantifies how strongly each input pixel and its associated synapses affect pre-activation under faults. Second, the Gini-coefficient module searches over 1D projection angles and selects the direction along which the accumulated sensitivity is most evenly distributed, defining a fair axis for fragmentation. Third, the fragment processing module cuts the image along this axis into equal-sensitivity fragments, normalizes each fragment’s energy via RMS normalization to keep pre-activations inside the surrogate corridor, and then accumulates time-step outputs with entropy-based weighting [1, 2]. Overall, these three modules cooperate by identifying fault-sensitive pixels, then choosing the most balanced way to partition them, and finally enforcing the pre-activation in the surrogate corridor by RMS normalization [1]. Moreover, we ensure the accurate decoding of fragment-oriented outputs by an entropy-based approach [2]. To address your concern, we have made a new Subsection named ‘Overview’ in Section 6 of the revised manuscript. Please refer to the items of the new Subsection (Subsection 5.1) of the revised manuscript.
>
> **W5**: We really appreciate your valuable comment on the contents that deal with FPGA in Abstract and Section 1. In our experiments, the proposed mechanism works on the FPGA device with the following procedures. We train the fault-injected SNN model in a software environment with the proposed mechanism since current FPGA-based SNN models do not support on-chip learning due to circuit-level limitations [3, 4]. After training, we save the trained model parameters and convert the Python script of the SNN models to the VHDL script. Then, we synthesize the FPGA circuit with the VHDL script and conduct inference with the synthesized SNN models. The proposed mechanism provides input fragments with the synthesized SNN models during inference, like training. Note that we directly implement SNN models on the FPGA chip and the proposed mechanism on the additional control processor connected to the FPGA chip. Please refer to the explanation in Subsection A.7 of Appendix. We try to demonstrate that our mechanism properly provides the input fragments to the FPGA-based SNN models and successfully improves the fault tolerance of the hardware-implemented SNN models because FPGA devices are necessary to develop hardware-based neural networks [3, 5]. Moreover, we agree with your comment that the main paper does not contain the FPGA-related contents. Thus, we reduce the contents of FPGA-based experiments from the main contributions in Section 1 of the revised manuscript. Instead, we mention that we conduct FPGA-based experiments in Section 6. Please refer to contributions in Section 1 and the first paragraph in Section 6 of the revised manuscript.

---

> > ### Author Response · Authors · 2025-11-13
> > **References for the rebuttal**
> >
> > # References
> >
> > [1] Zhang, Biao & Sennrich, Rico (2019). Root Mean Square Layer Normalization. In Advances in Neural Information Processing Systems 32 (pp. 12360–12371). Curran Associates Inc
> >
> > [2] Qiu, Zexuan; Ou, Zijing; Wu, Bin; Li, Jingjing; Liu, Aiwei; King, Irwin (2025). Entropy-Based Decoding for Retrieval-Augmented Large Language Models. In Proceedings of the 2025 Conference of the North American Chapter of the Association for Computational Linguistics (Long Papers) (pp. 4616–4627). Albuquerque, New Mexico: Association for Computational Linguistics
> >
> > [3] Carpegna, A., Savino, A., and Di Carlo, S., “Spiker+: a framework for the generation of efficient Spiking Neural Networks FPGA accelerators for inference at the edge,” IEEE Transactions on Emerging Topics in Computing, pp. 1–15, 2024, doi: 10.1109/TETC.2024.3511676
> >
> > [4] Tao, Y., Ma, R., Shyu, M.-L., and Chen, S.-C., “Challenges in Energy-Efficient Deep Neural Network Training with FPGA,” in Proceedings of the 2020 IEEE/CVF Conference on Computer Vision and Pattern Recognition Workshops (CVPRW), 2020, pp. 1602–1611, doi: 10.1109/CVPRW50498.2020.00208
> >
> > [5] Mehrzad Karamimanesh, Ebrahim Abiri, Mahyar Shahsavari, Kourosh Hassanli, André van Schaik, and Jason Eshraghian, “Spiking neural networks on FPGA: A survey of methodologies and recent advancements,” Neural Networks, vol. 186, p. 107256, 2025, doi: 10.1016/j.neunet.2025.107256

---

> ### Comment · Reviewer_EvpD · 2025-11-13
> **Appreciate the rebuttal from authors**
>
> Thanks for the responses to the questions. Most of the concerns have been solved, and I have raised the score of the presentation to 2, and the overall score to 4 accordingly.
>
> 1,3,5. Thanks for the response; it addressed my concerns.
>
> 2. "SNN-based neuromorphic devices" is confusing (e.g., it is not clear how neuromorphic hardware is based on SNNs?), and it is not usually used. I suggest revising it to make the concept clearer.
>
> 4. Adding an overview at the beginning of this section makes the methods much clearer and addresses my concerns. Thanks for the revisions.

---

> ### Author Response · Authors · 2025-11-14
> **Thank you for your comment on our rebuttal!**
>
> We really appreciate you for raising the score! Following your suggestion, we have changed the title of Subsection 3.2 to ‘Mechanisms to improve fault tolerance of hardware-implemented SNNs’ and replaced the expressions similar to SNN-based neuromorphic in Subsection with ‘hardware-implemented SNNs in neuromorphic devices’. Please refer to the title and first paragraph of Subsection 3.2 in the revised manuscript. Thanks again for raising the score!

---

> ### Author Response · Authors · 2025-11-18
>
> Dear, Reviewer EvpD
>
> Thank you again for the constructive feedback. If you have any further comments or suggestions on the revised manuscript, we would be very happy to address them.

---

### Official Review · Reviewer_Q1C8 · 2025-10-30

**Soundness:** 2
**Presentation:** 4
**Contribution:** 3
**Rating:** 4
**Confidence:** 5

**Summary:**

The paper studies fault tolerance in SNNs, which are prone to performance degradation from faults like Stuck-At-Faults (SAFs) in synaptic weights. The authors identify a bottleneck problem: faults cause pre-activation values to drift outside the surrogate gradient corridor, leading to vanishing gradients and a severe reduction in the network's usable learning capacity.

To solve this, they propose

i) a novel and simple mechanism inspired by flow control in computer networks. Instead of modifying the SNN's weights or architecture, their method controls the input. The core idea is to fragment input images into smaller pieces based on a sensitivity score that combines image complexity (edges, texture) and fault influence.
ii) They find the optimal cutting angle by minimizing the Gini coefficient of the 1D projection of this score, ensuring each fragment has a balanced information load. These fragments are then fed sequentially to the SNN, and the outputs are aggregated using an entropy-based weighting scheme.
iii) The proposed method is evaluated extensively on various SNN models (MLP, VGG-7/11/15, ResNet-18/34) and datasets (MNIST, FMNIST, CIFAR-10/100, Tiny-ImageNet, UCI-HAR) under different fault types (SAFs, RWFs, CEFs), the mechanism yields the highest accuracy at a given fault ratio versus benchmarks (ECOC, SoftSNN, Routing, Astrocyte, FalVolt, LIFA).

**Strengths:**

The authors have touched each dimension of originality, quality, clarity, and significance.

Originality: The authors focus on input fragmentation plus fault-influence guidance over weight-level scanning, explicitly targeting the surrogate-gradient corridor to prevent gradient bottlenecks. The approach of using input fragmentation controlled by a Gini-optimized strategy is novel and also provides a theoretical analysis (on corridor occupancy/gradient attenuation and capacity thresholds) that explains failure modes under SAF/RWF/CEF and why the mechanism helps. The analogy to network flow control is creative and provides a strong, intuitive foundation.

Quality: The authors do not show just an empirical demonstration but support a thorough motivation study that meticulously shows how faults lead to pre-activation drift and gradient collapse. The appendices provide a rigorous mathematical framework for both the problem and the near-optimality of their solution. The experimental evaluation is extensive, covering multiple models, datasets, fault types, time steps, ablation studies, hyperparameter sensitivity, and even a comparison with DNNs and a real FPGA implementation.

Clarity: The paper is well-written and structured. The problem is clearly motivated, the mechanism is explained step-by-step with the help of key points, and the figures and tables effectively support the claims. The use of a simple, high-level analogy (flow control) makes the complex underlying concept more accessible.

Significance: The proposed mechanism directly targets a critical and common limitation in deployed SNNs, especially for resource-constrained or neuromorphic platforms. Its low complexity and implementation compatibility make it highly relevant for practitioners and researchers seeking robust edge AI solutions.

**Weaknesses:**

1. The explanation of the "bottleneck problem" in the paper (at line 60) lacks conceptual clarity and mixes two different learning regimes. The authors first state that "when faults appear in SNNs' synapses, the weights of the faulty synapses become fixed during training," implying that training is happening on-chip, where hardware faults would indeed interfere with plasticity and learning. However, the next line attributes the capacity degradation to surrogate gradient vanishing due to abnormal pre-activation values, this is clearly a reference to offline training using backpropagation through time (BPTT) and surrogate gradients, as implemented in frameworks like snnTorch or SpikingJelly.

This conflation is problematic. In most practical settings, SNNs are trained offline on fault-free software platforms, and then deployed on neuromorphic hardware. If faults arise, they typically occur after training, during deployment, due to physical issues such as resistance drift, electromigration, peripheral CMOS aging, or read-disturb effects. Thus, during offline BPTT training, the weights are unaffected by hardware faults. On the other hand, if the authors intend to analyze on-chip learning, then the appropriate learning rule would be local, online methods like STDP, not surrogate-gradient-based BPTT. In that case, the "gradient vanishing" explanation does not apply.

In summary, it is implying a vague explanation and needs clarification for accurately motivating the problem and for aligning the theoretical analysis with real-world neuromorphic deployment.


2.) The paper focuses solely on synaptic faults (e.g., stuck-at, random-weight), but this overlooks other critical fault modes, especially given that the model is deployed on FPGA hardware. In digital neuromorphic systems, faults can also arise in core arithmetic components such as adders, multipliers, counters, and comparators, which directly impact spiking neuron-level computations. Prior works
https://ieeexplore.ieee.org/abstract/document/10658724
https://ieeexplore.ieee.org/abstract/document/10858960

have shown that such logic-level faults can significantly degrade SNN performance. A broader fault model or at least a discussion acknowledging these hardware-level vulnerabilities would strengthen the paper's scope and relevance for real-world neuromorphic deployment.

3.) The Section 4. Motivation Study frames the impact of synaptic faults entirely from the perspective of offline software-based BPTT training, which is not representative of how SNNs operate on neuromorphic hardware. In practice, BPTT cannot be used on neuromorphic devices, only online, local learning rules such as STDP are hardware-compatible. Prior works (e.g., Vatajelu et al., 2019; Lee & Lim, 2023) have correctly modeled faults within unsupervised, on-chip STDP-based learning, which reflects real-world behavior. Without grounding the fault impact in such realistic learning settings, the motivation for the proposed input-fragmentation mechanism remains speculative. A more appropriate justification would consider how faults affect STDP-based learning dynamics or inference-time reliability post offline training.

4.) While the authors provide code as supplementary material, there is no accompanying README or documentation explaining how to execute it. This makes it difficult to reproduce the experiments or understand the workflow. Additionally, although the paper reports FPGA-based results, the supplementary materials contain only Python (.py) files, with no Verilog: hardware-specific code required for actual FPGA deployment. Including these files or at least providing a pointer to the hardware implementation would significantly enhance the reproducibility and credibility of the hardware claims.

5.) In Appendix A.7 (line 1296), the authors mention FPGA evaluation using only an MLP model. However, other models used in the paper, such as VGG-7/11/15 and ResNet-18/34 are not included in the hardware experiments. It remains unclear why these deeper architectures were omitted, especially since they form a core part of the software evaluation. Including them would provide a more complete and realistic assessment of the proposed method's hardware applicability.

6.) Hardware results show the pattern but the device/precision configuration dominates performance. A clearer breakdown of numeric formats, bit-width per layer, and resource utilization vs accuracy would strengthen reproducibility claims for hardware.

7.) The proposed approach is heavily designed towards 2D images. How would the approach generalize it to other data modalities, such as audio (1D time series) or text? The paper's significance would be greatly amplified if the core idea could be shown to be more broadly applicable.

8.) Though the method is optimized using sensitivity metrics and Gini coefficients, practical constraints (batch effects, alignment in hardware) may limit its generality. Further discussion of trade-offs in real-world settings (e.g., latency, fragment count vs. accuracy) would be valuable.

9.) Is there potential to dynamically rather than statically per batch adapt the fragmentation strategy during training as fault characteristics evolve, and how might this affect convergence and hardware cost?

10.) The method focuses on gradient-based supervised SNN training. Would similar fragmentation principles benefit unsupervised SNNs?

**Questions:**

I would request authors to answer all points that are raised in Weaknesses.

---

> ### Author Response · Authors · 2025-11-21
> **Rebuttal to Reviewer Q1C8 (1/9)**
>
> # Thanks for Reviewer Q1C8
>
> We really appreciate your comments on improving the accuracy of the hardware environmental assumptions and the reliability of the experimental results in our paper. Your comments provide us with meaningful insights into hardware settings and assumptions of on-chip learning, substantially strengthening the concreteness of our paper.
>
> ---
>
> # Responses to the weaknesses
>
> **W1**: We really appreciate your valuable comment on the learning rules in online and offline learning methods. We target the scenario that gradient-based learning is executed on chip during training, and hardware faults occur during training due to hardware instability [1, 2]. As you pointed out, neuromorphic devices using surrogate gradient use offline training methods, and the faults occur during inference. **However, current studies have shown that neuromorphic devices, such as SpiNNaker 2, Intel Loihi, and BrainScaleS-2, support full or partial on-chip learning with surrogate gradients [3, 4, 5, 6, 7, 8, 9].** On-chip learning based on surrogate gradients is increasingly important to improve the learning performance of neuromorphic devices due to the following two reasons [5, 6, 8, 9]. First, offline-based learning suffers from device mismatch problems, leading to performance degradation due to synaptic weights that are not adjusted with hardware characteristics [6]. Second, to adapt the edge AI devices depending on the surrounding environments of the devices, high-performance on-chip learning based on gradients is important [3, 5, 9]. **As on-chip learning with gradients becomes more significant than before, hardware faults that occur during training in on-chip learning devices become important [1, 2, 3, 4, 5, 6, 7]. Therefore, studying the fault tolerance of on-chip learning neuromorphic devices using surrogate gradients is necessary.** Our problem definition, the bottleneck problem, is precisely aligned with the direction in which current neuromorphic hardware is evolving, rather than being a purely idealized setting detached from reality. To address your comment, we clarify that our mechanism’s assumption on the hardware setting is valid in Section 1. Please refer to the fifth paragraph of Section 1 in the revised manuscript.
>
> **W2**: We really appreciate your insightful comment on the fault mitigation ability of our mechanism with logic-level faults in FPGA devices. In digital neuromorphic systems, faults in adders and multipliers cause SAFs since the synaptic weights do not change or are not multiplied to input samples due to faults in adders or multipliers [10]. To investigate faults in counters and comparators, we adopt the level fault models in the work that you mentioned [11]. We inject missing/delayed and spurious spike faults into all neurons of our FPGA-implemented MLP model for MNIST and FMNIST classification. These spike faults occur because neurons cannot compare the membrane potential and threshold due to faults in comparators, causing abnormal spikes [11]. The following tables present the classification accuracy of the MLP model using the baseline, benchmarks, and proposed mechanism under spike faults with 2 time steps.
>
> **MNIST**
>
> | Fault type                 | Baseline        | ECOC            | SoftSNN         | Routing         | Astrocyte       | Falvolt          | LIFA             | Proposed         |
> |---------------------------|-----------------|------------------|------------------|------------------|------------------|-------------------|-------------------|-------------------|
> | Missing/delayed spikes    | 75.87±4.26      | 78.36±4.05       | 77.51±4.14       | 80.76±3.96       | 36.34±2.83       | 79.42±4.48        | 40.69±3.28        | 80.04±3.88        |
> | Spurious spikes            | 77.15±4.38      | 79.52±4.27       | 78.93±4.20       | 82.79±4.13       | 36.96±3.16       | 80.37±4.25        | 41.81±3.69        | 83.24±4.34        |
>
> **FMNIST**
>
> | Fault type               | Baseline    | ECOC       | SoftSNN     | Routing     | Astrocyte   | Falvolt     | LIFA        | Proposed    |
> |--------------------------|-------------|------------|-------------|-------------|-------------|-------------|-------------|-------------|
> | Missing/delayed spikes   | 54.34±5.23  | 58.55±5.16 | 57.61±5.36  | 61.6±5.09   | 18.44±1.86  | 59.92±5.39  | 22.69±3.02  | 62.37±5.42  |
> | Spurious spikes           | 55.91±5.64  | 59.33±5.73 | 59.58±5.47  | 62.48±4.91  | 20.68±2.25  | 61.08±5.51  | 25.81±3.19  | 65.69±5.52  |
>
> As shown in the tables, our mechanism successfully enhances the fault tolerance of hardware-implemented SNNs. This is because our mechanism reduces the power of input samples by fragmentation, degrading the probability of abnormal spike occurrence.

---

> ### Author Response · Authors · 2025-11-21
> **Rebuttal for Reviewer Q1C8 (2/9)**
>
> **W3**: We really appreciate your valuable comment on the assumption of the gradient-based learning in our motivation study. **As we mentioned in the response to weakness 1, hardware-implemented SNN models can execute the gradient-based learning methods on chips (FPGA and neuromorphic devices) during training [3, 4, 5, 6, 7, 8, 9]. The necessity of gradient-based on-chip learning approaches has increased substantially, and analysis for the faults in the hardware-implemented SNNs based on these learning approaches becomes important.** Therefore, we apply hardware fault models for neuromorphic devices using gradient-based on-chip learning approaches [10, 12]. We agree with your opinion that STDP-based on-chip learning is also important for the implementation of SNNs in neuromorphic devices. To demonstrate that our fragmentation mechanism is valid with STDP-based SNNs, we adopt our mechanism to Diehl&Cook2015 architecture, which is a widely used unsupervised SNN model based on STDP, as presented in response to your weakness 10 [13]. Additionally, we explain how our mechanism enhances the fault tolerance. Hardware faults increase the absolute value of membrane potential (pre-activation in SNN using gradient-based learning rules) significantly, and this abnormal increase causes the over- or under-firing of spiking neurons of Diehl&Cook2015 [14, 15, 16]. When faulty Diehl&Cook2015 obtains the whole data samples that are not fragmented, fault-injected neurons’ membrane potential always increases or decreases excessively, preventing the neurons from spiking properly since the pixel values easily enter the fault-injected synapses. Specifically, the over-firing neurons always omit spikes dominantly and suppress other neurons’ spike activity because of the lateral inhibition caused by the Winner-Takes-All (WTA) algorithm. We fragment the input samples into small pieces by considering the complexity of the samples and the influence of faults to minimize the adversarial effects of faults and prevent neurons from over- or under-firing. Through fragmentation, our mechanism reduces the chance that the input pixels enter faulty synapses, and the neurons’ membrane potential avoids increasing or decreasing abnormally. Please refer to our response for your weakness 10 and Subsection B.7 of Appendix in the revised manuscript to check the detailed experimental results.
>
> **W4**: We really appreciate your valuable comment on the reproducibility of our experimental results. To address your concern, we have written a README.md file and included it in our supplementary material (.zip file). Furthermore, we have included the VHDL scripts in our supplementary material.  Please refer to the new version of our supplementary material.

---

> ### Author Response · Authors · 2025-11-21
> **Rebuttal for Reviewer Q1C8 (3/9)**
>
> **W5**: We really appreciate your insightful comment on the implementation of complex SNN models with FPGA devices. We cannot implement complex models such as ResNet-18/34 due to limitations in computational resources and implementation complexity [16, 17, 18]. However, current studies provide methods to implement intermediate-sized SNN models, such as 5-layer MLP and VGG-13, in FPGA devices [20, 21]. We utilize [20] to implement VGG-7/11/15 in our FPGA device and adopt the benchmarks and proposed mechanism to enhance the fault tolerance of hardware-implemented SNNs, as we did with [19]. The following table exhibits the classification accuracy of VGG-7/11/15 with the baseline, benchmarks, and proposed mechanism.
>
> **VGG-7**
> |Fault ratio (%)|Baseline|ECOC|SoftSNN|Routing|Astrocyte|FalVolt|LIFA|Proposed|
> |-------------|--------|----|--------|--------|----------|--------|------|---------|
> |0|53.84±4.94|51.09±4.86|54.08±4.67|**54.15±4.26**|53.21±4.60|53.91±4.57|52.72±4.65|53.89±4.48|
> |10|34.18±4.15|39.26±4.07|22.48±4.46|33.37±4.12|32.16±4.54|36.92±4.82|35.38±4.73|**45.93±4.31**|
> |20|18.73±3.07|31.97±3.65|17.52±5.11|24.06±3.71|22.18±3.95|31.91±3.79|21.99±4.02|**41.23±4.05**|
> |30|11.62±1.62|25.78±3.14|16.80±3.92|10±0|10±0|20.70±4.15|10±0|**35.97±4.88**|
> |40|10±0|19.29±3.28|15.04±3.17|10±0|10±0|17.28±3.84|10±0|**26.29±3.96**|
> |50|10±0|12.81±2.25|10±0|10±0|10±0|10±0|10±0|**13.39±2.94**|
> |60|10±0|10±0|10±0|10±0|10±0|10±0|10±0|10±0|
> |70|10±0|10±0|10±0|10±0|10±0|10±0|10±0|10±0|
> |80|10±0|10±0|10±0|10±0|10±0|10±0|10±0|10±0|
> |90|10±0|10±0|10±0|10±0|10±0|10±0|10±0|10±0|
>
> **VGG-11**
> |Fault ratio (%)|Baseline|ECOC|SoftSNN|Routing|Astrocyte|FalVolt|LIFA|Proposed|
> |-------------|--------|----|--------|--------|----------|--------|------|---------|
> |0|52.74±4.63|50.34±4.51|**54.82±4.32**|54.29±4.45|52.13±4.63|53.89±4.38|53.02±4.47|52.87±4.20|
> |10|33.27±3.95|33.02±3.84|26.53±3.72|32.98±3.93|34.16±4.05|41.81±4.24|35.57±4.13|**45.02±4.08**|
> |20|26.91±3.28|30.83±3.37|18.88±3.60|24.17±3.56|24.93±3.75|27.30±3.43|24.89±3.94|**42.58±3.79**|
> |30|10±0|25.65±3.51|17.63±3.92|10±0|16.89±3.81|10±0|11.46±1.46|**36.62±3.84**|
> |40|10±0|20.27±3.64|10±0|10±0|10±0|10±0|10±0|**26.84±3.29**|
> |50|10±0|14.44±3.76|10±0|10±0|10±0|10±0|10±0|**17.92±3.65**|
> |60|10±0|10±0|10±0|10±0|10±0|10±0|10±0|10±0|
> |70|10±0|10±0|10±0|10±0|10±0|10±0|10±0|10±0|
> |80|10±0|10±0|10±0|10±0|10±0|10±0|10±0|10±0|
> |90|10±0|10±0|10±0|10±0|10±0|10±0|10±0|10±0|
>
> **VGG-15**
> |Fault ratio (%)|Baseline|ECOC|SoftSNN|Routing|Astrocyte|FalVolt|LIFA|Proposed|
> |-------------|--------|----|--------|--------|----------|--------|------|---------|
> |0|45.04±4.07|41.92±3.95|**47.39±3.81**|43.58±3.86|43.02±3.77|45.58±3.26|44.61±3.74|44.35±4.61|
> |10|40.16±5.24|31.19±4.76|14.21±1.91|41.50±4.83|32.28±5.13|41.87±4.59|29.84±5.33|**42.94±5.09**|
> |20|21.42±4.64|23.70±4.59|13.22±2.08|25.28±4.29|10±0|10±0|20.37±3.91|**41.47±4.96**|
> |30|10±0|21.26±3.54|12.89±1.75|10±0|10±0|10±0|10±0|**33.62±5.17**|
> |40|10±0|15.28±2.94|11.01±1.01|10±0|10±0|10±0|10±0|**27.59±4.25**|
> |50|10±0|10±0|10±0|10±0|10±0|10±0|10±0|**11.74±1.19**|
> |60|10±0|10±0|10±0|10±0|10±0|10±0|10±0|10±0|
> |70|10±0|10±0|10±0|10±0|10±0|10±0|10±0|10±0|
> |80|10±0|10±0|10±0|10±0|10±0|10±0|10±0|10±0|
> |90|10±0|10±0|10±0|10±0|10±0|10±0|10±0|10±0|
>
> As shown in the table, our mechanism significantly enhances the fault tolerance of hardware-implemented SNNs compared to the benchmarks. To address your concern, we have added these experimental results to Subsection B.8 of Appendix. Please refer to Subsection B.8 of Appendix in the revised manuscript.

---

> ### Author Response · Authors · 2025-11-21
> **Rebuttal for Reviewer Q1C8 (4/9)**
>
> **W6**: We really appreciate your valuable comment on resource settings for the hardware-implemented SNN with the FPGA device. To address your comment, we set the bit width for the membrane potential and synaptic weights to 2, 4, 6, and 8, measuring the classification accuracy of the FPGA-based MLP model with our mechanism. The following tables show the classification accuracy of the model with different bit-width settings for the membrane potential and synaptic weights using our mechanism under a 30% fault ratio of SAFs. We use MNIST and FMNIST for the experiments, setting the number of time steps to 2.
>
> **Changing the bit-width of the membrane potential, using 6 bits for the synaptic weights (MNIST)**
> |           | 2 bits       | 4 bits        | 6 bits         | 8 bits         |
> |-----------|--------------|---------------|----------------|----------------|
> | Accuracy (%) | 69.23±4.58 | 89.84±3.81 | 90.28±3.32 | 90.57±3.16 |
>
> **Changing the bit-width of the synaptic weights, using 8 bits for the membrane potential (MNIST)**
> |            | 2 bits | 4 bits | 6 bits       | 8 bits       |
> |------------|--------|--------|--------------|--------------|
> | Accuracy (%) | 10±0 | 10±0 | 90.57±3.16 | 90.64±3.09 |
>
> **Changing the bit-width of the membrane potential, using 6 bits for the synaptic weights (FMNIST)**
> |           | 2 bits      | 4 bits       | 6 bits        | 8 bits        |
> |-----------|-------------|--------------|---------------|---------------|
> | Accuracy (%) |62.83±4.54 |78.25±3.98   |81.34±3.66     |81.71±3.7      |
>
> **Changing the bit-width of the synaptic weights, using 8 bits for the membrane potential (FMNIST)**
> |           | 2 bits | 4 bits | 6 bits       | 8 bits        |
> |-----------|--------|--------|--------------|---------------|
> | Accuracy (%) |10±0   |10±0   |80.48±3.42   |81.71±3.7      |
>
> As shown in the tables, the classification accuracy increases as the bit-widths of the membrane potential and synaptic weights increase. This is because the hardware-implemented SNN uses more bits to represent the potential and weights, thereby improving the precision of the values when the bit-width is large [21].

---

> ### Author Response · Authors · 2025-11-21
> **Rebuttal for Reviewer Q1C8 (5/9)**
>
> **W7**: We really appreciate your valuable comment on the applications in 1D sequential data. Beyond UCI-HAR, a 1D dataset containing human activity data samples, we utilize AudioMNIST to evaluate the fault mitigation ability of our mechanism across various 1D sequential datasets, addressing your concern. The following table exhibits the classification accuracy (%) of the MLP model using the baseline, benchmarks, and proposed mechanism under SAFs with 2 time steps.
>
> |Fault ratio (%)|Baseline|ECOC|SoftSNN|Routing|Astrocyte|FalVolt|LIFA|Proposed|
> |---|---|---|---|---|---|---|---|---|
> |0|96.45±0.91|**96.88±0.97**|95.89±0.92|96.21±0.85|96.29±0.93|96.34±0.89|96.09±0.93|96.31±0.87|
> |10|94.47±1.07|94.56±1.79|92.56±1.61|94.03±1.84|93.08±1.92|94.47±1.58|93.28±2.18|**94.98±1.54**|
> |20|93.17±1.56|93.29±1.98|88.15±2.84|92.59±2.02|85.24±3.68|93.29±2.13|88.57±4.26|**93.78±1.98**|
> |30|92.89±2.05|93.33±2.32|75.83±3.51|91.70±2.75|51.47±5.56|92.76±2.47|55.81±6.01|**93.43±3.09**|
> |40|89.50±3.21|92.51±2.15|62.49±5.07|91.24±3.18|37.55±6.27|90.63±2.99|39.64±6.80|**92.65±3.17**|
> |50|86.15±4.18|85.46±3.83|48.61±6.92|86.45±3.97|22.13±4.49|87.51±4.24|24.48±5.07|**88.93±4.16**|
> |60|52.87±5.71|66.42±5.20|34.62±5.33|64.92±5.34|10.96±0.96|65.97±5.93|12.74±2.38|**67.83±5.77**|
> |70|14.60±1.23|21.64±2.15|14.58±1.40|25.84±3.42|10±0|28.18±3.85|10±0|**37.17±3.01**|
> |80|11.77±0.95|13.38±0.94|9.83±1.05|14.19±1.36|10±0|15.53±2.36|10±0|**19.75±2.89**|
> |90|11.21±1.01|11.86±0.87|9.46±1.22|11.68±0.96|10±0|10±0|10±0|**12.83±1.02**|
>
> As shown in the table, our mechanism exhibits higher fault mitigation ability than the benchmarks when we use AudioMNIST. We have added this experimental result to Subsubsection B.1.2 of Appendix. Please refer to Subsubsection B.1.2 of Appendix in the revised manuscript.
>
> We also conduct experiments with the FPGA-based MLP model using AudioMNIST. The following table shows the classification accuracy (%) of the MLP model using the baseline, benchmarks, and proposed mechanism under SAFs with 2 time steps.
>
> |Fault ratio (%)|Baseline|ECOC|SoftSNN|Routing|Astrocyte|FalVolt|LIFA|Proposed|
> |---|---|---|---|---|---|---|---|---|
> |0|93.76±1.34|**93.82±1.6**|92.02±1.66|92.63±1.21|92.36±1.53|91.38±1.27|93.33±1.29|92.34±1.38|
> |10|91.80±1.22|90.89±2.12|89.32±2.01|89.87±2.22|90.01±2.45|90.71±2.45|90.41±2.74|**91.56±1.93**|
> |20|89.45±2.1|88.92±2.56|84.39±3.41|89.59±2.50|81.61±3.81|89.42±2.68|84.47±4.79|**90.86±2.83**|
> |30|89.04±2.44|89.65±2.77|72.35±3.93|88.19±3.28|48.31±5.73|89.57±3.01|51.64±6.53|**89.8±3.42**|
> |40|86.24±3.59|90.26±2.58|59.43±5.48|87.99±3.97|34.69±6.61|87.8±3.42|35.73±7.49|**89.27±4.13**|
> |50|83.50±4.63|82.63±4.43|45.26±7.31|82.39±4.42|19.38±5.2|83.21±4.93|21.22±5.56|**86.29±4.9**|
> |60|49.11±6.32|62.20±5.82|31.23±5.78|61.16±5.94|10±0|62.48±6.63|10±0|**63.88±6.34**|
> |70|10±0|16.74±2.45|10±0|21.92±3.84|10±0|24.72±4.47|10±0|**34.75±3.76**|
> |80|10±0|10±0|10±0|10±0|10±0|13.15±2.14|10±0|**17.29±3.55**|
> |90|10±0|10±0|10±0|10±0|10±0|10±0|10±0|10±0|
>
> Our mechanism works well when the FPGA-based model classifies AudioMNIST data samples. We have added this result to Section B.8 of Appendix. Please refer to Subsection B.8 of Appendix in the revised manuscript.

---

> ### Author Response · Authors · 2025-11-21
> **Rebuttal for Reviewer Q1C8 (6/9)**
>
> **W8**: We really appreciate your valuable comment on the need for a more explicit discussion about the trade-offs between the accuracy and the number of fragments (latency). To address your concern, we conduct experiments with hardware, changing the number of fragments, and measure the classification accuracy and latency for processing a single sample of the FPGA-based MLP model with our mechanism. The following tables present the accuracy and latency of the model with different fragmentation numbers when the fault ratio of SAFs is 30%, using MNIST and FMNIST.
>
> **MNIST**
>
> |           | 2 time steps | 4 time steps | 8 time steps |
> |-----------|-------------|--------------|--------------|
> | Accuracy (%) | 90.57±3.16 | 91.28±2.85 | 74.31±3.9 |
> | Latency (µs) | 1.24±0.12 | 2.63±0.19 | 5.54±0.27 |
>
> **FMNIST**
>
> |           | 2 time steps | 4 time steps | 8 time steps |
> |-----------|-------------|--------------|--------------|
> | Accuracy (%) | 81.71±3.7 | 83.98±3.26 | 85.36±3.54 |
> | Latency (µs) | 1.43±0.18 | 3.02±0.23 | 5.97±0.25 |
>
> As shown in the tables, the accuracy and latency of the hardware-implemented SNN using our mechanism straightforwardly increase with the number of time steps (fragments).
>
> Furthermore, we change the batch size and distribution of samples in each batch to investigate the effects of different batch settings on our mechanism in the FPGA-based MLP. The following table exhibits the classification accuracy of the model with different batch sizes when the fault ratio of SAFs is 30% and the number of time steps is 2, using MNIST and FMNIST.
>
> |                       | batch size 50  | batch size 100 | batch size 200 |
> |-----------------------|---------------|----------------|----------------|
> | MNIST Accuracy (%)     | 90.29±3.25    | 90.57±3.16     | 90.48±3.54     |
> | FMNIST Accuracy (%)    | 81.38±3.59    | 81.71±3.7      | 81.95±3.86     |
>
> The tables demonstrate that the model with our mechanism successfully enhances the model’s fault tolerance, regardless of the batch size.
>
> We make the samples in the training batches non-Independent and Identically Distributed (non-IID). Non-IID batches have two and five classes of data samples. The following table presents the classification accuracy of the model with IID and Non-IID batches when the fault ratio of SAFs is 30% and the number of time steps is 2, using MNIST and FMNIST.
>
> |                       | IID batch | Non-IID batch (5 classes)  | Non-IID batch (2 classes) |
> |-----------------------|---------------|----------------|---|
> | MNIST Accuracy (%)     | 90.57±3.16 |  90.51±3.26  | 90.39±3.65    |
> | FMNIST Accuracy (%)   | 81.71±3.7   |  80.62±4.29  | 78.32±4.82    |
>
> As presented in the table, the accuracy of the FPGA-based MLP incorporating our mechanism does not degrade significantly despite non-IID data samples in batches.
>
> We also investigate the effects of hardware alignments (bit-width) and observe that the accuracy of the FPGA-based model with our mechanism increases as the bit-width increases. Please refer to our response to weakness 6.

---

> ### Author Response · Authors · 2025-11-21
> **Rebuttal for Reviewer Q1C8 (7/9)**
>
> **W9**: We really appreciate your insightful comment on adopting the case in which hardware faults dynamically change. **Our fragmentation operates dynamically in a batch-wise manner (We mention it in Subsection 5.1 of the revised manuscript.).** Due to the batch-wise dynamicity, our mechanism can identify the characteristics of dynamically changing faults and establish proper strategies to enhance fault tolerance. For each mini batch, we recompute the sensitivity scores and partition all input samples in that batch into fragments, as described in Section 5 of the revised manuscript. In the selectivity score computation, WP captures the adversarial effects of first-layer faults, while SP identifies input pixels whose perturbations under faults lead to substantial output changes. Because WP and SP are recomputed for every mini batch, the fragmentation strategy is updated batch by batch, enabling our mechanism to track evolving fault characteristics. To present the convergence of the hardware cost of the FPGA-based MLP model with our mechanism, with dynamically changing faults, we dynamically change the position and stuck value of SAFs during training and inference. We also compare the results of the model with dynamically changing SAFs to those of the model with static SAFs. The following tables show classification accuracy (%) and energy consumption (mJ) to process a single data sample of the FPGA-based MLP model with the number of epochs using MNIST and FMNIST under the 30% fault ratio of SAFs.
>
> **MNIST**
>
> |                        | 10 epochs     | 25 epochs     | 50 epochs     | 100 epochs     |
> |------------------------|--------------|--------------|--------------|----------------|
> | Static accuracy        | 35.61±10.38  | 88.46±4.52    | 90.57±3.16   | 90.82±2.59     |
> | Dynamic accuracy       | 13.49±3.49   | 64.97±9.16    | 85.43±6.28   | 89.21±3.94     |
> | Static energy consumption  | 102.84±1.25 | 84.33±1.37    | 67.16±0.82   | 66.53±0.70     |
> | Dynamic energy consumption | 130.27±2.96 | 115.51±2.64   | 89.39±2.26   | 78.49±1.06     |
>
> **FMNIST**
>
> |                         | 10 epochs     | 25 epochs     | 50 epochs     | 100 epochs     |
> |-------------------------|--------------|--------------|--------------|----------------|
> | Static accuracy         | 40.98±9.64   | 78.53±6.27   | 81.71±3.7    | 82.09±2.96     |
> | Dynamic accuracy        | 21.96±8.17   | 62.82±9.63   | 79.55±4.36   | 80.18±3.58     |
> | Static energy consumption   | 121.11±2.36  | 100.98±1.76  | 78.37±0.95   | 77.64±0.82     |
> | Dynamic energy consumption  | 146.85±3.83  | 127.44±3.63  | 95.68±2.91   | 87.86±1.75     |
>
> As demonstrated in the tables, the synaptic weights of the FPGA-based MLP incorporating our mechanism stably converge on the values that ensure accurate classification. The energy consumption of the MLP does not increase substantially when the faults dynamically change.

---

> ### Author Response · Authors · 2025-11-21
> **Rebuttal for Reviewer Q1C8 (8/9)**
>
> **W10**: We really appreciate your comment on the availability of our mechanism with the STDP-based SNN model. To address your comment, we apply the benchmarks and proposed mechanism to Diehl&Cook2015, which is a widely used SNN architecture with STDP [13, 22]. We set the number of time steps to 250 and the number of fragments to 2. The model obtains fragment 1 repeatedly during the first 125 time steps and fragment 2 during the second 125 time steps. We remove ECOC from the experiment since it is not available in Deihl&Cook2015 due to its implementation difficulty. The following tables exhibit the classification accuracy (%) of the Diehl&Cook2015 model using the baseline, benchmarks, and proposed mechanism under SAFs.
>
> **MNIST**
>
> |Fault ratio (%)|Baseline|SoftSNN|Routing|Astrocyte|FalVolt|LIFA|Proposed|
> |---|---|---|---|---|---|---|---|
> |0|86.37±1.23|85.96±1.45|86.34±1.28|86.49±1.65|**86.41±1.61**|86.16±1.54|85.69±1.72|
> |10|77.07±1.86|79.34±2.51|78.5±2.08|79.16±1.93|78.89±2.02|79.58±2.37|**81.03±2.44**|
> |20|76.41±1.9|78.48±2.67|77.27±2.36|78.91±1.88|78.04±2.45|78.61±2.51|**80.27±2.53**|
> |30|74.45±2.79|76.59±2.81|77.03±3.05|77.62±2.56|78.04±3.16|78.72±2.98|**79.86±3.14**|
> |40|70.87±4.52|71.95±4.93|72.35±4.74|72.48±4.59|73.19±4.72|73.43±4.6|**76.57±3.66**|
> |50|69.7±5.15|70.83±5.26|72.08±4.91|71.27±5.38|71.96±5.84|71.78±5.64|**74.36±5.42**|
> |60|67.79±5.98|69.25±6.2|70.7±6.12|69.49±5.84|70.35±6.27|70.96±6.32|**72.18±5.96**|
> |70|63.46±5.76|65.51±5.69|65.28±5.57|66.93±5.65|65.97±5.96|68.08±5.84|**70.4±5.71**|
> |80|54.62±4.84|56.26±4.98|55.93±5.05|57.15±4.72|56.59±5.13|59.27±4.99|**62.11±5.27**|
> |90|33.25±3.61|38.59±4.06|40.27±3.96|39.45±4.11|41.73±4.5|43.98±4.76|**52.64±4.98**|
>
> **FMNIST**
>
> |Fault ratio (%)|Baseline|SoftSNN|Routing|Astrocyte|FalVolt|LIFA|Proposed|
> |---|---|---|---|---|---|---|---|
> |0|27.4±2.68|27.61±2.75|27.57±2.46|27.43±2.37|**27.77±2.84**|27.59±2.7|26.76±2.36|
> |10|22.66±3.83|24.54±3.96|24.8±3.69|24.96±3.92|25.11±3.63|25.7±3.91|**26.17±3.94**|
> |20|20.84±3.95|22.55±4.12|22.89±4.17|22.74±3.86|22.68±4.09|23.59±4.28|**23.85±4.47**|
> |30|18.81±4.06|20.72±4.27|21.63±4.15|20.98±4.07|21.16±3.96|22.07±3.96|**22.83±4.25**|
> |40|15.47±2.97|16.8±3.11|17.25±3.08|16.97±3.2|17.15±3.19|18.23±3.38|**20.65±3.8**|
> |50|12.35±1.76|13.15±2.57|12.98±2.61|13.41±2.85|13.63±2.79|14.02±2.91|**14.58±3.13**|
> |60|10±0|10±0|10±0|10±0|10±0|10±0|10±0|
> |70|10±0|10±0|10±0|10±0|10±0|10±0|10±0|
> |80|10±0|10±0|10±0|10±0|10±0|10±0|10±0|
> |90|10±0|10±0|10±0|10±0|10±0|10±0|10±0|
>
> Our mechanism also exhibits the best fault mitigation ability with Deihl&Cook2015. This is because our mechanism prevents neurons from over- or under-firing caused by the fault-induced abnormal membrane potential, as mentioned in our response to weakness 3. We add these experimental results in Subsection B.7 of the revised manuscript.

---

> ### Author Response · Authors · 2025-11-21
> **Rebuttal for Reviewer Q1C8 (9/9)**
>
> Furthermore, we adopt our mechanism to STDP-based SNNs implemented on the FPGA device using VHDL codes in https://github.com/rafamedina97/FPGA_SNN_STDP [24]. The size of the model is 784-20-10 (input-hidden-output), and other settings are the same as Deihl&Cook2015. The following table presents the classification accuracy (%) of the STDP-based SNN model on our FPGA device using the baseline, benchmarks, and proposed mechanism under SAFs with MNIST and FMNIST.
>
> **MNIST**
>
> | Fault ratio (%) | Baseline    | SoftSNN     | Routing     | Astrocyte   | FalVolt     | LIFA        | Proposed     |
> |-----------------|-------------|-------------|-------------|-------------|-------------|-------------|-------------|
> | 0               | 82.18±2.38  | 82.75±2.46  | **83.66±2.52** | 83.27±2.29  | 83.79±2.51  | 83.54±2.48  | 83.64±2.6   |
> | 10              | 75.49±3.45  | 77.25±3.58  | 78.59±3.74  | 78.62±3.94  | 78.92±3.85  | 78.77±3.61  | **79.29±3.79** |
> | 20              | 72.7±3.93   | 72.96±4.01  | 74.81±4.36  | 74.75±3.87  | 75.01±4.2   | 75.05±4.19  | **76.38±4.28** |
> | 30              | 69.26±4.26  | 70.16±3.96  | 73.69±4.15  | 73.51±4.42  | 72.99±4.39  | 73.68±4.35  | **74.53±4.24** |
> | 40              | 61.54±4.81  | 63.48±4.45  | 64.04±4.77  | 64.29±4.65  | 65.53±4.84  | 64.99±4.64  | **67.46±5.15** |
> | 50              | 26.33±2.87  | 29.29±3.14  | 30.8±2.9    | 31.38±2.94  | 31.46±3.27  | 31.54±2.86  | **38.91±3.87** |
> | 60              | 12.61±1.56  | 15.87±2.29  | 14.06±2.38  | 15.14±2.16  | 15.48±2.74  | 14.19±2.55  | **18.25±3.16** |
> | 70              | 10±0        | 10±0        | 10±0        | 10±0        | 10±0        | 10±0        | **12.78±1.83** |
> | 80              | 10±0        | 10±0        | 10±0        | 10±0        | 10±0        | 10±0        | 10±0|
> | 90              | 10±0        | 10±0        | 10±0        | 10±0        | 10±0        | 10±0        | 10±0|
>
> **FMNIST**
>
> | Fault ratio (%) | Baseline     | SoftSNN     | Routing      | Astrocyte    | FalVolt     | LIFA        | Proposed     |
> |------------------|-------------|-------------|--------------|--------------|-------------|-------------|-------------|
> | 0                | 25.1±2.93   | 25.02±2.81  | 25.83±2.49 | 25.29±2.91  | 25.34±2.64  | **26.19±2.56** | 25.61±2.38 |
> | 10               | 16.54±3.48  | 18.35±3.62  | 18.61±3.80   | 17.96±3.77   | 18.57±3.49  | 18.81±3.65  | **21.55±3.71** |
> | 20               | 11.25±1.25  | 12.16±1.85  | 12.72±2.01   | 11.58±1.48   | 11.96±1.32  | 12.39±1.96  | **15.08±2.26** |
> | 30               | 10±0        | 10±0        | 10±0 | 10±0        | 10±0        | 10±0        | **11.91±1.08** |
> | 40               | 10±0        | 10±0        | 10±0         | 10±0         | 10±0        | 10±0        | **10.83±0.83** |
> | 50               | 10±0        | 10±0        | 10±0         | 10±0         | 10±0        | 10±0        | 10±0        |
> | 60               | 10±0        | 10±0        | 10±0         | 10±0         | 10±0        | 10±0        | 10±0        |
> | 70               | 10±0        | 10±0        | 10±0         | 10±0         | 10±0        | 10±0        | 10±0        |
> | 80               | 10±0        | 10±0        | 10±0         | 10±0         | 10±0        | 10±0        | 10±0        |
> | 90               | 10±0        | 10±0        | 10±0         | 10±0         | 10±0        | 10±0        | 10±0        |
>
> As shown in the tables, our mechanism successfully improves the fault tolerance of the hardware-implemented SNN model with STDP.

---

> ### Author Response · Authors · 2025-11-21
> **References for rebuttal (1/2)**
>
> # References
>
> [1] Eslami, M. R., Biswas, D., Takhtardeshir, S., Sharif, S. S., & Banad, Y. M. (2024). On-Chip Learning with Memristor-Based Neural Networks: Assessing Accuracy and Efficiency Under Device Variations, Conductance Errors, and Input Noise. arXiv:2408.14680 [cs.NE]
>
> [2] Martemucci, M., Rummens, F., Malot, Y. et al. A ferroelectric–memristor memory for both training and inference. Nat Electron 8, 921–933 (2025). https://doi.org/10.1038/s41928-025-01454-7
>
> [3] Rostami, A., Vogginger, B., Yan, Y., & Mayr, C. G. (2022). “E-prop on SpiNNaker 2: Exploring online learning in spiking RNNs on neuromorphic hardware.” Frontiers in Neuroscience, 16. https://doi.org/10.3389/fnins.2022.1018006
>
> [4] Lagorce, Xavier; Stromatias, Evangelos; Galluppi, Francesco; Plana, Luis A.; Liu, Shih-Chii; Furber, Steve B.; Benosman, Ryad B. “Breaking the millisecond barrier on SpiNNaker: implementing asynchronous event-based plastic models with microsecond resolution.” Frontiers in Neuroscience, Vol. 9 (2015): Article 206. DOI: 10.3389/fnins.2015.00206
>
> [5] K. Stewart, G. Orchard, S. B. Shrestha and E. Neftci, "Live Demonstration: On-chip Few-shot Learning with Surrogate Gradient Descent on a Neuromorphic Processor," 2020 2nd IEEE International Conference on Artificial Intelligence Circuits and Systems (AICAS), Genova, Italy, 2020, pp. 128-128, doi: 10.1109/AICAS48895.2020.9073961
>
> [6] B. Cramer,S. Billaudelle,S. Kanya,A. Leibfried,A. Grübl,V. Karasenko,C. Pehle,K. Schreiber,Y. Stradmann,J. Weis,J. Schemmel, & F. Zenke, Surrogate gradients for analog neuromorphic computing, Proc. Natl. Acad. Sci. U.S.A. 119 (4) e2109194119, https://doi.org/10.1073/pnas.2109194119 (2022)
>
> [7] Yin, M., Cui, X., Wei, F., Liu, H., Jiang, Y. and Cui, X. A reconfigurable FPGA-based spiking neural network accelerator. Microelectronics Journal 152, 106377 (2024). https://doi.org/10.1016/j.mejo.2024.106377
>
> [8] M. Payvand, M. E. Fouda, F. Kurdahi, A. M. Eltawil and E. O. Neftci, "On-Chip Error-Triggered Learning of Multi-Layer Memristive Spiking Neural Networks," in IEEE Journal on Emerging and Selected Topics in Circuits and Systems, vol. 10, no. 4, pp. 522-535, Dec. 2020, doi: 10.1109/JETCAS.2020.3040248
>
> [9] Renner, A., Sheldon, F., Zlotnik, A. et al. The backpropagation algorithm implemented on spiking neuromorphic hardware. Nat Commun 15, 9691 (2024). https://doi.org/10.1038/s41467-024-53827-9
>
> [10] Vatajelu, E.-I., Di Natale, G., & Anghel, L. (2019). Special Session: Reliability of Hardware-Implemented Spiking Neural Networks (SNN). In Proceedings of the 2019 IEEE 37th VLSI Test Symposium (VTS) (pp. 1–8). IEEE. https://doi.org/10.1109/VTS.2019.8758653
>
> [11] A. Mishra, A. Das and N. Kandasamy, "Model-Based Approach Towards Correctness Checking of Neuromorphic Computing Systems," 2024 IEEE 29th Pacific Rim International Symposium on Dependable Computing (PRDC), Osaka, Japan, 2024, pp. 11-21, doi: 10.1109/PRDC63035.2024.00015
>
> [12] Eslami, M. R., Biswas, D., Takhtardeshir, S., Sharif, S. S., & Banad, Y. M. (2024). On-Chip Learning with Memristor-Based Neural Networks: Assessing Accuracy and Efficiency Under Device Variations, Conductance Errors, and Input Noise. arXiv preprint arXiv:2408.14680. Available at: https://arxiv.org/abs/2408.14680
>
> [13] Diehl, P. U., & Cook, M. (2015). Unsupervised learning of digit recognition using spike-timing-dependent plasticity. Frontiers in Computational Neuroscience, 9, 99. https://doi.org/10.3389/fncom.2015.00099
>
> [14] Rachmad Vidya Wicaksana Putra, Muhammad Abdullah Hanif, and Muhammad Shafique. 2022. SoftSNN: low-cost fault tolerance for spiking neural network accelerators under soft errors. In Proceedings of the 59th ACM/IEEE Design Automation Conference (DAC '22). Association for Computing Machinery, New York, NY, USA, 151–156. https://doi.org/10.1145/3489517.3530657
>
> [15] Zhuangyu Han, A N M Nafiul Islam, and Abhronil Sengupta. 2023. Astromorphic self-repair of neuromorphic hardware systems. In Proceedings of the Thirty-Seventh AAAI Conference on Artificial Intelligence and Thirty-Fifth Conference on Innovative Applications of Artificial Intelligence and Thirteenth Symposium on Educational Advances in Artificial Intelligence (AAAI'23/IAAI'23/EAAI'23), Vol. 37. AAAI Press, Article 878, 7821–7829. https://doi.org/10.1609/aaai.v37i6.25947
>
> [16] Rastogi, M., Lu, S., Islam, N. and Sengupta, A., 2021. On the self-repair role of astrocytes in STDP enabled unsupervised SNNs. Frontiers in Neuroscience, 14, 603796. https://doi.org/10.3389/fnins.2020.603796

---

> ### Author Response · Authors · 2025-11-21
> **References for rebuttal (2/2)**
>
> [17] Murat Isik. A Survey of Spiking Neural Network Accelerator on FPGA. arXiv preprint arXiv:2307.03910, 2023
>
> [18] Kakani, V.; Li, X.; Cui, X.; Kim, H.; Kim, B.-S.; Kim, H. Implementation of Field-Programmable Gate Array Platform for Object Classification Tasks Using Spike-Based Backpropagated Deep Convolutional Spiking Neural Networks. Micromachines 2023, 14, 1353. https://doi.org/10.3390/mi14071353
>
> [19] Geng S, Wang Z, Liu Z, Zhang M, Zhu X, Dan Y. 2025. Hardware implementation of FPGA-based spiking attention neural network accelerator. PeerJ Computer Science 11:e3077 https://doi.org/10.7717/peerj-cs.3077
>
> [20] S. Panchapakesan, Z. Fang and J. Li, "SyncNN: Evaluating and Accelerating Spiking Neural Networks on FPGAs," 2021 31st International Conference on Field-Programmable Logic and Applications (FPL), Dresden, Germany, 2021, pp. 286-293, doi: 10.1109/FPL53798.2021.00058
>
> [21] A. Carpegna, A. Savino and S. D. Carlo, "Spiker+: A Framework for the Generation of Efficient Spiking Neural Networks FPGA Accelerators for Inference at the Edge," in IEEE Transactions on Emerging Topics in Computing, vol. 13, no. 3, pp. 784-798, July-Sept. 2025, doi: 10.1109/TETC.2024.3511676
>
> [22] Hananel Hazan, Daniel J.Saunders, Hassaan Khan, Devdhar Patel, DarpanT. Sanghavi, HavaT. Siegelmann, and Robert Kozma. Bindsnet: A machine learning-oriented spiking neural networks library in Python. Frontiers in Neuroinformatics, 12, 2018.doi: 10.3389/fninf.2018.00089. URL https://www.frontiersin.org/article/10.3389/fninf.2018.00089
>
> [23] Hyun-Jong Lee and Jae-Han Lim. Adaptive synaptic adjustment mechanism to improve learning performances of spiking neural networks. Computational Intelligence, 40(5):e70001, 2024. doi: https://doi.org/10.1111/coin.70001. URL https://onlinelibrary.wiley.com/doi/abs/10.1111/coin.70001
>
> [24] R. M. Morillas and P. Ituero, "STDP Design Trade-offs for FPGA-Based Spiking Neural Networks," 2020 XXXV Conference on Design of Circuits and Integrated Systems (DCIS), Segovia, Spain, 2020, pp. 1-6, doi: 10.1109/DCIS51330.2020.9268614

---

> ### Comment · Reviewer_Q1C8 · 2025-11-24
>
> Thank you for your response. I appreciate the effort and results; I am changing my score.

---

> ### Author Response · Authors · 2025-11-24
> **Thank you for your comment on our rebuttal!**
>
> We really appreciate raising the soundness score from 2 to 3 and the overall rating from 4 to 8! Indeed, your insightful comments have significantly improved the quality of our paper. Thanks again for your valuable comments and for increasing the score.

---

> ### Public Comment · ~Hyun-Jong_Lee1 · 2026-04-27
> **Reviewer Q1C8's comment about the previous works for fault modeling**
>
> 'Lee & Lim, 2023' is the work of the authors.

---

### Official Review · Reviewer_Nuth · 2025-10-31

**Soundness:** 3
**Presentation:** 4
**Contribution:** 2
**Rating:** 4
**Confidence:** 5

**Summary:**

The paper presents a method for fault tolerance to stuck-at and synaptic faults in spiking neural networks running on neuromorphic hardware using flow control methods based on sensitivity analysis of neurons. The system uses the Gini coefficient to analyse sensitivity and route inputs past problematic nodes that cause gradient collapse due to anomalously large spiking values. This is tested against some other neuromorphic approaches using the Leaky Integrate and Fire method. Overall, the paper presents this as a highly theoretically grounded approach and draws on behavioral models of faults in neuromorphic hardware and tests this in an FPGA.

**Strengths:**

The paper's theoretical section is very solid, in particular the appendix proof that provides theoretical guarantees of the system's optimality. Similarly, it presents the use of flow control as a very intuitive approach, comparing it to flow control in computer networks - this makes a complex idea and approach seem much more digestible. I believe that this combination of complex, thorough theory and solid explanation by analogy is what makes the presentation of this paper a very sound addition to the conference provided its weaknesses are addressed.

Its hardware based experimental section is also very good - the use of real hardware for fault tolerance analysis is a mark of good experimentation.

**Weaknesses:**

I am worried about the paper's experimental section, in particular its contrast to prior art in the hardware space that has modelled Leaky Integrate and Fire neurons in non-FPGA formats. The FPGA is not the only cutting-edge accelerator hardware being examined in the field, and resilience approaches in prior art have also looked at:

1) Setting anomalous values to zero based on neuron output statistics in an inference or training episode [1] or using DropOut [2]. This seems a very lightweight approach, as opposed to requirements of routing and flow control that impose interconnect and communication overheads - manageable in an FPGA but may be harder in analog, compute-in-memory or GPU substrates. How does your approach contrast to this? I would like to see a discussion of that.

2) Persistent faults in the network - which are ideally addressed using flow control - have been addressed using testing based approaches (online and offline self-test in [2]; and a signature-based compact test strategy in [3]) which has the benefit of amortizing test overhead over number of inferences, rather than being an always-on strategy. It would be good to show how the online approaches examined here, applied as they are to persistent faults, contrast with test strategies that may be amortizable over the training process.

[1] A. Saha, C. Amarnath and A. Chatterjee, "A Resilience Framework for Synapse Weight Errors and Firing Threshold Perturbations in RRAM Spiking Neural Networks," 2023 IEEE European Test Symposium (ETS), Venezia, Italy, 2023, pp. 1-4, doi: 10.1109/ETS56758.2023.1017422

[2] T. Spyrou, S. A. El-Sayed, E. Afacan, L. A. Camuñas-Mesa, B. Linares-Barranco and H. -G. Stratigopoulos, "Neuron Fault Tolerance in Spiking Neural Networks," 2021 Design, Automation & Test in Europe Conference & Exhibition (DATE), Grenoble, France, 2021, pp. 743-748, doi: 10.23919/DATE51398.2021.9474081.

[3] A. Saha, C. Amarnath, K. Ma and A. Chatterjee, "Signature Driven Post-Manufacture Testing and Tuning of RRAM Spiking Neural Networks for Yield Recovery," 2024 29th Asia and South Pacific Design Automation Conference (ASP-DAC), Incheon, Korea, Republic of, 2024, pp. 740-745, doi: 10.1109/ASP-DAC58780.2024.10473874.

**Questions:**

1) Could the authors provide a discussion w.r.t. prior art in the hardware space? A few papers are cited above, but the approaches involving forward-pass resilience seem to be lower overhead than flow-control approaches, especially when applied to non-FPGA hardware.

2) Could the authors provide a short discussion contrasting these on-line approaches with offline or online periodic self-test and repair systems in hardware that would allow for resilience overhead to be amortized over a larger number of computations, at the cost of potentially allowing some faults through? What are the pros and cons, and the application domains, of each approach?

---

> ### Author Response · Authors · 2025-11-17
> **Rebuttal for Reviewer Nuth (1/3)**
>
> # Thanks to Reviewer Nuth
>
> We really appreciate your comments on the discussion of the proposed mechanism with previous lightweight methods to improve the fault tolerance of hardware-implemented SNNs. Your recommendation to add the discussion on these lightweight approaches improves our paper’s concreteness against comparison with other simple approaches.  We have made a new Section A named “Discussion” to present valuable discussions that you recommend in Appendix. Please refer to Section A in Appendix of the revised manuscript.
>
> ---
>
> # Responses to the weaknesses
>
> **Your comment - '~prior art in the hardware space that has modelled Leaky Integrate and Fire neurons in non-FPGA formats.'**: We really appreciate your insightful comment regarding the hardware studies modeling LIF neurons on non-FPGA platforms. **Our comparison remains valid because the prior works do not rely on device-specific characteristics, allowing their algorithms to be faithfully reproduced on general digital substrates such as an FPGA. Importantly, our mechanism is hardware-agnostic, as it modifies only the input stream and does not interact with synaptic weights, routing, or membrane-potential dynamics. Therefore, an FPGA serves as an appropriate and representative testbed for our approach. Additionally, FPGA-based SNNs are widely used in recent studies due to their reconfigurability and suitability for rapid hardware-level prototyping [1, 2].**
>
> **W1**: We really appreciate your comment about the comparison to the 0-replacement and dropout methods [3, 4] (prior works [1, 2] in your comment). **First, we would like to notify you that our mechanism does not use any routing methods or modifications to internal components in neuromorphic devices. Instead, we only control the input streams without methods to modify internal components to enhance the fault tolerance of neuromorphic devices.** To address your concern about the comparison to other lightweight approaches, we have thoroughly read the mentioned papers and pointed out the different points from our mechanism.
>
> - **Ref [3]**: This work proposes a simple hardware-level fault mitigation approach by monitoring internal currents and firing thresholds, detecting abnormal inputs caused by perturbations in LIF neurons’ thresholds, and preventing the neurons from obtaining abnormal input spikes. This mechanism requires the entire probe to monitor the errors in synaptic weights and direct weight or circuit modification to block the abnormal input spikes (setting the weights along abnormal input spikes to zero). The authors use small SNN models such as LeNet-5 and VGG-5 with simple datasets (MNIST and FMNIST). In these scenarios, the overhead of scanning the entire weight matrix in the models does not increase significantly. **However, the complicated current SNN models, such as spiking VGG-7/11/15 and ResNet-18/34, contain large amounts of weights, and the overhead of scanning the entire weights in the hardware version of these models increases dramatically [1, 2, 5].** Furthermore, this mechanism removes the abnormal pre-activation values that are severely affected by faults. This elimination causes data loss during the forward pass.
>
> - **Ref [4]**: This work presents a dropout-based method that improves the fault tolerance of hardware-implemented SNNs by ignoring neurons that emit abnormal output spikes during the forward pass in a lightweight manner. To detect the abnormal neurons to ignore, they should monitor the output spikes from neurons during the online self-test. **This monitoring creates additional burdens on neuromorphic devices [6, 7].** In addition, ignoring abnormal neurons demands an external method that forcibly modifies the weights or circuits of abnormal neurons to 0, although modifying the synaptic weights or circuits in neuromorphic devices is difficult [8].
>
> These methods enhance the fault mitigation ability of inference-based neuromorphic devices with lightweight methods. However, they increase the burden on neuromorphic devices when an SNN model’s size is large because there are lots of neurons to monitor status changes in real time and weights to modify with direct access to hardware circuits. Unlike the methods in Refs. 3 and 4, our mechanism simply divides the input data into small fragments instead of monitoring the entire neurons’ dynamicity in real time and forcibly modifying the synaptic weights or circuits of hardware-implemented SNNs. Moreover, the communication overhead between the In-and-Out (IO) module for the external processor that invokes our mechanism and analog compute-in-memory devices or a GPU is small, not increasing the computational complexity of the integrated system [9, 10, 11]. This is because our mechanism does not demand to receive large data, such as entire weight values or neuron status, like [3, 4]. Instead, it only needs a sensitivity score derived from the information from input data and the weight values of only a single layer.

---

> ### Author Response · Authors · 2025-11-17
> **Rebuttal for Reviewer Nuth (2/3)**
>
> **W2**: We really appreciate your comment that suggests comparing our mechanism to test-based approaches. To address your comment, we provide a comparison between our mechanism and the test-based approaches in detail.
>
> - Test-based approaches: **These approaches do not always invoke testing to diagnose faults of hardware-implemented SNNs in neuromorphic devices.** As you mentioned, they amortize the overhead of testing on inferences that are executed repeatedly, and this amortization reduces their computational burden on neuromorphic devices [4, 12] ([12] is the prior work [3] in your comment.). This is because they do not always operate in real-time data processing, and this characteristic prevents them from dealing with permanent faults that frequently occur due to aging, thermal noise, and external shocks during inferences [13].
>
> - Our mechanism: **The proposed mechanism always divides input data into small fragments in real time during training and inference.** Unlike the test-based approaches, it cannot amortize the overhead on repeated inferences. Despite this continual execution, its computational cost over a single execution is small (please refer to Appendix C of the revised manuscript). Additionally, our mechanism adjusts input flow according to the current status of hardware faults in agile, unlike the test-based approach with a signature, which only tests the chip once before inference [11]. This advantage makes our mechanism exhibit more significant fault mitigation ability than test-based approaches.
>
> In summary, test-based approaches do not incur additional computational cost over time, since they do not always run tests in real time during inference. This point makes them lightweight and reduces computational burden on the neuromorphic devices that mainly perform inferences. However, they cannot simultaneously respond to permanent fault occurrence during inference. On the other hand, our mechanism calculates the sensitivity score and generates data fragments in real time. Nevertheless, our mechanism does not increase the overall computational burden significantly since it has a small computational complexity. Moreover, by controlling input flow according to the current status of hardware faults, our mechanism mitigates the adversarial effects of faults in agile environments, improving the fault tolerance more significantly than test-based approaches.
>
> ---
>
> # Response to the questions
>
> **Q1**: We really appreciate your question on the detailed discussion with the SOTA works from a hardware perspective. Here, we provide a discussion on the forward-pass fault resilience mechanisms and our mechanisms in detail.
>
> - Forward-pass resilience mechanisms: The prior works that you have mentioned indeed have little overhead on small SNN models. Despite this advantage, these works have a limitation. They do not consider large SNN models, such as VGG-19, ResNet-18, and ResNet-34. Researchers have implemented large models in hardware [14]. **The computational overhead of the forward-pass resilience methods increases linearly with the size of SNN models.** This is because the mentioned methods monitor entire synaptic weights or neurons, and the large-sized SNN models contain numerous synaptic weights and neurons. Furthermore, the overhead of aggregating the monitored results of the weights or neurons and determining which elements to ignore increases linearly when the SNN model is large [15, 16].
>
> - Our mechanism: **In contrast, our mechanism has constant computational costs regardless of the hardware model’s size. It also has a small overhead for interconnection between a hardware-implemented SNN model and the external processor.** Because the actual overhead for communication between the hardware SNN model and the external processor is small [9, 10, 11]. Furthermore, our mechanism does not require large metadata of hardware SNNs, such as entire weight values or states of neurons in hardware SNNs, for scanning and diagnosis. This advantage of our mechanism makes it suitable for modern neuromorphic devices that usually use large-sized SNN models, making our mechanism practical.
>
> In other words, forward-pass resilience mechanisms have small overheads on simple SNN models. However, their overheads increase significantly as the size of hardware-implemented SNN models grows. On the other hand, our mechanism has constant overheads regardless of the size of hardware-implemented SNN models. Owing to this point, our mechanism fits modern hardware-implemented SNNs, becoming pragmatic for neuromorphic devices.

---

> ### Author Response · Authors · 2025-11-17
> **Rebuttal for Reviewer Nuth (3/3)**
>
> **Q2**: We really appreciate your helpful question regarding how our online input-fragmentation mechanism compares to periodic self-test and repair systems in neuromorphic hardware.  Below, we present the key differences in terms of advantages, limitations, and application domains.
>
> - Self-test and repair systems: **Periodic self-test and repair approaches amortize diagnostic cost across many computations, which makes them computationally efficient and suitable for hardware where inference workloads dominate [17].** They are a good lightweight solution because they amortize the execution burden over inferences. **Their main limitation is responsiveness. This is because testing occurs only at scheduled intervals, which means faults that emerge between test cycles may temporarily affect computation, and performing such tests more frequently can impose significant overhead [4, 12].**  These approaches therefore fit devices that use fixed, pre-trained models and applications that tolerate intermittent performance degradation, such as Speck and Xylo of SynSense (https://www.synsense.ai/products/speck-2/).
>
> - Our mechanism: **In contrast, our mechanism operates continuously during training and inference by generating input fragments online without accessing internal hardware states. This enables immediate mitigation of fault-induced adversarial effects as they manifest in the SNNs' outputs, supporting more stable real-time behavior.** Although this continual operation incurs a higher per-inference computational cost than periodic testing, it provides rapid adaptation to hardware faults in agile environments. This point makes our mechanism well-suited for neuromorphic systems, such as SpiNNaker (https://www.humanbrainproject.eu/en/collaborate-hbp/innovation-industry/technology-catalogue/spinnaker/) and memristor-based devices, that perform on-chip learning or require strict real-time performance (adaptive control or continual learning at edge devices) [18, 19].
>
> In summary, periodic self-test/repair systems offer low overhead but limited responsiveness due to their intermittent nature [4, 12]. On the other hand, our mechanism incurs higher per-inference computation but delivers immediate, ongoing fault mitigation suitable for dynamic, real-time neuromorphic applications that require high agility.

---

> ### Author Response · Authors · 2025-11-17
> **Experimental results for comparing our mechanism to other lightweight approaches**
>
> To demonstrate that our mechanism has a smaller overhead than the lightweight approaches, such as dropout and test-based methods, we conduct experiments to measure the classification accuracy (%) and inference time (sec) of the models with our mechanism and other lightweight benchmarks. The following tables present the accuracy and summation of inference time over 100 iterations (assuming the actual scenarios and using samples in the test dataset for each inference iteration) of the SNN models using the lightweight approaches and the proposed mechanism under a 20% fault ratio of SAFs. Please note that the prior lightweight works [3, 4, 12] in our rebuttal are the works [1, 2, 3] in your comment.
>
> **Accuracy in software-based SNN models**
>
> | Datasets (models) | Input suppression [3] | Fault hopping [4] | Threshold tuning  [12]|         Proposed |
> | ----------------- | ----------------: | ------------: | ---------------: | ---------------: |
> | MNIST (MLP)       |         88.38 ± 1.15 |     86.41 ± 0.98 |        87.23 ± 1.52 | **93.79 ± 1.06** |
> | FMNIST (MLP)      |            80.45 ± 1.22 |      79.86 ± 1.16 |   78.98 ± 1.39 | **85.47 ± 0.92** |
> | CIFAR-10 (VGG-7)  |      34.67 ± 3.21 |  30.28 ± 3.34 |   27.91 ± 3.26 | **45.94 ± 3.17** |
>
> **Accuracy in FPGA-implemented SNNs**
>
> | Datasets (models) | Input suppression [3] | Fault hopping [4] | Threshold tuning [12] |         Proposed |
> | ----------------- | ----------------: | ------------: | ---------------: | ---------------: |
> | MNIST (MLP)       |        86.7 ± 1.32 |     84.38 ± 1.28 |        85.87 ± 1.65 | **91.43 ± 1.17** |
> | FMNIST (MLP)      |       78.46 ± 0.98 |   75.39 ± 1.14 |        73.63 ± 1.27 |   **84.05 ± 1.09** |
> | CIFAR-10 (VGG-7)  |     31.94 ± 3.8 |  27.14 ± 3.72 |     24.75 ± 2.87 | **38.11 ± 3.58** |
>
> **Inference time in software-based SNN models**
>
> | Datasets (models) | Input suppression [3] | Fault hopping [4] | Threshold tuning [12] |          Proposed |
> | ----------------- | ----------------: | ------------: | ---------------: | ----------------: |
> | MNIST (MLP)       |     283.64 ± 4.85 |  254.43 ± 2.1 |    213.67 ± 2.38 | **191.28 ± 1.25** |
> | FMNIST (MLP)      |     285.27 ± 5.17 | 255.79 ± 1.98 |    215.09 ± 2.51 | **193.54 ± 1.62** |
> | CIFAR-10 (VGG-7)  |     500.59 ± 7.34 | 327.11 ± 3.83 |    292.41 ± 4.86 | **277.09 ± 3.47** |
>
> **Inference time in FPGA-implemented SNN models**
>
> | Datasets (models) | Input suppression [3] | Fault hopping [4] | Threshold tuning [12] |         Proposed |
> | ----------------- | ----------------: | ------------: | ---------------: | ---------------: |
> | MNIST (MLP)       |      80.26 ± 1.06 |  62.35 ± 0.93 |     50.39 ± 0.61 | **45.26 ± 0.73** |
> | FMNIST (MLP)      |      81.38 ± 1.19 |  64.84 ± 1.25 |     51.08 ± 0.67 | **46.38 ± 0.76** |
> | CIFAR-10 (VGG-7)  |     124.22 ± 2.42 | 101.57 ± 2.61 |      78.53 ± 1.3 | **68.91 ± 1.05** |
>
> As shown in the tables, the models with our mechanism classify data samples most accurately under SAFs, consuming the least inference time among the models with the four fault mitigation methods. Our experimental results demonstrate that our mechanism incurs no greater overhead than previous lightweight approaches while significantly enhancing the fault tolerance of SNN models. We have added these experimental results in Section A of the revised manuscript. We have also added the prior works that you mentioned in Subsection 3.2 of the revised manuscript, presenting SOTA works. Please refer to Subsection 3.2 and Section A of Appendix in the revised manuscript.

---

> ### Author Response · Authors · 2025-11-25
> **References for rebuttal (1/2)**
>
> # References
>
> [1] Mehrzad Karamimanesh, Ebrahim Abiri, Mahyar Shahsavari, Kourosh Hassanli, André van Schaik, and Jason Eshraghian, “Spiking neural networks on FPGA: A survey of methodologies and recent advancements,” Neural Networks, vol. 186, p. 107256, 2025, doi: 10.1016/j.neunet.2025.107256
>
> [2] A. Carpegna, A. Savino and S. D. Carlo, "Spiker+: A Framework for the Generation of Efficient Spiking Neural Networks FPGA Accelerators for Inference at the Edge," in IEEE Transactions on Emerging Topics in Computing, vol. 13, no. 3, pp. 784-798, July-Sept. 2025, doi: 10.1109/TETC.2024.3511676
>
> [3] A. Saha, C. Amarnath and A. Chatterjee, "A Resilience Framework for Synapse Weight Errors and Firing Threshold Perturbations in RRAM Spiking Neural Networks," 2023 IEEE European Test Symposium (ETS), Venezia, Italy, 2023, pp. 1-4, doi: 10.1109/ETS56758.2023.1017422
>
> [4] T. Spyrou, S. A. El-Sayed, E. Afacan, L. A. Camuñas-Mesa, B. Linares-Barranco and H. -G. Stratigopoulos, "Neuron Fault Tolerance in Spiking Neural Networks," 2021 Design, Automation & Test in Europe Conference & Exhibition (DATE), Grenoble, France, 2021, pp. 743-748, doi: 10.23919/DATE51398.2021.9474081
>
> [5] Wei Fang, Yanqi Chen, Jianhao Ding, Zhaofei Yu, Timothée Masquelier, Ding Chen, Liwei Huang, Huihui Zhou, Guoqi Li, and Yonghong Tian. “SpikingJelly: An open-source machine learning infrastructure platform for spike-based intelligence.” Science Advances, vol. 9, no. 40, eadi1480, 2023. doi:10.1126/sciadv.adi1480
>
> [6] A. K. Mishra, A. Das and N. Kandasamy, "Online Performance Monitoring of Neuromorphic Computing Systems," 2023 IEEE European Test Symposium (ETS), Venezia, Italy, 2023, pp. 1-4, doi: 10.1109/ETS56758.2023.10173860
>
> [7] PatSnap. “How to Verify and Debug Neuromorphic Hardware Systems.” PatSnap Eureka Technical Report, 3 September 2025. Available at: https://eureka.patsnap.com/report-how-to-verify-and-debug-neuromorphic-hardware-systems (accessed 15 November 2025)
>
> [8] Luíza C. Garaffa, Abdullah Aljuffri, Cezar Reinbrecht, Said Hamdioui, Mottaqiallah Taouil, and Johanna Sepulveda. “Revealing the Secrets of Spiking Neural Networks: The Case of Izhikevich Neuron.” In Proceedings of the 2021 24th Euromicro Conference on Digital System Design (DSD), pp. 514–518, 2021. doi:10.1109/DSD53832.2021.00083
>
> [9] Khan, Asif Ali; Lima, João Paulo C. de; Farzaneh, Hamid; Castrillón, Jeronimo. “The Landscape of Compute-near-memory and Compute-in-memory: A Research and Commercial Overview.” arXiv preprint arXiv:2401.14428 (2024). Available at: https://arxiv.org/abs/2401.14428
>
> [10] Yiran Chen, Yuan Xie, Linghao Song, Fan Chen, Tianqi Tang, A Survey of Accelerator Architectures for Deep Neural Networks, Engineering, Volume 6, Issue 3, 2020, Pages 264-274, ISSN 2095-8099, https://doi.org/10.1016/j.eng.2020.01.007
>
> [11] Lagorce, Xavier; Stromatias, Evangelos; Galluppi, Francesco; Plana, Luis A.; Liu, Shih-Chii; Furber, Steve B.; Benosman, Ryad B. “Breaking the millisecond barrier on SpiNNaker: implementing asynchronous event-based plastic models with microsecond resolution.” Frontiers in Neuroscience, Vol. 9 (2015): Article 206. DOI: 10.3389/fnins.2015.00206
>
> [12] A. Saha, C. Amarnath, K. Ma and A. Chatterjee, "Signature Driven Post-Manufacture Testing and Tuning of RRAM Spiking Neural Networks for Yield Recovery," 2024 29th Asia and South Pacific Design Automation Conference (ASP-DAC), Incheon, Korea, Republic of, 2024, pp. 740-745, doi: 10.1109/ASP-DAC58780.2024.10473874
>
> [13] Vatajelu, E.-I., Di Natale, G., & Anghel, L. (2019). Special Session: Reliability of Hardware-Implemented Spiking Neural Networks (SNN). In Proceedings of the 2019 IEEE 37th VLSI Test Symposium (VTS) (pp. 1–8). IEEE. https://doi.org/10.1109/VTS.2019.8758653
>
> [14] M. T. L. Aung et al., "DeepFire2: A Convolutional Spiking Neural Network Accelerator on FPGAs," in IEEE Transactions on Computers, vol. 72, no. 10, pp. 2847-2857, Oct. 2023, doi: 10.1109/TC.2023.3272284
>
> [15] Rachmad Vidya Wicaksana Putra, Muhammad Abdullah Hanif, and Muhammad Shafique. Rescuesnn: enabling reliable executions on spiking neural network accelerators under permanent faults. Frontiers in Neuroscience, 17, 2023. ISSN1662-453X. doi: 10.3389/fnins.2023.1159440. URL https://www.frontiersin.org/journals/neuroscience/articles/10.3389/nins.2023.1159440
>
> [16] K. Zhao et al., "FT-CNN: Algorithm-Based Fault Tolerance for Convolutional Neural Networks," in IEEE Transactions on Parallel and Distributed Systems, vol. 32, no. 7, pp. 1677-1689, 1 July 2021, doi: 10.1109/TPDS.2020.3043449
>
> [17] F. Meng, F. S. Hosseini, and C. Yang, “A Self-Test Framework for Detecting Fault-Induced Accuracy Drop in Neural Network Accelerators,” in Proceedings of the 26th Asia and South Pacific Design Automation Conference (ASP-DAC ’21), Tokyo, Japan, 2021, pp. 722–727

---

> ### Author Response · Authors · 2025-11-25
> **References for the rebuttal (2/2)**
>
> [18] Rostami, A., Vogginger, B., Yan, Y., & Mayr, C. G. (2022). “E-prop on SpiNNaker 2: Exploring online learning in spiking RNNs on neuromorphic hardware.” Frontiers in Neuroscience, 16. https://doi.org/10.3389/fnins.2022.1018006
>
> [19] Eslami, M. R., Biswas, D., Takhtardeshir, S., Sharif, S. S., & Banad, Y. M. (2024). “On-Chip Learning with Memristor-Based Neural Networks: Assessing Accuracy and Efficiency Under Device Variations, Conductance Errors, and Input Noise.” arXiv:2408.14680. https://arxiv.org/abs/2408.14680

---

> ### Author Response · Authors · 2025-11-25
> **Gentle reminder**
>
> Dear Reviewer Nuth,
>
> We have compared our mechanism to the lightweight approaches that you mentioned. Additionally, we have posted a thorough discussion about our mechanism and the mentioned lightweight approaches in our rebuttal. If you have any further questions or suggestions, we would be happy to clarify them. Thank you again for your time and consideration

---

> ### Author Response · Authors · 2025-11-25
>
> Dear, AC
>
> For Reviewer Nuth, who has not yet responded, we would greatly appreciate it if you could carefully consider our rebuttal and revisions and, where appropriate, infer their likely reaction when making your decision.

---

### Official Review · Reviewer_E666 · 2025-11-08

**Soundness:** 3
**Presentation:** 2
**Contribution:** 3
**Rating:** 4
**Confidence:** 2

**Summary:**

This paper proposed an input fragmentation mechanism inspired by flow control in computer networks. It tackles the important issue of fault tolerance in Spiking Neural Networks (SNNs).

**Strengths:**

The main strength of this paper is that they have done various experiments on several models.

**Weaknesses:**

The main weakness is that the idea presentation could be clearer.

There could be more information for SAFs and RWFs since they are important characteristics in fault tolerant research.

Line 167-170, this paragraph has a lot of parameters with neither explanation nor citations.

Equations (1)–(4) are very complicated, which needs a more sufficient explanation or derivative. If they are not completely proposed by the authors, some citations would be better.

Datasets could be used with citations.

**Questions:**

What is the Gini coefficient? If it is proposed by the authors, could they add some introductions for this?

---

> ### Author Response · Authors · 2025-11-12
> **Rebuttal for Reviewer E666**
>
> # Thanks to Reviewer E666
>
> We appreciate your thorough comments on the presentation clarity of our paper. Indeed, your insightful comments help us a lot to improve the presentation quality of our paper, especially in equations.
>
> ---
>
> # Responses to the weaknesses
>
> **W1**: We really appreciate your valuable comment. The main idea of our paper is **'We improve hardware-implemented SNNs’ fault tolerance under synaptic faults by splitting inputs into fragments by dividing input samples with the angles that optimally ensure inhibition of the pre-activation drift.'** To address your comment, we have written the main idea of our paper in Section 5 of the revised manuscript. Please refer to the first paragraph of Section 5 in the revised manuscript.
>
> **W2**: We really appreciate your valuable comment on the description of fault models. SAFs are faults that make synapses permanently stuck at a max or min weight, regardless of training or input [1]. RWFs are faults that make synapses randomly fluctuate around their original weight values due to thermal noise [2]. To address your comment, we have strengthened the explanation of SAFs and RWFs in Subsection 2.2 of the revised manuscript. Please refer to the first paragraph of Subsection 2.2 in the revised manuscript.
>
> **W3**: We really appreciate your valuable comment on the clarity of equations in Subsection 4.1. To address your comment, we replace the complicated equations using lots of parameters with a representative equation of pre-activation $z$: the linear combination of inputs in linear layers. Here, the pre-activation $z$ is a linear combination of inputs using citation, $W$ is the synaptic weights, $x$ is an input sample, and $b$ is a bias. $\Delta W$ indicates a weight change caused by hardware faults. We explain these parameters in item 1 of Subsection 4.1 of the revised manuscript. Please refer to the first paragraph in Subsection 4.1 of the revised manuscript.
>
> **W4**: We really appreciate your comment on the core equations of the proposed mechanism. To address your concern, we have written additional explanations about the equations. The following items show the additional information that we have added for the equations.
>
> - Eq 1: $SP$ identifies the input pixels whose perturbations cause large shifts in pre-activations and, consequently, substantial changes in the final output. $\odot$ indicates the Hadamard product. $PN$ is a percentile normalization, which normalizes the values of the fault influence map in the range of 0 to 1. We have added the explanation about saliency $SP$, Hadamard product $\odot$, and percentile normalization $PN$ in Subsection 5.2 of the revised manuscript.
>
> - Eq 2: This equation explains the Gini coefficient [3]. We have removed complex mathematical expressions from the explanations in Subsection 5.3 of the revised manuscript and described the components of equations more easily than in the original version.
>
> - Eq 3: $\tilde x_t=g_t x_t,\quad \|\tilde x_t\|_2=\alpha\;\Rightarrow\; |z_t|\le \|\hat w\|_2\\alpha $ (changed version). This equation explains RMS normalization [4]. We have explained the mathematical components of the equations thoroughly and described the core effect of RMS normalization in our mechanism in Subsection 5.4 of the revised manuscript.
>
> - Eq 4: $\bar\ell=\sum_{t=1}^T e_t\,\ell_t,\qquad e_t \=\ \frac{\exp\\big(-\tau H(\mathrm{softmax}(\ell_t)\big)}{\sum_{s=1}^{T} \exp\\big(-\tau H(\mathrm{softmax}(\ell_s)\big)}$ (changed version). This equation explains entropy-based decoding [5]. We have changed the equation for $e_t$ and provided the additional explanations for mathematical components such as $\bar \ell$ and $s$ in Subsection 5.4 of the revised manuscript.
>
> For Eqs. 2, 3, and 4, we have also added references that provide detailed descriptions of the equations. We have written additional explanations to provide information of the equations of Section 5 in the revised manuscript. Please refer to the paragraphs under the equations of Section 5 in the revised manuscript.
>
> **W5**: We really appreciate your valuable comment. To address your concern, we added the citations for the datasets in Subsection 6.1 of the revised manuscript [6, 7, 8, 9, 10, 11]. Please refer to the head of Subsection 6.1 in the revised manuscript.
>
> ---
>
> # Responses to the question
>
> **Q1**: We really appreciate your comment on the Gini coefficient. The Gini coefficient, defined by Corrado Gini, measures inequality in a distribution (most often income or wealth in Economics) [3]. It ranges from 0 to 1 (0 = perfect equality and 1 = one unit holds everything). We use Gini coefficient to ensure that each fragment has an equal sensitivity score value. To address your concern, we added a citation that explains the Gini coefficient in detail. Please refer to the head of Subsection 5.2 in the revised manuscript.

---

> ### Author Response · Authors · 2025-11-18
> **References for the rebuttal**
>
> # References
>
> [1] Lee, H.-J., & Lim, J.-H. (2023). Analysis on Effects of Fault Elements in Memristive Neuromorphic Systems. In Proceedings of the IJCAI 2023 GLOW Workshop. arXiv preprint arXiv:2312.04840. Retrieved from https://arxiv.org/abs/2312.04840
>
> [2] Vatajelu, E.-I., Di Natale, G., & Anghel, L. (2019). Special Session: Reliability of Hardware-Implemented Spiking Neural Networks (SNN). In Proceedings of the 2019 IEEE 37th VLSI Test Symposium (VTS) (pp. 1–8). IEEE. https://doi.org/10.1109/VTS.2019.8758653
>
> [3] Frank A. Farris. “The Gini Index and Measures of Inequality.” The American Mathematical Monthly, vol. 117, no. 10, 2010, pp. 851–64. JSTOR, https://doi.org/10.4169/000298910x523344. Accessed 12 Nov. 2025
>
> [4] Zhang, B. & Sennrich, R. (2019). Root Mean Square Layer Normalization. In Advances in Neural Information Processing Systems 32 (pp. 12360–12371). DOI: 10.5555/3454287. (NeurIPS 2019)
>
> [5] Qiu, Z., Ou, Z., Wu, B., Li, J., Liu, A. & King, I. (2025). Entropy-Based Decoding for Retrieval-Augmented Large Language Models. In Proceedings of the 2025 Conference of the North American Chapter of the Association for Computational Linguistics (NAACL), Long Papers, Albuquerque, New Mexico, pp. 4616–4627. DOI: 10.18653/v1/2025.naacl-long.236
>
> [6] LeCun, Y., Bottou, L., Bengio, Y. & Haffner, P. (1998). Gradient-Based Learning Applied to Document Recognition. Proceedings of the IEEE, 86(11), 2278–2324. DOI: 10.1109/5.726791
>
> [7] Xiao H., Rasul K. & Vollgraf R. (2017). Fashion-MNIST: A Novel Image Dataset for Benchmarking Machine Learning Algorithms. arXiv preprint arXiv:1708.07747. Available at: https://arxiv.org/abs/1708.07747
>
> [8] Krizhevsky, A. (2009). Learning Multiple Layers of Features from Tiny Images. Technical Report, University of Toronto. Available at: https://www.cs.toronto.edu/~kriz/learning-features-2009-TR.pdf
>
> [9] Reyes-Ortiz, J. L., Anguita, D., Ghio, A., Oneto, L. & Parra, X. (2013). Human Activity Recognition Using Smartphones (UCI Machine Learning Repository). DOI: 10.24432/C54S4K
>
> [10] Deng J., Dong W., Socher R., Li, L-J., Li K. & Fei-Fei L. (2015). Tiny ImageNet Challenge. Stanford CS231n course dataset. Available at: http://cs231n.stanford.edu/tiny-imagenet-200.zip
>
> [11] Becker, S., Vielhaben, J., Ackermann, M., Müller, K.-R., Lapuschkin, S. & Samek, W. (2024). AudioMNIST: Exploring Explainable Artificial Intelligence for audio analysis on a simple benchmark. Journal of the Franklin Institute, 361(1), 418–428. DOI: 10.1016/j.jfranklin.2023.11.038

---

> ### Author Response · Authors · 2025-11-24
> **Gentle reminder**
>
> Dear Reviewer E666,
>
> We thank you for your detailed comments and would be happy to further clarify any remaining concerns. In particular, we believe we have addressed the main points as follows. First, we clarify our explanation about the main point and fault models. Second, we supplement the additional descriptions to explain the equations for our mechanism. Please let us know if there are further comments that would be most helpful for your evaluation.

---

> ### Author Response · Authors · 2025-12-04
>
> Dear, AC
>
> For Reviewer E666, who has not yet responded, we would greatly appreciate it if you could carefully consider our rebuttal and revisions and, where appropriate, infer their likely reaction when making your decision.

---

### Author Response · Authors · 2025-11-12
**Comment for the new AC (2/2)**

## Reviewer Q1C8

**Main concerns: the environmental assumption of hardware learning, reproducibility of the FPGA-based experimental settings, and our mechanism's effectiveness in other various settings in hardware**

To address concerns of reviewer Q1C8, we collected prior works, conducted various additional experiments, and addressed the reviewer's concerns successfully.

- We demonstrated that our assumption, that the gradient-based neuromorphic devices support on-chip learning, is valid.

- We presented that our mechanism enhances the fault tolerance of hardware-implemented SNNs against faults caused by components in FPGA devices.

- We demonstrated that our mechanism enhances the fault tolerance of SNNs using unsupervised learning, such as STDP, by applying our mechanism to STDP-based SNNs and conducting experiments. This is important since neuromorphic devices that utilize STDP for on-chip learning are widely used, as the reviewer mentioned.

- We updated our supplementary material to contain VHDL scripts to reproduce our FPGA-based experimental results.

- We built deep convolution SNN models and adopted our mechanism to these hardware-implemented SNN models. Then, we experimentally presented that our mechanism successfully enhances the fault tolerance of these models.

- We conducted experiments with our FPGA devices by changing the bit-width settings, datasets (AudioMNIST), number of time steps, batch size, distributions of data samples in batches, and fault occurrence (static vs dynamic).

We revised our paper to reflect these valuable changes gathered by the author-reviewer discussion. For example, we posted the additional experimental results in Section B of Appendix in the revised manuscript. **After posting our thorough rebuttal, Q1C8 raised the soundness score from 2 to 3 and the overall rating from 4 to 8 on November 24th. Please check Q1C8's additional response and the revision history.** We believe that we have addressed Q1C8's concerns about the environmental assumption (on-chip learning availability in gradient-based neuromorphic devices), reproducibility, and effectiveness of our mechanism in various hardware settings.

## Reviewer Nuth

**Main concerns: the additional discussion about our mechanism and other lightweight approaches based on self-testing and dropout**

To address concerns of Nuth, we read the papers that Nuth recommended and thoroughly compared these works to our work. Moreover, we addressed the concern about the validity of our experiments.

- We demonstrated that our comparison is valid because the prior works do not rely on device-specific characteristics such as non-linearities in weight updates. This point enables their algorithms to be faithfully reproduced on general digital substrates, such as FPGA devices

- We discussed our mechanism with the input ignorance and neuron dropout methods in detail. Compared to these works, our mechanism exhibits smaller overhead than these works when the SNN models are deep, which are widely used in current SNN societies.

- We thoroughly discussed our mechanism (continual execution) with the existing approaches based on self-testing (periodic execution). We presented that our execution manner enhances the fault tolerance of SNNs more significantly than these approaches due to its agility. In addition, we mentioned that our mechanism's overhead is manageable in neuromorphic devices through references.

- We discussed our mechanism with the forward-pass resilience mechanisms in various aspects. We emphasized that our mechanism has a small overhead in SNNs, especially in large SNNs, and the interconnection overhead between the neuromorphic device core and additional processor for flow control is not a significant burden through citing prior works.

- We presented the advantages, disadvantages, and major application fields of self-testing and repair systems, and our mechanism in detail. The continual execution of our mechanism allows it to improve the fault tolerance of SNNs with high agility. Owing to this advantage, our mechanism is suitable for neuromorphic devices that support on-chip learning or require strict real-time performance.

- We additionally conducted experiments to compare our mechanism's fault mitigation ability and overhead to that of the existing lightweight approaches mentioned by reviewer Nuth. Our experimental results demonstrate that our mechanism exhibits more significant fault mitigation ability than the existing approaches, with smaller overhead than these approaches.

We revised our paper to reflect these valuable discussions. We made a new section named Discussion in Appendix, and posted the discussion with our mechanism and the existing light mechanisms. Furthermore, we added the experimental results of our mechanism and the existing lightweight mechanisms. After posting our thorough rebuttal, reviewer Nuth has not appeared yet and has not posted additional responses to our rebuttal.

---

### Author Response · Authors · 2025-11-27
**Comment for the new AC (1/2)**

# Thanks to the new AC

We understand the chaotic situation caused by the data leak and the subsequent rollback and appreciate the new AC's great efforts on our paper. Since the discussion period has been cut short and the numerical scores have been reverted to their pre-discussion state, we post a comment for the new AC to introduce the reviewers' comments, our rebuttal, and the revised paper in detail. Please sincerely read our comments for the AC, rebuttal to reviewers below, and the revised manuscript, reflecting them for the evaluation.

---

# Strengths of our paper

- **Theoretically grounded contents**: Nuth and Q1C8 mention that our paper's theoretical base is solid. Our mechanism is based on a thorough theoretical proof in Appendix, which demonstrates that it ensures a proper fault mitigation strategy according to the fault status of SNNs.

- **Intuitive and simple approach**: Nuth and Q1C8 mention that our mechanism, which is motivated by flow control in computer networks, is intuitive and easy to understand, guaranteeing solid explanations of our paper.

- **Thorough motivation study**: Q1C8 and EvpD mention that we did a thorough motivation study. This motivation study provides a fundamental analysis of our problem definition and supports the powerful fault mitigation ability of our mechanism concretely.

- **Broad experimental coverages**:  Nuth, Q1C8, and E666 confirm that we adopted our mechanism to various SNN models and compared it to six benchmarks. Furthermore, we applied different time steps, hyperparameters, and combinations of sub-mechanisms of our mechanism for in-depth studies.

- **Hardware validation**: As Nuth, Q1C8, and EvpD wrote, we conducted experiments with an actual FPGA device and demonstrated that our mechanism works well with hardware environments.

---

# Concerns of the reviewers

The four reviewers gave us the following valuable comments.

- **The two reviewers (E666 and EvpD)** pointed out **the presentation quality of our paper.**

- **Reviewer Q1C8** pointed out **the environmental assumption of hardware learning, reproducibility of the FPGA-based experimental settings, and our mechanism's effectiveness in other various settings (faults caused by FPGA components, large SNN models on the FPGA device, 1D datasets, various batch settings, dynamic fault environments, and unsupervised learning SNNs).**

- **Reviewer Nuth** requested **to discuss our mechanism with other lightweight approaches based on self-testing and dropout.**

---

# Summary of our rebuttal and the reviewers' additional responses

## Reviewer E666 and Reviewer EvpD

**Main concerns: the clarification on the main idea, scope, equations, and presentation**

To address concerns of reviewers E666 and EvpD, we thoroughly explained the main idea, our mechanism's procedures, and the equations used for our mechanism.

- We clarified the main idea and concept in the revised manuscript (E666).

- We clearly presented that our paper focuses on the hardware faults occurring in the neuromorphic devices (EvpD).

- We added an overview of the proposed mechanism (EvpD).

- We changed the position of the FPGA-based contents in the paper (EvpD).

- We provided the detailed explanations of the equations to the reviewer and added them to the revised manuscript (E666).

- We presented citations of the datasets to the reviewer and added them to the revised manuscript (E666).

We revised our paper to reflect these valuable changes gathered by the author-reviewer discussion. For example, we added thorough explanations about equations to Section 5 of the revised manuscript. **After posting our responses, EvpD raised the presentation score from 1 to 2 and the overall rating from 2 to 4 on November 14th. Please check EvpD's additional response and the revision history.** E666 has not appeared yet and has not posted additional responses to our rebuttal. We believe that we have addressed their concerns about the presentation issue.

---

### Author Response · Authors · 2025-11-29
**Notification**

# Ratings and Confidences before the rollback

- **Reviewer EvpD**: Rating: 4 / Confidence: 3

- **Reviewer Q1C8**: Rating: 8 / Confidence: 5

- **Reviewer Nuth**: Rating: 4 / Confidence: 5

- **Reviewer E666**: Rating: 4 / Confidence: 2

**Average rating**: 5.00 (Min: 4, Max: 8)

**Average confidence**: 3.75 (Min: 2, Max: 5)

---

# Scores before the rollback

- **Reviewer EvpD**: Soundness: 2 / Presentation: 2 / Contribution: 2

- **Reviewer Q1C8**: Soundness: 3 / Presentation: 4 / Contribution: 3

- **Reviewer Nuth**: Soundness: 3 / Presentation: 4 / Contribution: 2

- **Reviewer E666**: Soundness: 3 / Presentation: 2 / Contribution: 3

---

# Highlight colors in the revised manuscript

- **Reviewer EvpD**: Orange

- **Reviewer Q1C8**: Blue

- **Reviewer Nuth**: Green

- **Reviewer E666**: Red

---

After we submitted a detailed rebuttal with many additional experimental results and clarifications, two reviewers updated their ratings and scores. In particular, reviewer **Q1C8**, who reported the most weaknesses with **confidence 5**, increased **the rating from 4 to 8 on November 24th**. Reviewer **EvpD** also increased **the rating from 2 to 4 on November 14th** after our revisions improved the presentation quality. In addition, Q1C8 increased the soundness score from 2 to 3, and EvpD increased the presentation score from 1 to 2. **These changes can be confirmed in the revision history of their review comments and in our thorough rebuttal comments below.**

Reviewers E666 (presentation issues) and Nuth (request for further discussion) did not return any comments until November 28th. However, we have thoroughly addressed their concerns in our rebuttal and in the revised manuscript.

We highlighted the revised parts of our paper with orange (EvpD), blue (Q1C8), green (Nuth), and red (E666). Please refer to the colors when confirming the changes in the revised manuscript.

---

### Meta-Review · Area_Chair_hwHj · 2026-01-06

**Summary:**

This paper proposes a simple input control mechanism for improving the fault tolerance of hardware-implemented spiking neural networks (SNNs). The core idea is to fragment input samples into smaller pieces based on a sensitivity analysis and a Gini-coefficient–based balancing strategy, inspired by flow control in computer networks. The mechanism is designed to mitigate a “bottleneck problem” where hardware faults cause pre-activation drift and reduce the effective learning capacity of SNNs. The authors support their approach with extensive motivation analysis, theoretical derivations, software simulations, and FPGA-based experiments.

During the review process, reviewers agreed that the paper is technically thorough, well-written, and supported by an unusually extensive set of experiments. The authors provided an exceptionally detailed rebuttal, adding large amounts of new experimental evidence (including additional fault models, STDP-based SNNs, 1D datasets, deeper models, dynamic fault scenarios, latency–accuracy trade-offs, and hardware resource analyses), clarifying presentation issues, and improving reproducibility. As a result, multiple reviewers increased their scores after the rebuttal.

However, despite the strengthened empirical evidence and the authors’ strong engagement, the discussion converged on several fundamental concerns that remain unresolved after rebuttal. These concerns relate primarily to the conceptual framing of the problem, the realism and consistency of the assumed learning scenario, and the incremental nature of the contribution relative to existing fault-tolerance mechanisms. These issues ultimately motivate the recommendation.

**Reviewer Concerns:**

Concerns addressed by the rebuttal
- Presentation clarity and organization (EvpD, E666):
The authors significantly improved clarity by revising section titles, adding an overview of the mechanism, clarifying fault models, and expanding explanations of equations and metrics (e.g., Gini coefficient, normalization, decoding).
- Breadth of experimental validation (Q1C8, Nuth):
The rebuttal added extensive experiments across datasets, architectures, fault types (including spike-level faults), learning rules (BPTT and STDP), hardware configurations, and latency–accuracy trade-offs, including FPGA-based validation.
- Comparisons with lightweight and test-based approaches (Nuth):
The authors added a detailed discussion and additional experiments contrasting the proposed method with dropout, input suppression, forward-pass resilience, and self-test–based methods.
- Reproducibility and hardware details (Q1C8):
The authors provided additional documentation, VHDL scripts, bit-width analyses, and hardware configuration details.

Concerns that remain outstanding
- Ambiguity and tension in the problem setting:
A central motivation of the paper relies on surrogate-gradient collapse during training under hardware faults. While the authors argue that on-chip gradient-based learning is emerging and relevant, this assumption remains only partially aligned with current large-scale neuromorphic practice, where faults most commonly manifest during inference after offline training. Although the authors demonstrate applicability to STDP-based models, the theoretical framing and motivation remain largely tied to a specific and still niche training regime, limiting the generality of the contribution.
- Incremental nature of the core idea:
While input fragmentation guided by sensitivity analysis is effective, it can be viewed as a refinement or recombination of existing concepts—input scaling, temporal decomposition, and load balancing—rather than a fundamentally new fault-tolerance paradigm. The mechanism improves robustness primarily by reducing instantaneous input energy and avoiding extreme activations, an effect that overlaps conceptually with prior lightweight mitigation strategies, even if implemented differently.
- Practicality across neuromorphic substrates remains unclear:
Although the approach is demonstrated on FPGA platforms, the reliance on continuous input fragmentation, sensitivity estimation, and external control raises questions about scalability and practicality on other neuromorphic substrates (e.g., analog or compute-in-memory systems). These concerns are discussed but not fully resolved, and the benefits over simpler, amortized test-based or inference-time suppression methods remain context-dependent.
- Contribution strength relative to venue standards:
Reviewers generally agreed that the work is solid, well-engineered, and carefully evaluated. However, the contribution is perceived as closer to an engineering optimization or system-level refinement rather than a conceptual or theoretical advance that substantially reshapes how fault tolerance in SNNs is approached.

**Reviewer Scores:**

One reviewer initially rated the paper strongly and increased the score to a high accept after rebuttal. Other reviewers raised their scores from reject or borderline reject to marginally below threshold. Importantly, multiple reviewers explicitly stated that they would not mind rejection, reflecting limited enthusiasm despite acknowledging improvements.

---

### Decision · Program_Chairs · 2026-01-26

Reject